# The initial stages of cement hydration at the molecular level

Xinhang Xu [1], Chongchong Qi [1,2,3] ✉, Xabier M. Aretxabaleta [4], Chundi Ma[1], Dino Spagnoli [2] & Hegoi Manzano [4]

Cement hydration is crucial for the strength development of cement-based materials; however, the mechanism that underlies this complex reaction remains poorly understood at the molecular level. An in-depth understanding of cement hydration is required for the development of environmentally friendly cement and consequently the reduction of carbon emissions in the cement industry. Here, we use molecular dynamics simulations with a reactive force field to investigate the initial hydration processes of tricalcium silicate ($C_3S$) and dicalcium silicate ($C_2S$) up to 40 ns. Our simulations provide theoretical support for the rapid initial hydration of $C_3S$ compared to $C_2S$ at the molecular level. The dissolution pathways of calcium ions in $C_3S$ and $C_2S$ are revealed, showing that, two dissolution processes are required for the complete dissolution of calcium ions in $C_3S$. Our findings promote the understanding of the calcium dissolution stage and serve as a valuable reference for the investigation of the initial cement hydration.

The emissions gap report published by the United Nations Environment Programme in 2022 shows that the international community is far from meeting the greenhouse gas (GHG) reduction goals set by the Paris Agreement[1]. The world must urgently reduce its GHG emissions to avoid a global climate catastrophe[2]. As the primary contributor to global GHG emissions (in terms of the sum of direct and indirect emissions), the industrial sector has received widespread attention as a crucial target area for reducing emissions[1,3]. The reduction of carbon emissions from global cement production has been identified as a key element in reducing industrial emissions[4,5]. To date, efforts to reduce carbon emissions from cement production have focused on initiatives to improve energy efficiency[6], such as the recovery of waste heat[7], the use of alternative fuels[8], and the development of energy-efficient machinery[9]. However, given the growing demand for cement production, relying solely on the above industrial modernization methods cannot reduce cement-related carbon emissions to the levels required to meet global emission reduction targets[10]. It is necessary to act on the material itself.

Ordinary Portland cement is the most widely used type of cement[11], and its main constituents include alite (50–70 wt.%), belite (15–30 wt.%), ferrite (5–15 wt.%), and aluminate (5–10 wt.%)[12]. Alite and belite, which comprise the highest proportions of cement clinker, are the impure forms of tricalcium silicate ($Ca_3SiO_5$ or $C_3S$) and dicalcium silicate ($Ca_2SiO_4$ or $C_2S$), respectively[13]. The hydration processes of $C_3S$ and $C_2S$ are considered to be the greatest contributors to strength development in cement-based materials[14]. Thus, an in-depth understanding of the hydration mechanism of $C_3S$ and $C_2S$ has key implications for the design of environmentally friendly cements[15] and represents one of the most radical approaches for reducing carbon emissions in the cement industry[16,17].

The hydration processes of $C_3S$ and $C_2S$ are very complex and are influenced by multiple coupled parameters[18]. Atomic and molecular-scale simulations can exclude the influence of impurities and provide a range of accurate insights into the initial hydration process and reactions. The molecular or dissociative adsorption of water molecules on the surface of $C_3S$ and $C_2S$ is considered to be the first step of the

[1]School of Resources and Safety Engineering, Central South University, Changsha, Hunan 410083, China. [2]School of Molecular Sciences, University of Western Australia, Perth, WA 6009, Australia. [3]School of Metallurgy and Environment, Central South University, Changsha, Hunan 410083, China. [4]Department of Physics, Faculty of Science and Technology, University of the Basque Country UPV/EHU, Barrio Sarriena s/n, Leioa, Bizkaia 48940, Spain. ✉e-mail: chongchong.qi@csu.edu.cn

hydration process[19–21]. In the case of single water adsorption, dissociative and molecular adsorption processes are preferred on $C_3S$ and $C_2S$, respectively[22]. However, for bulk water adsorption, the water molecules on $C_3S$ prefer molecular adsorption due to the presence of strong hydrogen bonding within the bulk water[23]. $C_3S$ generally tends to be more reactive than $C_2S$[24]; this behavior is related to the ionic oxygen in $C_3S$[25], which is more loosely bound than covalent oxygen and has higher degrees of freedom[23] and fewer valence electrons[24]. After water adsorption, protons diffuse to the interior of the material by proton hopping[26] while Ca ions are detached from the surface[27]; this constitutes a key step in the initial hydration process. However, numerous coupled chemical reactions influence the initial hydration process, the dissolution pathway of Ca ions remains unclear. The most recent studies have only investigated the dissolution pathways of two specific coordination Ca ions in $C_3S$[19]. Hence, it is necessary to simulate the initial hydration process to reveal the general dissolution pathway of Ca ions.

In this study, we use ReaxFF reactive force field simulations to reveal the initial hydration processes of $C_3S$ and $C_2S$ from multiple perspectives, including chemical bonding, dissolution of Ca ions, and surface structural evolution. Here, we present the breakthrough discovery of a new dissolution process of Ca ions on the $C_3S$ surface and reveal, for the first time, the general dissolution pathways of Ca ions from $C_3S$ and $C_2S$ surfaces. The unbiased molecular dynamics (MD) simulation, as validated by the free energy calculation, reveals that the hydroxylation state of the neighboring atoms is a key factor which enables Ca desorption. A new ligand teeth structure is discovered during the dissolution of Ca ions, which is critical for its dissolution process. In addition, the solid/water arrangement near the interface is characterized after the initial hydration process, including the appearance of aqueous layers and the small pores favoring H transfer in M3-$C_3S$. Overall, our discoveries promote the understanding of the dissolution process of Ca ions in the study of the initial hydration processes of $C_3S$ and $C_2S$.

## Results

### Initial hydration process

To characterize the initial hydration process, the numbers of Ca-$O_w$ bonds (where $O_w$ represents oxygen ions from water), O-H bonds, and dissolved Ca ions are investigated over time. The near-surface ranges of M3-$C_3S$ (010) and β-$C_2S$ (100), where the initial hydration process mainly occurs[28], are illustrated (Fig. 1a). During the initial hydration process, the number of Ca-$O_w$ bonds continues to increase for both M3-$C_3S$ (010) and β-$C_2S$ (100) (Fig. 1b). Rapid increases in bonding in M3-$C_3S$ (010) and β-$C_2S$ (100) are observed within 9.5 ps and 1.7 ps, respectively, due to the interaction and reactions of water with the initially bare surface. The under-coordination of surface Ca ions is significantly reduced, or even depleted, after the rapid water adsorption, resulting in the slow increase of Ca-$O_w$ bonds afterwards. M3-$C_3S$ (010) then exhibits a second rapid increase (3.5 new Ca-$O_w$ bonds $nm^{-2}$) in the time range between 1.5–6 ns, which is 84.21% more than that observed for β-$C_2S$ (100) (1.9 new bonds $nm^{-2}$) over the same period. This second rapid increase in M3-$C_3S$ (010) is due to the dissolution of Ca ions, a phenomenon that is not observed in β-$C_2S$ (100) (Fig. 1c). There are two steps in the Ca ion dissolution mechanisms that promote this initial hydration. In the first one, the water molecules dissociation break the Ca-$O_s$ bonds (where $O_s$ represents surface oxygen ions) and forms a new Ca-$O_w$ and $O_s$-H bonds[19]. Then, the detachment of Ca ions from the surface creates vacancies that allow $O_w$ to form new Ca-$O_w$ bonds with the second layer of Ca ions (Supplementary Fig. 1). Notably, we find that the more rapid increase of Ca-$O_w$ bonds around the M3-$C_3S$ (010) remains valid when the influence of initial surface area and initial exposed Ca is excluded (Supplementary Fig. 2a and Supplementary Fig. 2b).

In the following text, we investigate the structure of $H_{ab}$ ions (where $H_{ab}$ represents H ions bonded to $O_s$ and $O_w$ with Ca ions) to characterize the initial hydration process. Most of the H atoms are still bonded to $O_w$ forming water molecules bonded to the surface (Fig. 1d). Since the $O_w$ ions are adsorbed on Ca ions, the $O_w$-$H_{ab}$ bonds exhibit an increasing trend that is similar to the Ca-$O_w$ bonds. Notably, an increase of $O_w$-$H_{ab}$ bonds (0.78 new bonds $nm^{-2}$ $ns^{-1}$) is observed in M3-$C_3S$ (010) between 1.5–6 ns, which is identical to the increase in the number of Ca-$O_w$ bonds. In β-$C_2S$ (100), an increase of 36 $O_w$-$H_{ab}$ bonds (0.79 new bonds $nm^{-2}$ $ns^{-1}$) occurs at 1.5–6 ns, which is 1.9 times the number of increased Ca-$O_w$ bonds (0.41 new bonds $nm^{-2}$ $ns^{-1}$). These results confirm that water adsorption on Ca ions is not completely achieved by molecular adsorption of water, and there is a generation of two $O_w$-$H_{ab}$ bonds per Ca-$O_w$ bond by the dissociation of water molecules. The differences in the rates of $O_w$-$H_{ab}$ bond increase may be due to the M3-$C_3S$ (010) tending to form more Ca-$O_w H_{ab}$-Ca structures (Supplementary Fig. 3) than β-$C_2S$ (100) during Ca ions dissolution. In this structure, multiple Ca-$O_w$ bonds are formed along with one $O_w$-$H_{ab}$ bond, resulting in the decreased ratio of Ca-$O_w$ and $O_w$-$H_{ab}$ (smaller than two in the molecular adsorption of water). Note that the presence of Ca-$O_w H_{ab}$-Ca still dominates in β-$C_2S$ (100) even at 1.5–6 ns, though the increased ratio of Ca-$O_w$ and $O_w$-$H_{ab}$ is high (Supplementary Tables 1, 2 and Note 1). A more detailed discussion is provided in the dissolution process of Ca ions section.

Due to the dissociation of water, a small fraction of the absorbed H ions are also transferred from $O_w$ ions to $O_s$ ions during the initial hydration process. The presence of more $O_i$-H bonds (where $O_i$ represents $O_s$ ions that are not bonded to Si ions) than $O_{si}$-H bonds (where $O_{si}$ represents $O_s$ ions that are bonded to Si ions) in M3-$C_3S$ (010) indicates that $O_i$ ions are more favorable adsorption sites for H ions than $O_{si}$ ions, as suggested in the literature[23,24].

### Dissolution process of Ca ions

In M3-$C_3S$ (010), we found that two dissolution processes are required for the complete dissolution of Ca ions. In the first process, the Ca ions form new Ca-$O_w$ bonds while breaking Ca-$O_s$ bonds. By definition, a Ca ion becomes dissolved ($Ca_{dis}$) once all the Ca-$O_s$ bonds for that Ca ion are broken[19]. However, these dissolved Ca ions do not achieve free movement in the water layer since they are still bound to $O_w$ ions that have been absorbed by other undissolved Ca ions (Fig. 2a). Hereafter, we use $Ca_{dis}$-$O_w$ (tridentate) to denote this structure, and use ligand teeth to indicate the $Ca_{dis}$-$O_w$ bonds absorbed by other undissolved Ca ions. The dissolved Ca ions within $Ca_{dis}$-$O_w$ (tridentate) must undergo a new bond-breaking process to become free after the first dissolution process.

The time required for this new bond-breaking process, i.e., the second dissolution process, can be tens of thousands of times longer than that required for the first dissolution process. Here, six-coordinated dissolved Ca ions are investigated because they account for 95.8% of all the dissolved Ca ions that appear within 40 ns of the initial hydration process (Supplementary Fig. 4a). We discover that this new bond-breaking process starts from Structure 1 (Fig. 2a), which consists of three $Ca_{dis}$-$O_w$H (tridentate) bonds and three $Ca_{dis}$-$H_2O_w$ bonds. The dissolved Ca ions then continuously transform: the ligand dissociate and form new Ca-$H_2O_w$ bonds, transforming progressively from tridentate to bidentate to monodentate (Fig. 2b). The transformation into and from these three structures is reversible. For all six-coordinated dissolved Ca ions, 92.2% of the Ca ions structures belong to Structures 1 to 3 (Supplementary Fig. 4b). The other structures of the six-coordinated Ca ions have polydentate structures; such structures are also present in the middle layer of the C-S-H structure[29].

From the unbiased trajectory, we perform umbrella sampling (US) simulations along the original transition path in two scenarios (Fig. 2c). First without any constraint on the system, which represents the MD

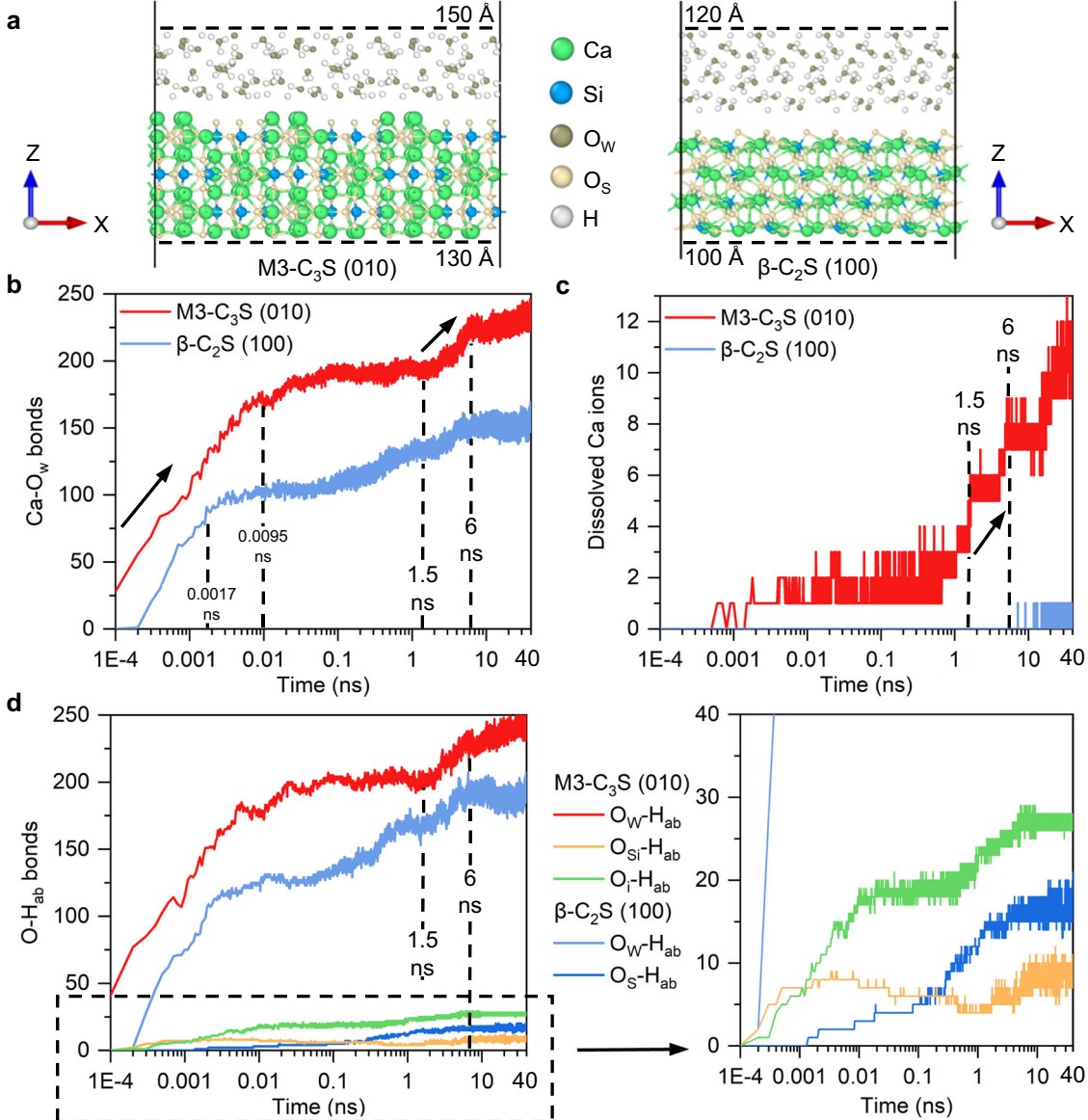

**Fig. 1 | Initial hydration process of M3-$C_3S$ (010) and β-$C_2S$ (100). a** The near-surface ranges of the slab model before MD simulations. **b** The number of Ca-$O_w$ bonds. **c** The number of dissolved Ca ions. **d** The number of O-$H_{ab}$ bonds. Note that Ca, Si, and O ions are used as indicators when counting Ca-O, Si-O, and O-H bonds, respectively. The corresponding bonds are counted when these indicator ions are observed within the near-surface range, even if the associated bonds are beyond this range. Dissolved Ca ions in (**c**) are defined as having no Ca-$O_s$ bonds but do not represent free Ca ions in solution, which is further detailed in the Dissolution process of Ca ions section. The zoomed-in region of the dashed box (0–40 in the y-axis) is also presented.

simulations done in this study (the H-jump scenario). Second, constraining the H hopping to prevent O hydroxylation, which mimics the scenario in common density functional theory (DFT) simulations or unreactive force field studies (the no H-jump scenario). When H hopping is allowed, a dissociated water molecule creates a hydroxyl group in a surface oxygen coordinated to Ca ion when Ca ion is at ~5.25 Å. That makes the Ca-O, where the O ion is also bonded to H, bond weaker and therefore facilitates the movement of the Ca ion, which reaches a stable state at >5.55 Å. The energy barrier is less than 1 kcal mol⁻¹ (4 kJ mol⁻¹). When the water molecule is not allowed to dissociate, the Ca-O bond remains strong, and increasing the distance for the Ca ion desorption is unfavorable. The free energy increases constantly up to 4 kcal mol⁻¹ (16 kJ mol⁻¹), and no final stable state is found. This illustrates the necessity of unbiased simulations to understand the dissolution mechanism before applying metadynamics or targeted molecular dynamics to characterize the energy barriers.

The second dissolution process can be divided into four stages (Fig. 2d). During the first stage (0.0017–0.7 ns), the dissolving Ca ions exist primarily as Structure 1. The appearance of Structure 1 lags in comparison to the first section Ca dissolution time of 0.6 ps (Fig. 1c). This is due to the presence of five- and seven-coordinated dissolved Ca ions that represent transitional structures[19]. In the second stage (0.7–5 ns), Ca ions switch between Structures 1 and 2, while Structure 3 begins to appear. In the third stage (5–20 ns), most of the dissolving Ca ions exist as Structure 2. Finally, stable Ca ions in the form of Structure 3 are clearly observed in the fourth stage (20–40 ns).

Notably, the second dissolution process is observed for Ca ions with different initial coordination numbers (Supplementary Fig. 5a, b), which can be regarded as a general bond-breaking process. The ligand teeth exist in two dissolution processes, which may explain why the observed $O_w$-$H_{ab}$ bond increase is less than twice that of the Ca-$O_w$ bonds (Fig. 1b, d).

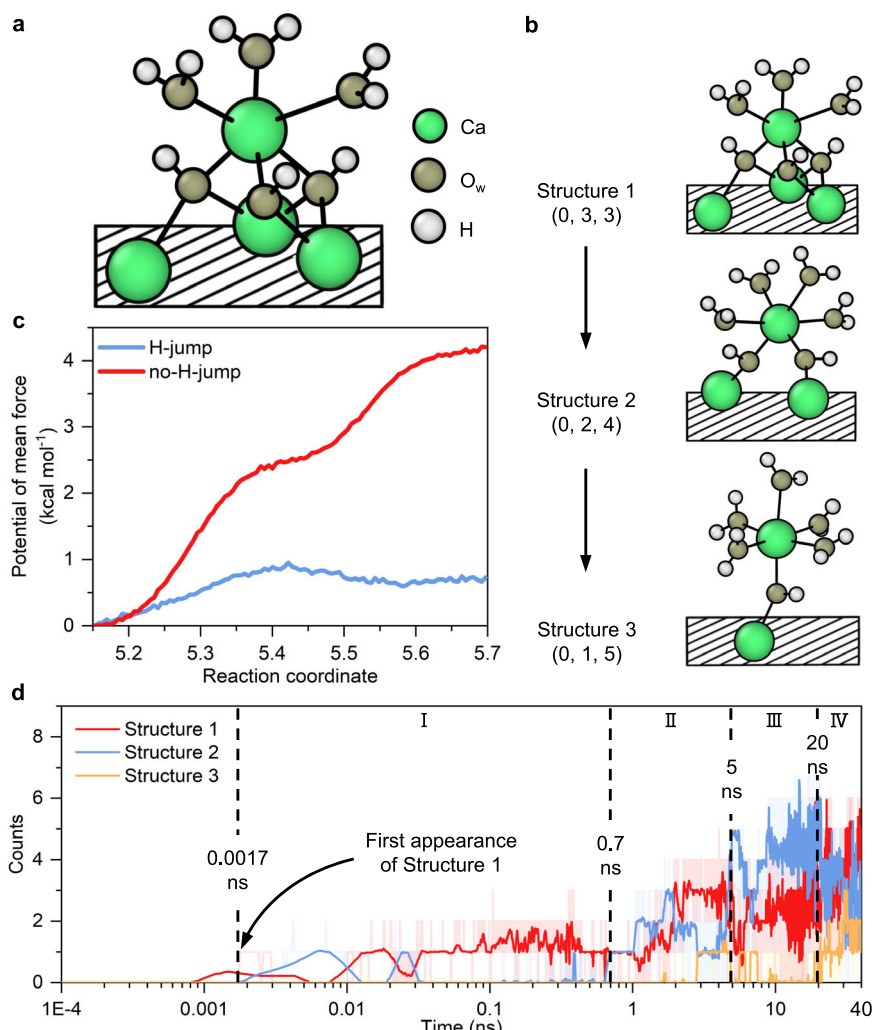

**Fig. 2 | The second dissolution process of Ca ions from M3-C₃S (010). a** The initial Ca ion structure of the second dissolution process, namely Structure 1. **b** The conceptualized flowchart illustrating the second dissolution process. **c** The potential of mean force of the transition from Structure 2 to 3. The transformation from Structure 2 to 3 might require a higher energy barrier than that from Structure 1 to 2 since it takes a much longer transformation time. **d** The evolution of Ca ions' structures with time. The structures of Ca ions are presented in parentheses as the number of Ca-O$_s$ bonds, the number of ligand teeth, and the number of Ca-H$_2$O$_w$ bonds. The surface is shown as the shaded box. The raw data curves in (**d**) are partially transparent and the smoothed curves are highlighted.

In β-C₂S (100), we investigated the process by which Ca ions detach from O$_s$ ions because almost all Ca ions remain at the first dissolution process within 40 ns of the initial hydration process. In other words, no stable detachment of Ca ions from O$_s$ ions is observed, and the Ca ions can only completely break the Ca-O$_s$ bonds at some instant (Fig. 1c). In the near-surface range, the initial coordination numbers of Ca ions are five, six, and eight (Supplementary Fig. 6). These Ca ions all undergo a transition step involving five Ca-O$_s$ bonds during the first dissolution process. In this step, the coordination numbers and structures of the Ca ions are complex (Supplementary Table 3). Numerous reaction pathways exist in this step due to the different initial coordination numbers and atomic positions of the Ca ions. After this transition step, the dissolution process is dominated by six-coordinated Ca ions (Supplementary Fig. 7), which we investigate to quantify the dissolution process. Note that only the most prevalent structure of six-coordinated Ca ions in each step is selected for illustration (Fig. 3a); all other structures are presented in the Supplementary information (Supplementary Fig. 8).

The Ca ions with four Ca-O$_s$ bonds are considered to be in the Step 1 following the transition step; in this instance, all Ca ions break at least one Ca-O$_s$ bond irrespective of the initial coordination number. The Ca ions transform sequentially from Step 1 to Step 4 by breaking Ca-O$_s$

bonds and forming new Ca-O$_w$ bonds. There is a higher proportion of Ca ions with ligand teeth in most dissolution steps. In particular, the structure of Ca ions without ligand teeth bonds only accounts for 0.05% of Ca ions in Step 4 (Supplementary Fig. 8). Even in Step 2, where Ca ions with ligand teeth only account for 43.07% of the total, this structure is the earliest to appear (Supplementary Fig. 9). This indicates that ligand teeth are an important element of the dissolution process of β-C₂S (100), consistent with observations from M3-C₃S (010). The dissolution process of Ca ions in β-C₂S (100) tends to form fewer ligand teeth than M3-C₃S (010) (Figs. 2a, 3a). The above results indicate a greater degree of molecular adsorption of water on Ca ions of β-C₂S (100), resulting in a faster rate of O$_w$-H$_{ab}$ bond increase than in M3-C₃S (010) (Fig. 1d).

The representative structures of each step appear sequentially with time (Fig. 3b). In this section, the structures of Ca ions are described by the number of Ca-O$_s$ bonds, the number of ligand teeth, and the number of Ca-H$_2$O$_w$ bonds, written in parentheses. The Ca ions mainly exist as (4, 1, 1) structures between 0.5 ps and 0.1 ns. The number of Ca ions that exist as (3, 0, 3) and (2, 1, 3) structures increases rapidly between 0.1 ns and 0.6 ns. The numbers of Ca ions with (4, 1, 1), (3, 0, 3), and (2, 1, 3) configurations are relatively similar between 0.6 ns and 2.3 ns. The Ca ions clearly exist as the (1, 1, 4) structure after 2.3 ns,

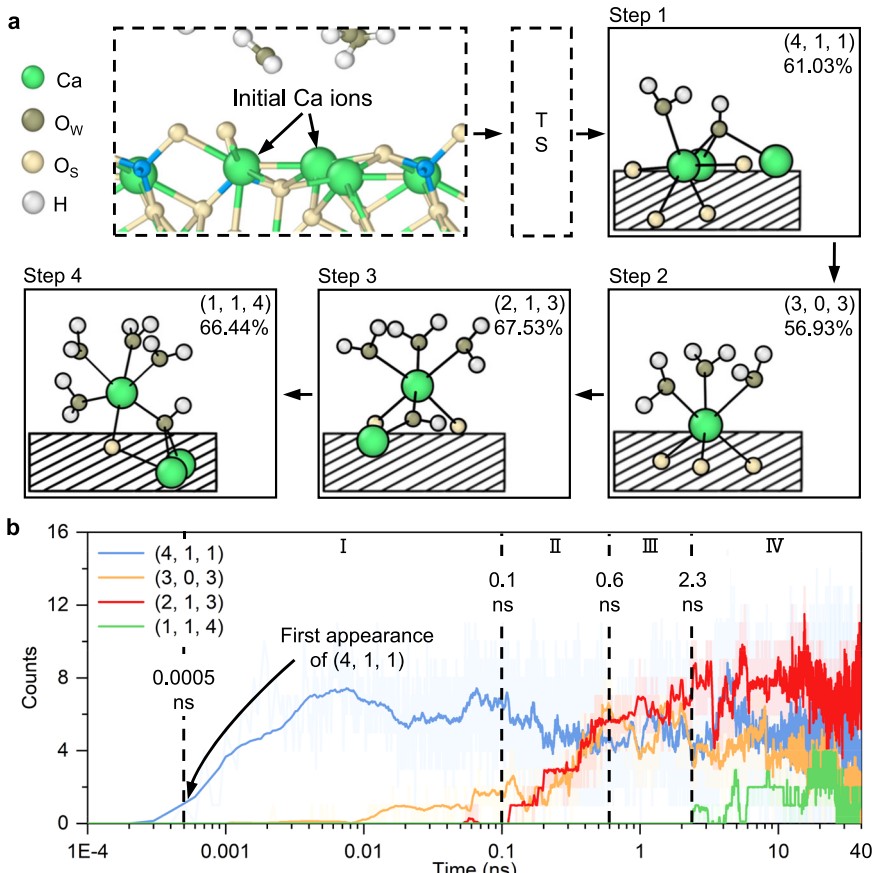

**Fig. 3 | The first dissolution process of Ca ions from β-C₂S (100). a** The conceptualized flowchart illustrating the dissolution process. **b** The evolution of Ca ions' structures with time. The structures of Ca ions are presented in parentheses as the number of Ca-O$_s$ bonds, the number of ligand teeth, and the number of Ca-H$_2$O$_w$ bonds. The surface is shown as the shaded box. The raw data curves in (**b**) are partially transparent and the smoothed curves are highlighted.

while the number of Ca ions that exist as the (2, 1, 3) structure remains high.

### Solid/Water interface and surface structure

In M3-C₃S (010), we observe that the interface can be divided into five regions at 40 ns (Fig. 4a). The first region corresponds to the bulk, up to 133 Å, which is characterized by the absence of H ions from water or OH. The second region (133–139 Å) is the gugenheim interface[30], in which the bulk structure is gradually lost and the solid and water coexist. The second region penetrates to the interior with the hopping of H ions and the reaction of O$_w$ with Ca ions (Supplementary Fig. 10). The third region is the stern layer (139–142 Å), and consists of dissolved Ca ions still linked to the surface and water molecules, with an O$_w$ density similar to that of bulk water. The third region gradually appears with the dissolution of Ca ions (Supplementary Fig. 10). We note that the third region contains not only free water molecules but also dissociative and molecularly adsorbed water molecules. The diffuse (fourth region) layer (142–147 Å) is defined as the transition layer, in which only a small amount of Ca ions and almost no dissociative adsorbed water molecules are present. It mainly consists of free and molecularly adsorbed water molecules. The fifth region is considered to be the bulk liquid and consists entirely of free water molecules. The same regions are also observed in β-C₂S (100) (Fig. 4b and Supplementary Fig. 11). The atomic and structural density of these four layers of β-C₂S (100) are similar to those in M3-C₃S (010).

Next, we present a comprehensive comparison of the surface structures before and after the initial hydration process. In M3-C₃S (010), the main peak of the radial distribution function of Ca-O$_s$ is located at 2.35 Å at 0 ns (Fig. 5a). The structure of M3-C₃S (010) is

characterized by numerous small peaks after 2.35 Å, showing clear long-range ordering. At 40 ns, the maximum value of the main Ca-O$_s$ peak decreases and shifts slightly to the right due to the complete detachment of some Ca ions from O$_s$ ions. Most of the small peaks disappear and the curve becomes smoother, indicating that the long-range order of M3-C₃S (010) is affected by the initial hydration process, i.e., the crystalline structure of M3-C₃S (010) is amorphized to some extent. The maximum value of the main Si-O$_s$ peak decreases at 40 ns; however, the peak intensity is still relatively high. This indicates that there is still a strong interaction between Si and O$_s$ ions despite being affected by the initial hydration process. For β-C₂S (100), both Ca-O$_s$ and Si-O$_s$ exhibit a greater number of small peaks at 0 ns than M3-C₃S (010), indicating a more ordered surface structure (Fig. 5b). At 40 ns, the maximum value of the main Ca-O$_s$ peak decreases slightly, and some small peaks are still present. This is because Ca ions are still mainly involved in the first dissolution process at 40 ns, leading to a greater degree of preservation of the crystalline structure of β-C₂S (100).

In addition, due to the initial hydration process, pores with diameters of 2–6 Å appear in the near-surface range of M3-C₃S (010) at 40 ns (Fig. 5c). These pores are beneficial to proton transfers in the near-surface region. However, such pores are not observed in the near-surface region of β-C₂S (100) at 40 ns (Fig. 5d). Large diameter pores (> 6 Å) in both structures are contributed vacuum layer and characterize the arrangement of the uppermost ions (Supplementary Fig. 12 and Note 2). At 0 ns, the large diameter pores distribution of β-C₂S (100) is more concentrated between 8–9 Å, having a higher peak value than that of M3-C₃S (010), indicating that the initial surface atomic arrangement of β-C₂S (100) is more ordered than that of M3-

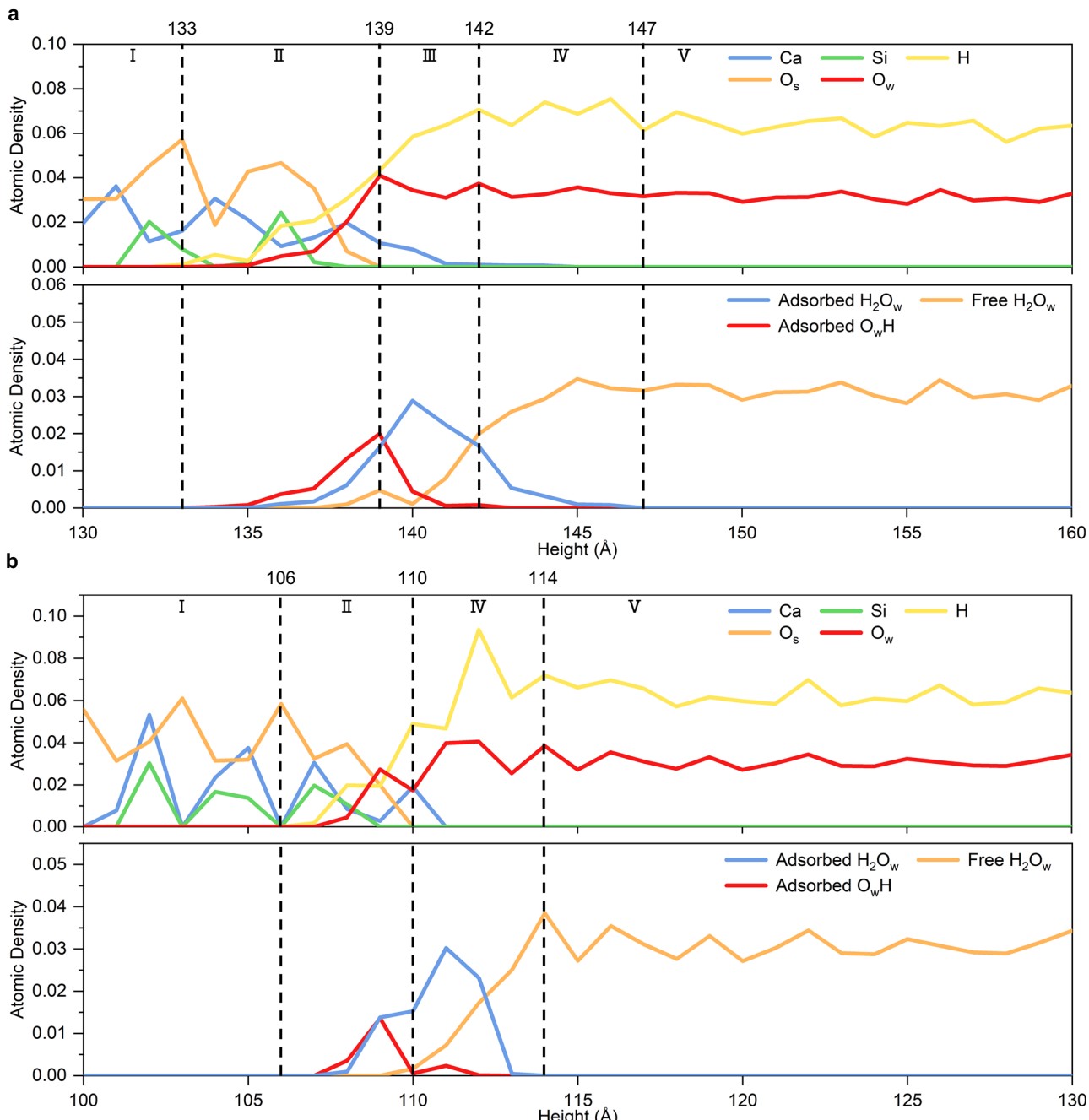

**Fig. 4 | The solid/water interface at 40 ns of the initial hydration process.** The atomic and structure density of (**a**) M3-C₃S (010) and (**b**) β-C₂S (100). The density is averaged 1 ps before 40 ns to avoid excessive volatility.

C₃S (010). At 40 ns, the range of large diameter pore sizes is larger than that at 0 ns, indicating a more disordered arrangement for the uppermost atomic structure of both M3-C₃S (010) and β-C₂S (100) after the initial hydration process.

## Discussion

During the initial hydration within 40 ns, dozens of Ca ions in M3-C₃S (010) are completely detached from O$_s$ ions and form a stable six-coordinated structure with water. The Ca ions in M3-C₃S (010) start to undergo the second dissolution process at 0.0017 ns. However, the Ca ions in β-C₂S (100) remain within the first dissolution process until 40 ns, and no stable detachment from O$_s$ ions is observed. At 40 ns, the stern layer and small pores appear in M3-C₃S (010); these features are not observed in β-C₂S (100). These

results corroborate that M3-C₃S (010) has higher reactivity than β-C₂S (100) even during the initial hydration process, and the difference originates from the crystal structure. The subsequent hydration of M3-C₃S might be further promoted by the appearance of the pore solution saturation. Furthermore, ligand teeth structures are observed during the Ca ions dissolution for both M3-C₃S (010) and β-C₂S (100). In other words, the dissolution of Ca ions is not a simple process of bonding a water molecule and breaking a Ca-O$_s$ bond, and it is not independent of the environment. Ca ions remain with a given coordination and hop only when the surface oxygen is hydroxylated. Ca ions form ligand teeth structures with O$_w$ ions, which facilitate the detachment of Ca ions from O$_s$ ions. The umbrella sampling results illustrate that unbiased simulations are necessary to understand the dissolution mechanism before

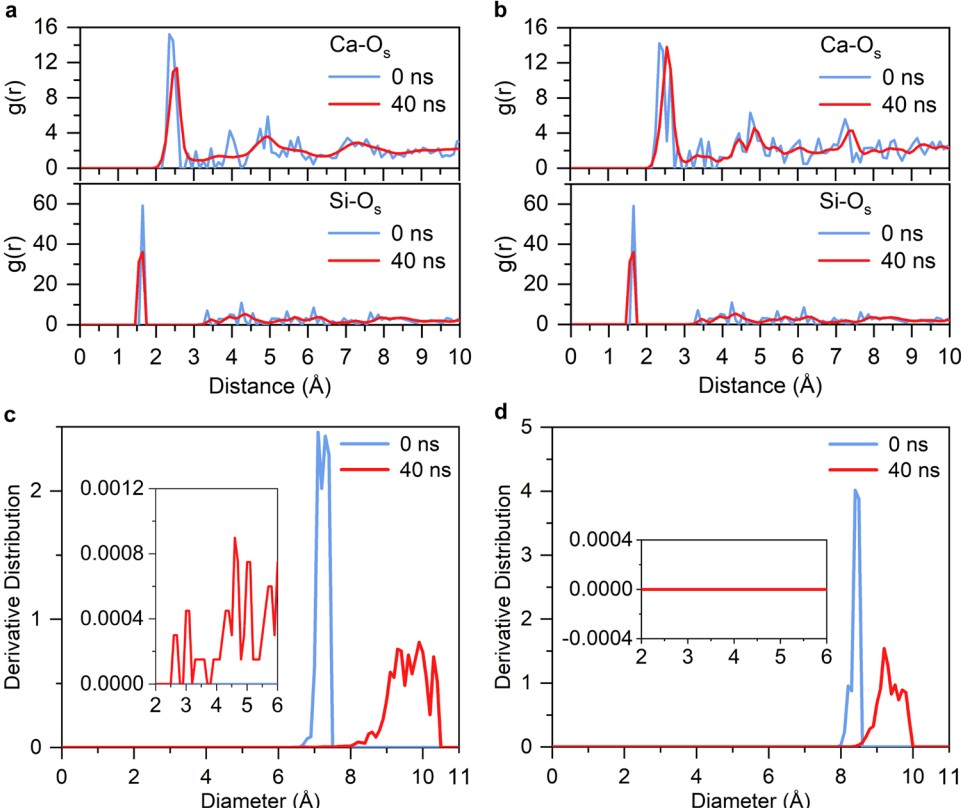

**Fig. 5 | The surface structures at 0 ns and 40 ns during the initial hydration process.** The radial distribution functions between Ca and Si ions and $O_s$ ions of (**a**) M3-$C_3$S (010) and (**b**) β-$C_2$S (100). The pore size distributions of (**c**) M3-$C_3$S (010) and (**d**) β-$C_2$S (100). The insets present the zoomed-in regions (diameters between 2 and 6 Å) in (**c**, **d**). A more detailed explanation and schematic figure of the large diameter pores are provided in the Supplementary Information (Supplementary Fig. 12 and Note 2).

applying metadynamics or targeted molecular dynamics to characterize the energy barriers. The authors note that additional validation is necessary to determine whether the revealed dissolution process can be utilized on other low-index surfaces of $C_2$S and $C_3$S, as well as other Ca-based minerals.

The higher reactivity of M3-$C_3$S (010) than β-$C_2$S (100) is due to the contribution of the $O_i$ ions in M3-$C_3$S. The O ions in the unit cell of M3-$C_3$S can be divided into four groups based on the Bader charge, with the first group ($O_i$ ions) having the lowest number of valence electrons[24]. The Ca ions have an average higher valence electron number in M3-$C_3$S than in β-$C_2$S due to bonding with the $O_i$ ions. The higher reactivity of Ca and O ions in M3-$C_3$S will facilitate the hydration reaction. Specifically, in water molecule adsorption, the adsorption energies are more negative on M3-$C_3$S (010) than on β-$C_2$S (100)[22], indicating that the water adsorption on M3-$C_3$S (010) is more energetically favorable. Meanwhile, these $O_i$ ions provide more favorable adsorption sites for H ions than $O_{si}$ ions, which thus benefits proton transfers and the formation of ligand teeth in M3-$C_3$S (010). Regarding the hydration rate, the faster hydration reaction of $C_3$S than $C_2$S also agrees well with the experimental data[31]. Specifically, the experimental undersaturated dissolution rate of anhydrous monoclinic $C_3$S in deionized water at 20 °C is determined to be -74.00 μmol m$^{-2}$ s$^{-1}$ while that of anhydrous β-$C_2$S is -16.78 μmol m$^{-2}$ s$^{-1}$.

The simulated results are consistent with the experimental results. For the simulated process, hydroxylated species have been observed at the instant of surface contact with water (1E-4 ns). This is consistent with the experimental results of Pustovgar et al.[32]. where hydroxylated $Q^0$ silicate species are formed on the particles surface before contact with a large amount of water. Also, the experiments show that only the near surface of $C_3$S particles is hydroxylated in low

amounts of polymeric silicate hydration products[31], which is consistent with the simulated process (Fig. 4a).

Moreover, it should be noted that there are significant differences in the MD simulation in this study compared to previous ones about cement hydration. Firstly, a long simulation time of 40 ns is performed in this study compared to previous MD studies (ranging from tens of ps to a few ns)[19,26]. The extension of the simulation time allows Ca ions to dissolve in an unbiased way, which is not possible in previous MD studies. Secondly, a very short time step of 0.1 fs is chosen in this study, compared to 0.2–0.5 fs in many previous MD studies[19,33], which ensures a more accurate and realistic simulation. Third, the simulation models in this study are more realistic. The simulation employs a large slab model (~23,000 in this study compared with <2000 in the literature[19]) together with the ReaxFF reactive force field, which allows for dynamic bonds formation and breaking, and accurately describes atomic motion in complex chemical environments[34,35]. Meanwhile, we do not use any accelerated dissolution method, i.e. adding biasing forces or energies to the system. Specifically, a large number of chemical reactions have occurred at the surfaces before Ca ions dissolution, and amorphous products are continuously formed at the surfaces, which is consistent with the real Ca ions dissolution process[32]. A detailed comparison of this work with two representative papers is provided in Supplementary Information (Supplementary Note 3). Furthermore, it is important to explore further the correlation between the Ca ions dissolution and the presence of OH ions, as well as the detachment process of Ca ions from the surface.

In conclusion, using unbiased MD simulations, this study makes four main contributions to the initial cement hydration, namely a new Ca dissolution process, two new general dissolution pathways, a key structure for Ca ions dissolution, and a detailed characterization of the

hydration process (Supplementary Note 4). Briefly, we find that Ca ions on the M3-$C_3S$ (010) surface must undergo two dissolution processes, whereas Ca ions in β-$C_2S$ (100) mainly remain in the first dissolution process during the studied 40-ns period. The general dissolution pathways of Ca ions in $C_3S$ and $C_2S$ are resolved in this work. A new ligand teeth structure is discovered in the dissolution of Ca ions in $C_3S$ and $C_2S$. In addition, the solid/water arrangement at the interface and the surface structure after the initial hydration process, such as the appearance of aqueous layers and the small pores favoring H transfer in M3-$C_3S$, are characterized. Moreover, the free energy calculation illustrates that the hydroxylation state of the neighboring atoms is a key factor which enables Ca ions desorption, implying the necessity of unbiased MD simulations. Due to the fast reactivity of dissolution at the initial stage of cement hydration, there are obvious technical limitations to solve the dissolution mechanism at the molecular level through laboratory experiments[36]. The detailed characterization of the cement/water interface in this study will provide an important reference for revealing the dissolution mechanism of initial cement hydration and clarifying the reaction pathway for Ca ions dissolution. Overall, these results promote the current understanding of the dissolution process of Ca ions and serve as a valuable reference for the investigation of the initial cement hydration.

## Methods

### Surface model

The M3-$C_3S$[37] and β-$C_2S$[38] polymorphs were chosen for the study due to their prevalence in industrial clinkers. The initial structures of M3-$C_3S$ and β-$C_2S$ were taken from experimental Xray diffraction determinations[39,40], and the low-index surfaces prepared from DFT optimized unit cells. The (010) surface with the lowest surface energy (1.00 J m$^{-2}$) among all symmetric surfaces was chosen for M3-$C_3S$[22]. The (100) surface with the lowest surface energy (0.84 J m$^{-2}$) was chosen for β-$C_2S$[22].

The optimization of unit cells and low-index surfaces was done using DFT as implemented in the Vienna Ab Initio Simulation Package (VASP) version 5.4.4[23]. The Perdew–Burke–Ernzerhof (PBE) functional of the generalized gradient approximation (GGA) was chosen to approximate the exchange–correlation potential[41]. $3s^23p^64s^2$, $3s^23p^2$, $2s^22p^4$, and $1s^1$, were selected as the valence electrons of Ca, Si, O, and H, respectively. After a full convergence test, the energy cutoff and energy tolerance values were chosen to be 600 eV and $1.0 \times 10^{-5}$ eV atom$^{-1}$, respectively. For unit cells, the maximum residual force on each atom was 0.01 eV Å$^{-1}$. The k-points for M3-$C_3S$ and β-$C_2S$ were $3 \times 4 \times 2$ and $4 \times 3 \times 2$, respectively. For the low-index surfaces, the periodicity of the in-plane crystals was preserved, and the slab sizes were tested for convergence. The vacuum layer was set at 15 Å. The total thicknesses of the slabs were 43.28 and 42.39 Å for M3-$C_3S$ (010) and β-$C_2S$ (100), respectively. During the surface optimization, the lattice parameters were fixed and all atoms in the slab model could be relaxed. The maximum residual force on each atom was 0.03 eV Å$^{-1}$ and a dipole correction (DFT-D3, dispersion correction 3rd version)[42] in the z-direction was applied. The k-points for M3-$C_3S$ (010) and β-$C_2S$ (100) were $2 \times 3 \times 1$ and $3 \times 2 \times 1$, respectively. The optimized unit cell and the cleaved surfaces were presented in the Supplementary information (Supplementary Figs. 13, 14).

### MD simulation systems

The simulation systems were constructed from optimized low-index surfaces using DFT. After convergence tests, values of -150 Å and -100 Å were chosen for the vacuum thickness and the surface thickness, respectively. M3-$C_3S$ (010) and β-$C_2S$ (100) were repeated three and four times in the x- and y-direction, respectively, to avoid size effects. The initial surface areas of the simulated boxes for M3-$C_3S$ (010) and β-$C_2S$ (100) were 9.08 nm$^2$ and 10.06 nm$^2$, respectively. The vacuum region was filled with water molecules with an initial density of

0.99 g cm$^{-3}$. The detailed MD model parameters were presented in the Supplementary Information (Supplementary Table 4).

### MD simulation

MD simulations were performed in this study using the Large-scale Atomic/Molecular Massively Parallel Simulator (LAMMPS), 3 Mar 2020 version[43]. The ReaxFF reactive force field was used[44]. The force field parameters for calcium silicate crystals merged the Ca-O/H and Si-O/H sets[45–47], and the atomic positions and lattice parameters were relaxed. The conjugate gradient algorithm was used for energy minimization. The energy and force cutoff tolerance value was set at $1.0 \times 10^{-5}$ kcal mol$^{-1}$. The low and high taper radius values in the force field were 0.0 and 10.0, respectively. The charge equilibration precision was $1.0 \times 10^{-6}$. The simulation box was simulated in an isothermal–isobaric (NPT) ensemble for 100 ps of thermal equilibration[48,49], and continuous simulations were performed at 298 K until 40 ns. A Nosé–Hoover thermostat was used to control the temperature and pressure with the coupling of 50 fs. A time step value of 0.1 fs with a velocity Verlet integrator was used to ensure the required accuracy.

### Umbrella sampling simulations

The umbrella sampling simulations were done with LAMMPS and the Colvars module (17-Sep-2020). The reaction coordinate has been chosen as the main distance from the dissolving Ca atom (id = 360) to the surrounding Ca atoms (ids = 7792, 220, 2904, 260, and 20). The calculations were done at room temperature using the Noose-Hoover thermostat in the Canonical ensemble (NVT) ensemble. The time step was set to 0.1 fs and the total simulation time was 100 ps. For each scenario, 12 simulations were done where the reaction coordinate was taken from 5.15 Å to 5.7 Å in steps of 0.05 Å. For the H-jump scenario the spring constant for the umbrella simulations was set to 50 kcal mol$^{-1}$·Å$^{-1}$ and for the no-H-jump scenario was set to 200 kcal mol$^{-1}$·Å$^{-1}$. In both scenarios, we made sure that all the path was correctly sampled. To recover the free energy from the different simulations we used the weighted histogram analysis method[50], with 100 bins.

### Post-processing of data

The Open Visualization Tool (OVITO) software version 3.7.7 was used for visualization and the calculation of the radial distribution functions[51]. The data from the radial distribution functions were time-averaged. Atomic density and structures within 40 ns were calculated using Python scripts. The Ca-O, O-Si, and O-H bonding cutoffs to define first neighbors or molecules were selected to be 2.83 Å[52], 2 Å[53], and 1.2 Å[54], respectively. The pore size distribution was calculated using the zeo + + code, which uses Voronoi decomposition to describe the pore geometry of the periodic system[55,56]. A probe radius of 1.1 Å (H-atom radius)[57] was used and 100,000 points were sampled in the near-surface region from which water molecules were removed.

## Data availability

The Source data used in this study are available in the Figshare database under accession code https://doi.org/10.6084/m9.figshare.25357063[58]. The atomic coordinates of initial and final configutations in this study are available in the Material Cloud database under accession code https://doi.org/10.24435/materialscloud:sj-db[59].

## Code availability

The analysis codes used in this study are available in the Github database under accession code https://github.com/macdiii/The-initial-stages-of-cement-hydration-at-the-molecular-level[60].

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

## Acknowledgements

This work was supported in part by National Key Research and Development Program for Young Scientists (No. 2021YFC2900400, C.Q.), Young Elite Scientists Sponsorship Program by CAST (No. 2023QNRC001, C.Q.), Natural Science Foundation of Hunan Province "Efficient and clean utilization of solid waste in the mining and minerals industry" (C.Q.), National Natural Science Foundation of China (Nos. 52004330 and 22376221, C.Q.), Departamento de Educación, Política Lingüística y Cultura del Gobierno Vasco (Grant No. IT1458-22, H.M.), Ministerio de Ciencia e Innovación (TED2021–130860B-I00, H.M.), and Transnational Common Laboratory "Aquitaine-Euskadi Network in Green Concrete and Cement-based Materials" (LTC-Green Concrete, H.M.). This work was also supported in part by the High Performance Computing Center of Central South University (C.Q.) and the Pawsey Supercomputing Center with funding from the Australian Government and the Government of Western Australia (D.S. and C.Q.).

## Author contributions

C.Q. and H.M. conceived and directed the project; C.Q., X.X. and X.M. performed the simulations; X.X. and C.Q. analyzed the simulation results and wrote the manuscript. C.M. and X.X. produced the python script. X.X. performed the visualisation. C.Q., X.X. and H.M. edited the manuscript before submission. D.S. and X.M. discussed the results and commented on the manuscript.

## Competing interests

The authors declare no competing interests.
