## [Peer Review File · Nature Communications]

The initial stages of cement hydration at the molecular levelREVIEWER COMMENTS

Reviewer #1 (Remarks to the Author):

Key results:

The paper describes the dissolution mechanism of Ca at an early stage of hydration, which deals with the atomistic modeling of the (010) surface of m3-C3S and (100) surface of β -C2S for 30 ns.

The initial adsorption of water to clinker interface through dissociation followed by the hopping process is well known in the literature. However, this paper explains the mechanism in detail of the interaction of water molecules and dissolving Ca by following two stages. This paper observes the 6 coordinated dissolving Ca-H₂O complex, which has complemented the previous investigation, however, the author did not consider the energy calculation during the transformation from structure 1 to 3 (Fig. 2b). Ligand teeth were observed for both m3-C3S and β -C2S, however, Ca in m3-C3S tend to form more ligand teeth during dissolution compared to the Ca in β -C2S. In general, the Ca from m3-C3S follows 2 steps dissolution mechanism till 30 ns period, whereas Ca from β -C2S follows one, however, the simulation should run for a longer time to observe the complete dissolution. Finally the radial distribution function and pore distribution mechanism explains the reactivity difference between the clinker phases.

In addition, a complete study is required to compare the reactivity difference between the two clinker phases.

the significance of the work

The scientific level of the work is quite good and relevant to the context of cement hydration. The research work is well-structured and original and deals with an important basis of cement hydration. The overall perspective of the Ca dissolution mechanism from both m3-C3S and β -C2S is quite interesting (observation of ligand teeth) and will be filling the knowledge gap of early cement hydration.

The flaws in the data analysis, and interpretation:

The manuscript is well written, however, the following points are needed to be considered
The complete dissolution of Ca from β -C2S is still missing.

Line 63: extra space after is (this behavior is related to the ionic oxygen)

Line 108: These results confirm that

Figure 1a: The visualization should be better, especially the crystal orientation and atomic rearrangements.

Figure 1d: please use exchange the yellow color for a more visible and suitable color.

Methodology:

The unit cell of m3-C3S and β -C2S are optimized by DFT, which is the comparatively accurate method. The hydration process for both clinker phases is done by ReaxFF. The Reactive Force Field theory (ReaxFF) has been successfully employed to describe the chemical reaction with sufficient accuracy at a reasonable computational cost. It is a unified description of various classes of materials. Therefore, the chosen methods and software/code used for visualization and result analysis are sound and the results can be reproduced.

The paper and supporting material are enough to explain the results

Reviewer #2 (Remarks to the Author):

This study utilized ab-initio and MD simulations to understand the hydration reactions of M3 C3S and beta-C2S. While the overall structure of the paper is well done and the results are thoroughly analyzed, I recommend the following modifications or additions prior to final publication:

1. The paper does not demonstrate a clear differentiation from previous MD simulation studies on C3S and C2S. This should be addressed to highlight the novelty of this study.
2. Please provide a description of the crystal structure parameters used in the initial structure, and include unit cell images of the M3-C3S and beta-C2S structures used in the simulations.
3. While the study's phenomenological results are well-discussed, the significance of the research findings should be emphasized further. Additionally, it is recommended that further discussion is added to demonstrate the relevance of these findings to the actual hydration processes of C3S and C2S, including the formation of hydrates.

Overall, the study has potential to contribute to the field, but these recommended modifications or additions would enhance the clarity and significance of the research findings.

Reviewer #3 (Remarks to the Author):

This manuscript investigated the initial hydration processes of M3-C3S (010) and β -C2S (100) from the dissolution of Ca ions, the solid/water interface and the surface structure. The authors presented many results, especially the novel dissolution pathways of Ca ions. It was found that Ca ions in M3-C3S (010) and β -C2S (100) surface undergo different dissolution processes, which may be due to the contribution

of more O_i in C3S.

The subject may be interesting, however, I wonder no significant new findings with regard to cement hydration have been unraveled, since large numbers of previous publications discussed about this topic. The authors should clarify this, especially in comparison with "Ab initio mechanism revealing for tricalcium silicate dissolution." Nature Communications 13.1 (2022): 1253. Has this work provided more understandings in initial cement hydration? Can the simulation results agree with the experiments? Furthermore, the mechanism of dissolution of Ca ions needs to be explained more concisely, which is the key part of hydration and not adequately discussed. And the following questions should be answered before consideration for publication in Nature Communications.

1. Fig 1: the number of Ca-Ow bond increase slowly before 1.5 ns, more discussions are needed.
2. In "Initial hydration process" section: what does the "Hab" means? Does it mean the H atoms bonded to Os and Ow (bonded to Ca)?
3. Line 105-107: the curves of the number of bonds, both the Ca-Ow bonds and O-Hab bonds, is fluctuating, so the gradients of them are preferred to be adopted rather than the specific increased values.
4. The reason for the differences in the increase rates of Ow-Hab bond in M3-C3S and β -C2S is not clear. How does the Hab bond to Ca in the Ca-Ow-Hab-Ca structure (Line 110-112:)?
5. Line 116 : the word, "bonds", is repetitive.
6. The ligand teeth are responsible for the difference between the Ow-Hab bond increase and twice that of Ca-Ow bonds. For the dissolution of Ca in β -C2S, why does the increase of Ow-Hab bonds is nearly twice (1.9 times, line 107) that of Ca-Ow bonds while "there is a higher proportion of Ca ions with ligand teeth in most dissolution steps" (line 190-191)?
7. Line 185: the "Fig. 2a" is initial Ca ion structure of M3-C3S, should this be Fig. 3a?
8. In "Solid/Water interface and ..." section: more discussions are needed for the five regions of the interface. Details of the atomic density change with time will be more informative.
9. Fig.4: the legend of subgraph a and b should be consistent, otherwise it may confuse the readers.
10. At 0 ns, the range of large pore sizes in β -C2S is larger than that in M3-C3S, and the number of pores in β -C2S is also more than that in M3-C3S (Fig. 5c-d), but why does the "M3-C3S (010) has higher reactivity than β -C2S (100)"(line 272)?

Reviewer #4 (Remarks to the Author):

This paper titled 'The initial stages of cement hydration at the molecular level' looks at the very early age dissolution of C3S and C2S surfaces using molecular dynamics with reactive force field. The paper is well written and well organized. But, I am not entirely convinced at the novelty and the impact of this paper.

ReaxFF has been used previously by the authors to look at C3S dissolution (2015 paper?). In the current paper, from what I understand some of the key additional aspects are the longer simulation (30ns instead of 10ns) and the coordination evolution of calcium ions during dissolution. The same methodology is applied to C2S to get a comparison. However, we do know from some of the DFT simulations (as authors themselves wrote) that the origin of low reactivity or dissolution of C2S is due to the absence of ionic oxygen.. What are we learning new and impactful here?

Now, a major drawback of this paper is the lack of discussion on how their results can be compared to the reality. Would dissolution happen at such perfect flat surfaces? What does the current understanding brings to the community. The complexity of dissolution is not discussed to give the perspective on how these simulations are original or a 'breakthrough'.

Though the paper might look interesting from the dissolution process of Ca ions, isn't it something one can expect during the dissolution process of any calcium based mineral? Would the 2015 paper already have such mechanisms observed in the trajectory? It is my personal view that I don't see if it brings anything significant to the understanding of dissolution of these minerals to the cement community. It is possibly an incremental knowledge that is gained and still limited to 30ns. Another aspect is how well the coordination chemistries of such a dynamic process is realistically captured by the Reaxff. I don't know if you compare this with the recent DFT simulations , reference number 19.

The authors conclude 'this study reproduces the initial hydration processes of ...' How is it 'reproducing'?

The initial stages of cement hydration at the molecular level

Xinhang Xu, Chongchong Qi, Xabier M. Aretxabaleta, Chundi Ma, Dino Spagnoli, Hegoi Manzano

Point-by-point responses to the reviewers

We thank all reviewers for their constructive comments on the article. The paper has improved significantly following their suggestions, especially with respect to a better explanation of the significance of the results, and the discussion about the novelty (compared to previous MD studies). We also used the revision time to extend the simulations from 30 ns to 40 ns, and to do this, we have spent about 2 million core hours over the span of 60 days. Finally, we performed extra free energy analysis along with the structural evolution. The free energy results illustrate the necessity of unbiased simulations like those from this work to understand the dissolution mechanism before applying metadynamics or targeted molecular dynamics to characterize the energy barriers.

We have responded to the reviewers' point-by-point responses in **blue** below, and have indicated which text in the manuscript was revised in **red**.

REVIEWER COMMENTS

Reviewer #1 (Remarks to the Author):

Key results:

The paper describes the dissolution mechanism of Ca at an early stage of hydration. which deals with the atomistic modeling of the (010) surface of m3-C3S and (100) surface of β -C2S for 30 ns.

The initial adsorption of water to clinker interface through dissociation followed by the hopping process is well known in the literature. However, This paper explains the mechanism in detail of the interaction of water molecules and dissolving Ca by following two stages. This paper observes the 6 coordinated dissolving Ca-H2O complex, which has complemented the previous investigation, however, Author did not consider the energy calculation during the transformation from structure 1 to 3 (Fig. 2b).

The study of the energetics during the transformation from Structure 1 to 3 (Fig. 2b) is indeed

a very interesting thing. We had explored the energy evolution (energy barrier) in the original research, but we found it extremely difficult to characterize. For this revision due to the comments from all the reviewers, we have tried to frame better the transition and the energy calculation could be one of the main points of the paper: **performing these energy calculations for this material is far from trivial, and the usual methods in the literature present important flaws.**

Dissolution events are usually studied as “rare events” that barely happen in the simulation timescale (ns) devising a collective variable and adding biasing forces or energies to it. The system is forced to evolve and the rare event takes place in an accessible time. Then, the free energy along that coordinate (actually, a potential of mean force) can be reconstructed. In these methods, it is common to assume that the only “event” in the system is the one tracked, or in other words, the evolution of the system is well represented by the change of the collective variable. In contrast, in this work, we are following the system evolution without assuming a specific reaction coordinate, without any “bias”. And it is necessary, because when we tried to apply the “biased” methods, we find that the reaction coordinate cannot be defined independently from the rest of the system.

To illustrate this, we have now done an energy analysis of the transformation from Structure 2 to 3 in C_3S dissolution, which is considered to require a higher energy barrier than the transformation from Structure 1 to 2. Previous studies, as the **Ab initio mechanism revealing for tricalcium silicate dissolution (Nature Communications 13.1 (2022): 1253)**¹⁴ use DFT and metadynamics to explore the Ca dissolution from the surface. The selected collective variable is the Ca-O coordination to the surface: water, which changes from 3:3 to 0:6, while the rest of the system remains “constant”. In our unbiased simulations, we observe that the desorption is not independent of the environment. **Ca remains with a given coordination and hops only when the surface oxygen is hydroxylated.** From the unbiased MD trajectory, we have performed umbrella sampling (US) simulations along the original transition path in two scenarios. First without any constraint on the system, which represents the MD simulations done in this work. Second, constraining the H hopping to prevent O hydroxylation, which mimics the scenario in common DFT simulations or unreactive force field studies.

The US simulations were done with LAMMPS and the Colvars module. The reaction coordinate has been chosen as the main distance from the dissolving Ca atom (id = 360) to the surrounding Ca atoms (ids = 7792, 220, 2904, 260, and 20). The calculations were done at room temperature using the Noose-Hoover thermostat in the NVT ensemble. The time step was set to 0.1 fs and the total simulation time was 100 ps. For each path, 12 simulations were done where the reaction coordinate was taken from 5.15 Å to 5.7 Å in steps of 0.05 Å. For the "H-jump" scenario the spring constant for the umbrella simulations was set to 50 kcal/(mol·Å) and for the "no-H-jump" was set to 200 kcal/(mol·Å). In both scenarios, we made sure that all the path was correctly sampled. To recover the free energy from the different simulations we used WHAM¹⁵, with 100 bins.

Clearly, the free energy is considerably different if H hopping is allowed or not. When H

hopping is allowed, a dissociated water molecule creates a hydroxyl group in a surface oxygen coordinated to Ca when Ca is at ~ 5.25 Å. That makes the Ca-O(H) bond weaker and therefore facilitates the movement of the Ca, which reaches a stable state at ~ 5.55 Å. The energy barrier is less than 1 kcal mol^{-1} (4 kJ mol^{-1}). When the water molecule is not allowed to dissociate, the Ca-O bond remains strong, and increasing the distance for the Ca desorption is unfavorable. The free energy increases constantly up to 4 kcal/mol (16 kJ/mol), and no final stable state is found.

This **key result** illustrates the necessity of unbiased simulations like those from this work to understand the dissolution mechanism before applying metadynamics or targeted molecular dynamics to characterize the energy barriers. Regarding the specific value for this Ca hop, $\sim 16 \text{ kJ/mol}$, it is lower than the experimental activation energy of C_3S (30 to 50 kJ/mol depending on the source), so we can conclude that it is not the limiting step in C_3S dissolution. More complex studies would be necessary to determine it.

Result section:

“From the unbiased trajectory, we perform umbrella sampling (US) simulations along the original transition path in two scenarios (Fig. 2c). First without any constraint on the system, which represents the MD simulations done in this study (the H-jump scenario). Second, constraining the H hopping to prevent O hydroxylation, which mimics the scenario in common DFT simulations or unreactive force field studies (the no H-jump scenario). When H hopping is allowed, a dissociated water molecule creates a hydroxyl group in a surface oxygen coordinated to Ca ion when Ca ion is at ~ 5.25 Å. That makes the Ca-O(H) bond weaker and therefore facilitates the movement of the Ca ion, which reaches a stable state at ~ 5.55 Å. The energy barrier is less than 1 kcal mol^{-1} (4 kJ mol^{-1}). When the water molecule is not allowed to dissociate, the Ca-O bond remains strong, and increasing the distance for the Ca ion desorption is unfavorable. The potential of mean force increases constantly up to 4 kcal mol^{-1} (16 kJ mol^{-1}), and no final stable state is found. This illustrates the necessity of unbiased simulations to understand the dissolution mechanism before applying metadynamics or targeted molecular dynamics to characterize the energy barriers.”

Fig. 2 The second dissolution process of Ca ions from M3-C₃S (010). **a** The initial Ca ion structure of the second dissolution process, namely Structure 1. **b** The conceptualized flowchart illustrating the second dissolution process. **c** The potential of mean force of the transition from Structure 2 to 3. The transformation from Structure 2 to 3 might require a higher energy barrier than that from Structure 1 to 2 since it takes a much longer transformation time. **d** The evolution of Ca ions' structures with time. The structures of Ca ions are presented in parentheses as the number of Ca-O_s bonds, the number of ligand teeth, and the number of Ca-H₂O_w bonds. The surface is shown as the shaded box. The raw data curves in Fig. 2d are partially transparent and the smoothed curves are highlighted.

Discussion section:

“In other words, the dissolution of Ca ions is not a simple process of bonding a water molecule and breaking a Ca-O_s bond, and it is not independent of the environment. Ca ions remain with a given coordination and hop only when the surface oxygen is hydroxylated. Ca ions form ligand

teeth structures with O_w ions, which facilitate the detachment of Ca ions from O_s ions. The umbrella sampling results illustrate that unbiased simulations are necessary to understand the dissolution mechanism before applying metadynamics or targeted molecular dynamics to characterize the energy barriers.”

Ligand teeth were observed for both $m3-C_3S$ and $\beta-C_2S$, however, Ca in $m3-C_3S$ tend to form more ligand teeth during dissolution compared to the Ca in $\beta-C_2S$. In general, the Ca from $m3-C_3S$ follows 2 steps dissolution mechanism till 30 ns period, whereas Ca from $\beta-C_2S$ follows one, however, the simulation should run for a longer time to observe the complete dissolution.

During the revision, we extended the simulation time from 30 ns to 40 ns and updated the data in the revised manuscript. It must be taken into account that ReaxFF simulations are very time-consuming compared to other classical force fields, yet it is necessary to use it due to the water dissociation reactions. The extension of MD simulation lasted around 60 days with about 2 million core hours being used. Running longer simulation times is simply unpractical, and new methodologies like reactive ML potentials will be needed to go further.

Regarding the lack of Ca dissolution on belite, it just shows that the dissolution is slower, as expected from real time dissolution experiments.

Finally the radial distribution function and pore distribution mechanism explains the reactivity difference between the clinker phases.

In addition, a complete study is required to compare the reactivity difference between the two clinker phases.

We have presented a detailed comparison of the reactivity difference between the two clinkers in the discussion section of the revised manuscript. Next, we will first answer your questions and place the revised part of the manuscript after the response section.

Firstly, in the first paragraph of the discussion, we summarized the results of the reactivity difference between the two clinkers obtained in this study. These results corroborate that $M3-C_3S$ (010) has higher reactivity than $\beta-C_2S$ (100) during the initial hydration process. The dissolution of Ca ions undergoes two processes in $M3-C_3S$, while in $\beta-C_2S$ it remains at the first process.

Then, in the second paragraph of the discussion, we show that the reactivity difference is clearly intrinsic from the crystalline composition. As previously suggested in the literature, we demonstrate that the reasons for the higher reactivity of $M3-C_3S$ than $\beta-C_2S$ would be the O_i ions in the unit cell of $M3-C_3S$, which has the lowest number of valence electrons¹⁶. The

Ca ions have a higher valence electron number in M3-C₃S than in β-C₂S due to bonding with the O_i ions. The higher reactivity of Ca and O_i ions in M3-C₃S will facilitate the hydration reaction. Specifically, in water molecule adsorption, the adsorption energies were more negative on M3-C₃S (010) than on β-C₂S (100), indicating that M3-C₃S (010) is more energetically-favorable for water molecule adsorption¹⁷. Meanwhile, these O_i ions provide more favorable adsorption sites for H ions than O_{si} ions, which benefits proton transfers and the formation of ligand teeth in M3-C₃S (010). Moreover, the results of a faster hydration reaction of C₃S than C₂S are in line with the early hydration experiments. The undersaturated dissolution rate of anhydrous Monoclinic C₃S in deionized water at 20 °C was -74.00 μmol m⁻² s⁻¹ while that of anhydrous β-C₂S was -16.78 μmol m⁻² s⁻¹¹⁸.

Discussion section:

“During the initial hydration within 40 ns, dozens of Ca ions in M3-C₃S (010) are completely detached from O_s ions and form a stable six-coordinated structure with water. The Ca ions in M3-C₃S (010) start to undergo the second dissolution process at 0.0017 ns. However, the Ca ions in β-C₂S (100) remain within the first dissolution process until 40 ns, and no stable detachment from O_s ions is observed. At 40 ns, the stern layer and small pores appear in M3-C₃S (010); these features are not observed in β-C₂S (100). These results corroborate that M3-C₃S (010) has higher reactivity than β-C₂S (100) even during the initial hydration process, and the difference originates from the crystal structure. The subsequent hydration of M3-C₃S might be further promoted by the appearance of the pore solution saturation. Furthermore, ligand teeth structures are observed during the Ca ions dissolution for both M3-C₃S (010) and β-C₂S (100). In other words, the dissolution of Ca ions is not a simple process of bonding a water molecule and breaking a Ca-O_s bond, and it is not independent of the environment. Ca ions remain with a given coordination and hop only when the surface oxygen is hydroxylated. Ca ions form ligand teeth structures with O_w ions, which facilitate the detachment of Ca ions from O_s ions. The umbrella sampling results illustrate that unbiased simulations are necessary to understand the dissolution mechanism before applying metadynamics or targeted molecular dynamics to characterize the energy barriers.

The reason for the higher reactivity of M3-C₃S (010) than β-C₂S (100) is due to the contribution of the O_i ions in M3-C₃S. The O ions in the unit cell of M3-C₃S can be divided into four groups based on the Bader charge, with the first group (O_i ions) having the lowest number of valence electrons¹⁶. The Ca ions have an average higher valence electron number in M3-C₃S than in β-C₂S due to bonding with the O_i ions. The higher reactivity of Ca and O ions in M3-C₃S will facilitate the hydration reaction. Specifically, in water molecule adsorption, the adsorption energies are more negative on M3-C₃S (010) than on β-C₂S (100)¹⁷, indicating that the water adsorption on M3-C₃S (010) is more energetically favorable. Meanwhile, these O_i ions provide more favorable adsorption sites for H ions than O_{si} ions, which thus benefits proton transfers and the formation of ligand teeth in M3-C₃S (010). Regarding the hydration rate, the faster hydration reaction of C₃S than C₂S also agrees well with the experimental data. Specifically, the experimental undersaturated dissolution rate of anhydrous monoclinic C₃S in deionized water at 20 °C is determined to be -74.00 μmol m⁻² s⁻¹ while that of anhydrous β-C₂S is -16.78 μmol m⁻² s⁻¹¹⁸.”

the significance of the work

The scientific level of the work is quite good and relevant to the context of cement hydration. The research work is well-structured and original and deals with an important basis of cement hydration. The overall, perspective of the Ca dissolution mechanism from both $m3-C3S$ and $\beta-C2S$ is quite interesting (observation of ligand teeth) and will be filling the knowledge gap of early cement hydration.

We are glad for the positive comments on the significance of the work. In the revised manuscript, we have further strengthened the significance of our work by providing a more detailed explanation of the novelty and key contributions, extending the MD simulation time, and adding a representative energy analysis.

The flaws in the data analysis, and interpretation:

The manuscript is well written, however, the following points are needed to be considered

The complete dissolution of Ca from $\beta-C2S$ is still missing.

We extended the simulation time from 30 ns to 40 ns and updated the data in the revised manuscript. The extension of MD simulation lasted around 60 days with about 2 million core hours being used. We observed that more Ca ions entered the dissolution Step 3 at 40 ns in $\beta-C_2S$. However, we did not observe a stable detachment of Ca ions from the surface ions in $\beta-C_2S$ even at 40 ns. The authors believe that even if the stable dissolution of Ca ions in C_2S is not observed, the 40 ns hydration simulation provides a significant advance for the Ca dissolution process of cement hydration. The authors noted that achieving stable dissolution of Ca ions in $\beta-C_2S$ still requires enormous computational resources and time, which might be investigated from a different technique. Machine learning potentials for instance are an alternative faster than ReaxFF that we are exploring.

Line 63: extra space after is (this behavior is related to the ionic oxygen)

Line 108: These results confirm that

Thanks for your careful review. We have removed the extra space and we have corrected the typo. We have rechecked the manuscript once again to avoid further typos.

Figure 1a: The visualization should be better, especially the crystal orientation and atomic rearrangements.

Thanks for your valuable suggestions. We improved the visualization of Fig. 1a. The crystal structure parameters used in the initial structure are provided in the Supplementary information (Supplementary Table 4).

“The detailed MD model parameters were presented in the Supplementary Information (Supplementary Table 4).”

Moreover, we have provided visualization of unit cells and low-index surfaces of M3-C₃S and β-C₂S to facilitate the reproduction of the surface models (Supplementary Fig. 12 and Fig. 13).

“The optimized unit cell and the cleaved surfaces were presented in the Supplementary information (Supplementary Fig. 12 and Fig. 13).”

Fig. 1 Initial hydration process of M3-C₃S (010) and β-C₂S (100). a The near-surface ranges of the slab model before MD simulations.

Supplementary Table 4. The MD model parameters of β-C₂S (100) and M3-C₃S (010).

Type	a (Å)	b (Å)	c (Å)*	α (°)	β (°)	γ (°)
β-C ₂ S (100)	27.03	37.21	259.93	90	90	90
M3-C ₃ S (010)	34.60	27.84	292.05	90	90	70.54

* The difference between β-C₂S (100) and M3-C₃S (010) thicknesses was due to the selection of the same period at the surfaces and the preservation of the periodicity of the in-plane crystals.

Supplementary Figure 12. The optimized unit cell structures of M3-C₃S and β -C₂S.

Supplementary Figure 13. The cleaved surfaces of M3-C₃S (010) and β -C₂S (100).

Figure 1d: please use exchange the yellow color for a more visible and suitable color.

Thanks for your valuable suggestions. We have used green color instead of yellow color in the revised manuscript, which can be more clearly observed.

Fig. 1 Initial hydration process of M3-C₃S (010) and β-C₂S (100). d The number of O-H bonds.

Methodology:

The unit cell of m3-C3S and β-C2S are optimized by DFT, which is the comparatively accurate method.

The hydration process for both clinker phases is done by ReaxFF.

The Reactive Force Field theory (ReaxFF) has been successfully employed to describe the chemical reaction with sufficient accuracy at a reasonable computational cost. It is a unified description of various classes of materials. Therefore, the chosen methods and software/code used for visualization and result analysis are sound and the results can be reproduced.

The paper and supporting material are enough to explain the results.

We greatly appreciate the positive comments and we have carefully revised all comments and suggestions. Thanks to the reviewer for the efforts in reviewing our research.

Reviewer #2 (Remarks to the Author):

This study utilized ab-initio and MD simulations to understand the hydration reactions of M3 C3S and beta-C2S. While the overall structure of the paper is well done and the results are thoroughly analyzed, I recommend the following modifications or additions prior to final publication:

1. The paper does not demonstrate a clear differentiation from previous MD simulation studies on C3S and C2S. This should be addressed to highlight the novelty of this study.

Sincerely thank you for your suggestions. We have clarified the differences from other MD simulations in the revised manuscript. Next, we will first answer your questions and place the revised part of the manuscript after the response section.

The significant difference is the simulation model and time. We have achieved a simulation time of 40 ns, which is the longest simulation time for the initial hydration of cement and even hydration of other calcium-based minerals achieved by ReaxFF. The simulation times in previous MD studies were usually between a few tens of ps and a few ns^{11, 19}. The extension of simulation time allows us to discover the dissolution mechanism of Ca ions, which cannot be achieved by previous MD studies. Meanwhile, we chose a very short time step of 0.1 fs, compared to the usual 0.2-0.5 fs in previous MD studies^{19, 20}, which ensures more accurate and realistic simulations. Our simulation is more realistic in that we used a large model (~23000 in this study compared with <2000 in the literature¹⁹) together with the ReaxFF, which can correctly describe chemical reactions, allow for dynamic bonds formation and breaking, and accurately describe atomic motion in complex chemical environments^{1, 4}. In our study, we did not use any accelerated dissolution method, i.e. adding biasing forces or energies to the system. This means that a large number of chemical reactions have occurred on the surface before Ca ions dissolution and amorphous products are continuously formed on the surface, which is consistent with the real Ca ions dissolution process²¹. Moreover, a comparison between M3-C₃S and β-C₂S has been presented in the current study, in contrast to previous literature which analyses one phase or the other¹¹. A detailed comparison of this work with two representative papers is provided in Supplementary Information (Supplementary Note 3).

Using the unbiased method **we have characterized the step-by-step dissolution process** (Supplementary Note 4). In particular, for M3-C₃S in previous studies, it was considered that Ca ions dissolve directly after breaking all bonds with surface O ions, which is the dissolution of Ca ions by definition. However, our results show that the complete detachment of Ca ions from surface O ions in M3-C₃S does not achieve free movement in the water layer, but enters a new dissolution process (the second dissolution process). The second dissolution process requires a tens of thousands fold longer period than the first dissolution process, and no Ca ions have went through the second dissolution process until 40 ns. The discovery of this new

process revises our current understanding of Ca ions dissolution in M3-C₃S and provides a benchmark knowledge for the dissolution mechanism of other calcium-based minerals. **We found a new ligand teeth structure**, which is specifically expressed as Ca-O_w-H_{ab}-Ca (Fig. 2a). The ligand teeth widely exist in the Ca ions dissolution of M3-C₃S and β-C₂S, the appearance of which facilitates the detachment of Ca ions from the surface O ions. The formation of more ligand teeth structures is observed during the Ca ions dissolution of M3-C₃S. Moreover, multiple ligand teeth structures are also observed in the intermediate layer of C-S-H structure¹⁰, and such ligand teeth structures will play an important role in hydrates formation. Finally, we characterized the initial hydration process of the cement at 40 ns in multiple ways. These characterizations also led to many new findings that are not previously concluded. For example, **the appearance of aqueous layers and the small pores that favor H transfer in M3-C₃S**. These characterization results demonstrate in great detail the differences in the early hydration of M3-C₃S and β-C₂S. Moreover, this study provides huge simulation data for the initial cement hydration process, which paves the way for further in-depth as well as multi-faceted studies of cement hydration in the future.

We have also clearly indicated the necessity of our unbiased simulation method. Due to the access to larger times, we can investigate the process without using any biased method like metadynamics, which let us **understand correctly the Ca desorption mechanism**. Our new energy calculations illustrate perfectly the key difference. Previous studies, as the **Ab initio mechanism revealing for tricalcium silicate dissolution (Nature Communications 13.1 (2022): 1253)**¹⁴ which recently raised attention in the community, use DFT and metadynamics to explore the Ca dissolution from the surface. The selected collective variable is the Ca-O coordination to surface:water, which changes from 3:3 to 0:6, while the rest of the system remains “constant”, i.e. no chemical reaction takes place, just change in the Ca coordination number. In our unbiased simulations, we observe that the desorption is not independent of the environment. Ca remains with a given coordination and hops only when the surface oxygen is hydroxylated. From the unbiased MD trajectory, we have performed umbrella sampling (US) simulations along the original transition path in two scenarios. First without any constraint on the system, which represents the MD simulations done in this work. Second, constraining the H hopping to prevent O hydroxylation, which mimics the scenario in common DFT simulations or unreactive force field studies.

The US simulations were done with LAMMPS and the Colvars module. The reaction coordinate has been chosen as the main distance from the dissolving Ca atom (id = 360) to the surrounding Ca atoms (ids = 7792, 220, 2904, 260, and 20). The calculations were done at room temperature using the Noose-Hoover thermostat in the NVT ensemble. The time step was set to 0.1 fs and the total simulation time was 100 ps. For each scenario, 12 simulations were done where the reaction coordinate was taken from 5.15 Å to 5.7 Å in steps of 0.05 Å. For the "H-jump" scenario the spring constant for the umbrella simulations was set to 50 kcal/(mol·Å) and for the "no-H-jump" was set to 200 kcal/(mol·Å). In both scenarios, we made sure that all the path was correctly sampled. To recover the free energy from the different simulations we used WHAM¹⁵, with 100 bins.

Clearly, the free energy is considerably different if H hopping is allowed or not. When H

hopping is allowed, a dissociated water molecule creates a hydroxyl group in a surface oxygen coordinated to Ca when Ca is at ~ 5.25 Å. That makes the Ca-O(H) bond weaker and therefore facilitates the movement of the Ca, which reaches a stable state at ~ 5.55 Å. The energy barrier is less than 1 kcal mol^{-1} (4 kJ mol^{-1}). When the water molecule is not allowed to dissociate, the Ca-O bond remains strong, and increasing the distance for the Ca desorption is unfavorable. The free energy increases constantly up to 4 kcal/mol (16 kJ/mol), and no final stable state was found. **This key result illustrates the necessity of unbiased simulations like those from this work to understand the dissolution mechanism** before applying metadynamics or targeted molecular dynamics to characterize the energy barriers.

Discussion section:

“Moreover, it should be noted that there are significant differences in the MD simulation in this study compared to previous ones about cement hydration. Firstly, a long simulation time of 40 ns is performed in this study compared to previous MD studies (ranging from tens of ps to a few ns)^{11, 19}. The extension of the simulation time allows Ca ions to dissolve in an unbiased way, which is not possible in previous MD studies. Secondly, a very short time step of 0.1 fs is chosen in this study, compared to 0.2-0.5 fs in many previous MD studies^{19, 20}, which ensures a more accurate and realistic simulation. Third, the simulations in this study are more realistic. The simulation employs a large slab model (~ 23000 in this study compared with <2000 in the literature⁵) together with the ReaxFF reactive force field, which allows for dynamic bonds formation and breaking, and accurately describes atomic motion in complex chemical environments^{1, 4}. Specifically, a large number of chemical reactions have occurred at the surfaces before Ca ions dissolution, and amorphous products are continuously formed at the surfaces, which is consistent with the real Ca ions dissolution process²¹. Fourth, we do not use any accelerated dissolution method, i.e. adding biasing forces or energies to the system. This means that a large number of chemical reactions have occurred on the surface before Ca ions dissolution and amorphous products are continuously formed on the surface, which is consistent with the real Ca ions dissolution process²¹. A detailed comparison of this work with two representative papers is provided in Supplementary Information (Supplementary Note 3).

In conclusion, using the unbiased method, this study makes four main contributions to the initial cement hydration, namely a new Ca dissolution process, two new general dissolution pathways, a key structure for Ca ions dissolution, and a detailed characterization of the hydration process (Supplementary Note 4). Briefly, we find that Ca ions on the M3-C₃S (010) surface must undergo two dissolution processes, whereas Ca ions in β -C₂S (100) mainly remain in the first dissolution process during the studied 40-ns period. The general dissolution pathways of Ca ions in C₃S and C₂S are resolved in this work. A new ligand teeth structure is discovered in the dissolution of Ca ions in C₃S and C₂S. In addition, the solid/water arrangement at the interface and the surface structure after the initial hydration process, such as the appearance of aqueous layers and the small pores favoring H transfer in M3-C₃S, are characterized. Moreover, the free energy calculation illustrates that the hydroxylation state of the neighboring atoms is a key factor which enables Ca ions desorption, implying the necessity of unbiased MD simulations. Due to the fast reactivity of

dissolution at the initial stage of cement hydration, there are obvious technical limitations to solve the dissolution mechanism at the molecular level through laboratory experiments²². The detailed characterization of the cement/water interface in this study will provide an important reference for revealing the dissolution mechanism of initial cement hydration and clarifying the reaction pathway for Ca ions dissolution. Overall, these results fill a gap in the current understanding of the dissolution process of Ca ions and open a new chapter in the study of the initial hydration process of cement.”

Introduction section:

“In this study, we use ReaxFF reactive force field simulations to reveal the initial hydration processes of C₃S and C₂S from multiple perspectives, including chemical bonding, dissolution of Ca ions, and surface structural evolution. Here, we present the breakthrough discovery of a new dissolution process of Ca ions on the C₃S surface and reveal, for the first time, the general dissolution pathways of Ca ions from C₃S and C₂S surfaces. The unbiased molecular dynamics (MD) simulation, as validated by the free energy calculation, reveals that the hydroxylation state of the neighboring atoms is a key factor which enables Ca desorption. A new ligand teeth structure is discovered during the dissolution of Ca ions, which is critical for its dissolution process. In addition, the solid/water arrangement near the interface is characterized after the initial hydration process, including the appearance of aqueous layers and the small pores favoring H transfer in M3-C₃S. Overall, our discoveries fill a gap in the understanding of the dissolution process of Ca ions as well as represent a new milestone in the study of the initial hydration processes of C₃S and C₂S.”

Result section:

“From the unbiased trajectory, we perform umbrella sampling (US) simulations along the original transition path in two scenarios (Fig. 2c). First without any constraint on the system, which represents the MD simulations done in this study (the H-jump scenario). Second, constraining the H hopping to prevent O hydroxylation, which mimics the scenario in common DFT simulations or unreactive force field studies (the no H-jump scenario). When H hopping is allowed, a dissociated water molecule creates a hydroxyl group in a surface oxygen coordinated to Ca ion when Ca ion is at ~ 5.25 Å. That makes the Ca-O(H) bond weaker and therefore facilitates the movement of the Ca ion, which reaches a stable state at ~ 5.55 Å. The energy barrier is less than 1 kcal mol⁻¹ (4 kJ mol⁻¹). When the water molecule is not allowed to dissociate, the Ca-O bond remains strong, and increasing the distance for the Ca ion desorption is unfavorable. The potential of mean force increases constantly up to 4 kcal mol⁻¹ (16 kJ mol⁻¹), and no final stable state is found. This illustrates the necessity of unbiased simulations to understand the dissolution mechanism before applying metadynamics or targeted molecular dynamics to characterize the energy barriers.”

Fig. 2 The second dissolution process of Ca ions from M3-C₃S (010). **a** The initial Ca ion structure of the second dissolution process, namely Structure 1. **b** The conceptualized flowchart illustrating the second dissolution process. **c** The potential of mean force of the transition from Structure 2 to 3. The transformation from Structure 2 to 3 might require a higher energy barrier than that from Structure 1 to 2 since it takes a much longer transformation time. **d** The evolution of Ca ions' structures with time. The structures of Ca ions are presented in parentheses as the number of Ca-O_s bonds, the number of ligand teeth, and the number of Ca-H₂O_w bonds. The surface is shown as the shaded box. The raw data curves in Fig. 2d are partially transparent and the smoothed curves are highlighted.

Discussion section:

“In other words, the dissolution of Ca ions is not a simple process of bonding a water molecule and breaking a Ca-O_s bond, and it is not independent of the environment. Ca ions remain with a given coordination and hop only when the surface oxygen is hydroxylated. Ca ions form ligand

teeth structures with O_w ions, which facilitate the detachment of Ca ions from O_s ions. The umbrella sampling results illustrate that unbiased simulations are necessary to understand the dissolution mechanism before applying metadynamics or targeted molecular dynamics to characterize the energy barriers.”

Supplementary Note section:

Supplementary Note 3

“The first comparison is made with the Nature Communications 13.1 (2022): 1253, in which the Ca dissolution process of M3-C₃S was initially discussed. However, the model in the literature contained less than 1000 atoms. Meanwhile, the simulation time remained on the scale of ps, approximately at 0.01 ns. The metadynamics approach was used in the literature to accelerate the structural evolution and pull the target Ca ions with special initial coordination out of a perfectly ordered crystal. Therefore, the simulation procedure in the literature was biased and not fully realistic, considering that the Ca dissolution only happened at a very limited and specific region. In this study, we do not use any accelerated dissolution method. This means that a large number of chemical reactions have occurred on the surface before Ca ions dissolution and amorphous products are continuously formed on the surface, which is consistent with the real Ca ions dissolution process. Therefore, we have a completely different configuration compared to this literature, which is believed to be more realistic. This will provide a more realistic interface and reaction path for cement hydration, which is essential for the comprehensive analysis of the cement hydration mechanism.

The second comparison is made with the study written by the authors in 2015, ACS Appl. Mater. Interfaces 2015, 7, 27, 14726-14733. The ACS study used the molecular dynamics of reaction force fields to understand the hydration of tricalcium silicate. First, static characteristics such as surface energy and water adsorption energy were analyzed. This led to the conclusion that a dynamic study of mineral hydration is necessary to observe local chemical reactions and reveal the hydration process. However, limited by the time scale (2 ns), it remained at the stage of water molecule adsorption, including the formation of hydroxyl pairs and the hopping of H ions to the interior, and no dissolution of Ca ions was observed. Therefore, the ACS paper could not shed light on the Ca dissolution, which is the major contribution of this study. Moreover, the simulation model is larger and a detailed comparison between M3-C₃S and β -C₂S has been presented in the current study.”

Supplementary Note 4

“This study provides four main contributions to the initial cement hydration, namely a new Ca dissolution process, two new general dissolution pathways, a key structure for Ca ions dissolution, and a detailed characterization of the hydration process.

A new Ca dissolution process: For M3-C₃S in previous studies, it was considered that Ca ions dissolve directly after breaking all bonds with surface O ions, which is the dissolution of Ca ions by definition. However, the results show that the complete detachment of Ca ions from surface O ions in M3-C₃S does not achieve free movement in the water layer, but enters a new dissolution

process, namely the second dissolution process. The second dissolution process requires a tens of thousands fold longer period than the first dissolution process. The discovery of this new process revises our current understanding of Ca ions dissolution in M3-C₃S and provides a benchmark knowledge for the dissolution mechanism of other silicate minerals.

Two new general dissolution pathways: In this study, the general pathways of Ca ions dissolution in M3-C₃S and β-C₂S have clarified after extensive statistics analysis, which is a breakthrough in Ca dissolution. The authors consider the revealed Ca ions dissolution pathways to be more realistic and representative since they are obtained from non-speculative, highly-accurate, and non-accelerated structural evolution MD simulations. That marks the opening of a new chapter in the study of initial cement hydration, from the stage of water molecules adsorption to the stage of Ca dissolution. The general Ca dissolution pathway will be an important theoretical basis for the Ca dissolution stage and a key step toward exploring the nucleation of cement hydration.

A key structure for Ca dissolution: In this study, a new ligand teeth structure is observed, which is specifically expressed as Ca-O_w-H_{ab}-Ca. The ligand teeth widely exist in the Ca ions dissolution of M3-C₃S and β-C₂S, the appearance of which facilitates the detachment of Ca ions from the surface O ions. The formation of more ligand teeth structures is observed during the Ca ions dissolution of M3-C₃S. Moreover, multiple ligand teeth structures are also observed in the intermediate layer of C-S-H structure, and such ligand teeth structures will play an important role in hydrates formation.

A detailed characterization of the hydration process: The initial hydration process of the cement at 40 ns is characterized in multiple ways. These characterizations also led to many new findings that are not previously found, for example, the appearance of aqueous layers and the small pores favoring H transfer in M3-C₃S. These characterization results demonstrate in great detail the differences in the early hydration of M3-C₃S and β-C₂S. Moreover, this study provides a huge simulation data for the initial cement hydration process, which paves the way for further in-depth as well as multi-faceted studies of cement hydration in the future.”

2. Please provide a description of the crystal structure parameters used in the initial structure, and include unit cell images of the M3-C₃S and beta-C₂S structures used in the simulations.

Thanks for your valuable suggestions. We have provided the optimized unit cell structures and MD model parameters of M3-C₃S and β-C₂S in the Supplementary information (Supplementary Fig. 12 and Table 4). Moreover, we have provided surface structures before optimization to facilitate the reproduction of the surface models (Supplementary Fig. 13).

“The detailed MD model parameters were presented in the Supplementary Information (Supplementary Table 4).”

“The optimized unit cell and the cleaved surfaces were presented in the Supplementary

information (Supplementary Fig. 12 and Fig. 13).”

Supplementary Figure 12. The optimized unit cell structures of M3-C₃S and β -C₂S.

Supplementary Table 4. The MD model parameters of β -C₂S (100) and M3-C₃S (010).

Type	a (Å)	b (Å)	c (Å)*	α (°)	β (°)	γ (°)
β -C ₂ S (100)	27.03	37.21	259.93	90	90	90
M3-C ₃ S (010)	34.60	27.84	292.05	90	90	70.54

* The difference between β -C₂S (100) and M3-C₃S (010) thicknesses was due to the selection of the same period at the surfaces and the preservation of the periodicity of the in-plane crystals.

Supplementary Figure 13. The cleaved surfaces of M3-C₃S (010) and β -C₂S (100).

3. While the study's phenomenological results are well-discussed, the significance of the research findings should be emphasized further. Additionally, it is recommended that further discussion is added to demonstrate the relevance of these findings to the actual hydration processes of C₃S and C₂S, including the formation of hydrates.

Sincerely thank you for your suggestions. We have further discussed the relevance of the results and experiments and enhanced the significance of the findings by revising the manuscript in four points. Next, we will first answer your questions and place the revised part of the manuscript after the response section.

In the first point, we compare in detail the differences between our study and the previous MD study and summarize our new findings, which are explained in detail in Question one. This helps the readers to understand how this study is original and contributes significantly to the cement hydration mechanism. Notably, the free energy discussion added in the revised manuscript clearly illustrates the necessity of unbiased simulations, the difference between this work and previous MD studies, and the contribution of the novel findings to the community.

In the second point, to further enhance the significance of the results, we present a detailed comparison between C₃S and C₂S in the discussion section. In the first paragraph of the discussion, we summarize the results of the reactivity differences between the two clinkers in this study. These results confirm that M3-C₃S (010) has higher reactivity than β-C₂S (100) during the initial hydration process. The higher reactivity of C₃S originates mainly from its crystal structure, which is further promoted by the pore solution saturation along with the initial hydration process. Then, in the second paragraph of the discussion, we indicated that the reason for the higher reactivity of M3-C₃S than β-C₂S would be the O_i ions in the unit cell of M3-C₃S, which has the lowest number of valence electrons¹⁶. The Ca ions have a higher valence electron number in M3-C₃S than in β-C₂S due to bonding with the O_i ions. The higher reactivity of Ca and O_i ions in M3-C₃S will facilitate the hydration reaction. Specifically, in water molecule adsorption, the adsorption energies were more negative on M3-C₃S (010) than on β-C₂S (100), indicating that M3-C₃S (010) is more energetically favorable for water molecule adsorption¹⁷. Meanwhile, these O_i ions provide more favorable adsorption sites for H ions than O_{si} ions, which thus benefits proton transfers and the formation of ligand teeth in M3-C₃S (010).

In the third point, we add more discussion about the relevance of the simulation results from this study to the actual hydration process. First, our simulation process is consistent with experimental observations. For example, Pustovgar et al.²¹ analyzed the hydration of C₃S by nuclear magnetic resonance spectroscopy. The experiment revealed the formation of hydroxylated Q⁰ silicate species on the particle surface before contact with large amounts of water. In the simulations, the hydroxylated species rapidly appeared at the instant of contact of the surface with water (1E-4 ns). Meanwhile, experiments showed that only the near

surface of Ca_3SiO_5 particles were hydroxylated at low amounts of polymeric silicate hydration products²¹, which is consistent with the simulated process (Fig. 4d). Moreover, we compare the simulated data with the experimental data. The undersaturated dissolution rate of anhydrous Monoclinic C_3S in deionized water at 20 °C was $-74.00 \mu\text{mol m}^{-2} \text{s}^{-1}$ while that of anhydrous $\beta\text{-C}_2\text{S}$ was $-16.78 \mu\text{mol m}^{-2} \text{s}^{-1}$ ¹⁸, implying a qualitative agreement between conclusions obtained from MD simulations and laboratory experiments.

In the fourth point, we explain the necessity of MD simulations by referring to experimental limitations. Laboratory experiments have certain limitations in investigating the initial cement hydration due to the intrinsic impurity within cementitious materials, the rapid hydration processes (i.e., dissolution and precipitation), and the investigation scales²². It is widely-accepted that the lack of proper description and characterization of the cement/water interface might be the main reason why early cement hydration is still poorly understood^{22, 23}. This study fills the knowledge gap in the early Ca dissolution stage by a detailed characterization of the hydration process of C_3S and C_2S within 40 ns. The authors believe that this will provide an important reference for the development of the dissolution mechanism of initial cement hydration and clarify the reaction path for initial cement hydration studies.

In summary, the above four points clarify the necessity, originality, and breakthrough of this study, and its consistency with early hydration processes.

Discussion section:

“During the initial hydration within 40 ns, dozens of Ca ions in M3- C_3S (010) are completely detached from O_s ions and form a stable six-coordinated structure with water. The Ca ions in M3- C_3S (010) start to undergo the second dissolution process at 0.0017 ns. However, the Ca ions in $\beta\text{-C}_2\text{S}$ (100) remain within the first dissolution process until 40 ns, and no stable detachment from O_s ions is observed. At 40 ns, the stern layer and small pores appear in M3- C_3S (010); these features are not observed in $\beta\text{-C}_2\text{S}$ (100). These results corroborate that M3- C_3S (010) has higher reactivity than $\beta\text{-C}_2\text{S}$ (100) even during the initial hydration process, and the difference originates from the crystal structure. The subsequent hydration of M3- C_3S might be further promoted by the appearance of the pore solution saturation. Furthermore, ligand teeth structures are observed during the Ca ions dissolution for both M3- C_3S (010) and $\beta\text{-C}_2\text{S}$ (100). In other words, the dissolution of Ca ions is not a simple process of bonding a water molecule and breaking a Ca- O_s bond, and it is not independent of the environment. Ca ions remain with a given coordination and hop only when the surface oxygen is hydroxylated. Ca ions form ligand teeth structures with O_w ions, which facilitate the detachment of Ca ions from O_s ions. The umbrella sampling results illustrate that unbiased simulations are necessary to understand the dissolution mechanism before applying metadynamics or targeted molecular dynamics to characterize the energy barriers.

The reason for the higher reactivity of M3- C_3S (010) than $\beta\text{-C}_2\text{S}$ (100) is due to the contribution of the O_i ions in M3- C_3S . The O ions in the unit cell of M3- C_3S can be divided into four groups based on the Bader charge, with the first group (O_i ions) having the lowest number of valence electrons¹⁶. The Ca ions have an average higher valence electron number in M3- C_3S than in $\beta\text{-C}_2\text{S}$

due to bonding with the O_i ions. The higher reactivity of Ca and O ions in M3-C₃S will facilitate the hydration reaction. Specifically, in water molecule adsorption, the adsorption energies are more negative on M3-C₃S (010) than on β -C₂S (100)¹⁷, indicating that the water adsorption on M3-C₃S (010) is more energetically favorable. Meanwhile, these O_i ions provide more favorable adsorption sites for H ions than O_{si} ions, which thus benefits proton transfers and the formation of ligand teeth in M3-C₃S (010). Regarding the hydration rate, the faster hydration reaction of C₃S than C₂S also agrees well with the experimental data. Specifically, the experimental undersaturated dissolution rate of anhydrous monoclinic C₃S in deionized water at 20 °C is determined to be $-74.00 \mu\text{mol m}^{-2} \text{s}^{-1}$ while that of anhydrous β -C₂S is $-16.78 \mu\text{mol m}^{-2} \text{s}^{-1}$ ¹⁸.

The simulated results are consistent with the experimental results. For the simulated process, hydroxylated species have been observed at the instant of surface contact with water (1E-4 ns). This is consistent with the experimental results of Pustovgar et al.²¹ where hydroxylated Q⁰ silicate species are formed on the particles surface before contact with a large amount of water. Also, the experiments show that only the near surface of C₃S particles is hydroxylated in low amounts of polymeric silicate hydration products²¹, which is consistent with the simulated process (Fig. 4a).

Moreover, it should be noted that there are significant differences in the MD simulation in this study compared to previous ones about cement hydration. Firstly, a long simulation time of 40 ns is performed in this study compared to previous MD studies (ranging from tens of ps to a few ns)^{11, 19}. The extension of the simulation time allows Ca ions to dissolve in an unbiased way, which is not possible in previous MD studies. Secondly, a very short time step of 0.1 fs is chosen in this study, compared to 0.2-0.5 fs in many previous MD studies^{19, 20}, which ensures a more accurate and realistic simulation. Third, the simulations in this study are more realistic. The simulation employs a large slab model (~23000 in this study compared with <2000 in the literature⁵) together with the ReaxFF reactive force field, which allows for dynamic bonds formation and breaking, and accurately describes atomic motion in complex chemical environments^{1, 4}. Specifically, a large number of chemical reactions have occurred at the surfaces before Ca ions dissolution, and amorphous products are continuously formed at the surfaces, which is consistent with the real Ca ions dissolution process²¹. Fourth, we do not use any accelerated dissolution method, i.e. adding biasing forces or energies to the system. This means that a large number of chemical reactions have occurred on the surface before Ca ions dissolution and amorphous products are continuously formed on the surface, which is consistent with the real Ca ions dissolution process²¹. A detailed comparison of this work with two representative papers is provided in Supplementary Information (Supplementary Note 3).

In conclusion, using the unbiased method, this study makes four main contributions to the initial cement hydration, namely a new Ca dissolution process, two new general dissolution pathways, a key structure for Ca ions dissolution, and a detailed characterization of the hydration process (Supplementary Note 4). Briefly, we find that Ca ions on the M3-C₃S (010) surface must undergo two dissolution processes, whereas Ca ions in β -C₂S (100) mainly remain in the first dissolution process during the studied 40-ns period. The general dissolution pathways of Ca ions in C₃S and C₂S are resolved in this work. A new ligand teeth structure is discovered in the dissolution of Ca ions in C₃S and C₂S. In addition, the solid/water arrangement at the interface and the surface

structure after the initial hydration process, such as the appearance of aqueous layers and the small pores favoring H transfer in M3-C₃S, are characterized. Moreover, the free energy calculation illustrates that the hydroxylation state of the neighboring atoms is a key factor which enables Ca ions desorption, implying the necessity of unbiased MD simulations. Due to the fast reactivity of dissolution at the initial stage of cement hydration, there are obvious technical limitations to solve the dissolution mechanism at the molecular level through laboratory experiments²². The detailed characterization of the cement/water interface in this study will provide an important reference for revealing the dissolution mechanism of initial cement hydration and clarifying the reaction pathway for Ca ions dissolution. Overall, these results fill a gap in the current understanding of the dissolution process of Ca ions and open a new chapter in the study of the initial hydration process of cement.”

Overall, the study has potential to contribute to the field, but these recommended modifications or additions would enhance the clarity and significance of the research findings.

We greatly appreciate the valuable suggestions and comments, which we have carefully revised to improve the manuscript quality. Thanks to the reviewer for the efforts in reviewing our research.

Reviewer #3 (Remarks to the Author):

This manuscript investigated the initial hydration processes of M3-C3S (010) and β -C2S (100) from the dissolution of Ca ions, the solid/water interface and the surface structure. The authors presented many results, especially the novel dissolution pathways of Ca ions. It was found that Ca ions in M3-C3S (010) and β -C2S (100) surface undergo different dissolution processes, which may be due to the contribution of more O_i in C3S.

The subject may be interesting, however, I wonder no significant new findings with regard to cement hydration have been unraveled, since large numbers of previous publications discussed about this topic. The authors should clarify this, especially in comparison with "Ab initio mechanism revealing for tricalcium silicate dissolution." Nature Communications 13.1 (2022): 1253.

Sincerely thank you for your suggestions. We have clarified the differences from other MD simulations in the revised manuscript. In the following, we will first answer your questions and place the revised part of the manuscript after the response section.

The significant difference is the simulation model and time. We have achieved a simulation time of 40 ns, which is the longest simulation time for the initial hydration of cement and even hydration of other calcium-based minerals achieved by ReaxFF. The simulation times in previous MD studies were usually between a few tens of ps and a few ns^{11, 19}. The extension of simulation time allows us to discover the dissolution mechanism of Ca ions, which cannot be achieved by previous MD studies. Meanwhile, we chose a very short time step of 0.1 fs, compared to the usual 0.2-0.5 fs in previous MD studies^{19, 20}, which ensures more accurate and realistic simulations. Our simulation is more realistic in that we used a large model (~23000 in this study compared with <2000 in the literature¹⁹) together with the ReaxFF, which can correctly describe chemical reactions, allow for dynamic bonds formation and breaking, and accurately describe atomic motion in complex chemical environments^{1, 4}. In our study, we did not use any accelerated dissolution method, i.e. adding biasing forces or energies to the system. This means that a large number of chemical reactions have occurred on the surface before Ca ions dissolution and amorphous products are continuously formed on the surface, which is consistent with the real Ca ions dissolution process²¹. Moreover, a comparison between M3-C₃S and β -C₂S has been presented in the current study, in contrast to previous literature which analyses one phase or the other¹¹. A detailed comparison of this work with two representative papers is provided in Supplementary Information (Supplementary Note 3).

Using the unbiased method **we have characterized the step-by-step dissolution process** (Supplementary Note 4). In particular, for M3-C₃S in previous studies, it was considered that Ca ions dissolve directly after breaking all bonds with surface O ions, which is the dissolution of Ca ions by definition. However, our results show that the complete detachment of Ca ions from surface O ions in M3-C₃S does not achieve free movement in the water layer, but enters

a new dissolution process (the second dissolution process). The second dissolution process requires a tens of thousands fold longer period than the first dissolution process. The discovery of this new process revises our current understanding of Ca ions dissolution in M3-C₃S and provides a benchmark knowledge for the dissolution mechanism of other silicate minerals. **We found a new ligand teeth structure**, which is specifically expressed as Ca-O_w-H_{ab}-Ca. The ligand teeth widely exist in the Ca ions dissolution of M3-C₃S and β -C₂S, the appearance of which facilitates the detachment of Ca ions from the surface O ions. The formation of more ligand teeth structures is observed during the Ca ions dissolution of M3-C₃S. Moreover, multiple ligand teeth structures are also observed in the intermediate layer of C-S-H structure¹⁰, and such ligand teeth structures will play an important role in hydrates formation. Finally, we characterized the initial hydration process of the cement at 40 ns in multiple ways. These characterizations also led to many new findings that are not previously concluded. For example, **the appearance of aqueous layers and the small pores that favor H transfer in M3-C₃S**. These characterization results demonstrate in great detail the differences in the early hydration of M3-C₃S and β -C₂S. Moreover, this study provides huge simulation data for the initial cement hydration process, which paves the way for further in-depth as well as multi-faceted studies of cement hydration in the future.

We have also clearly indicated the necessity of our unbiased simulation method. Due to the access to larger times, we can investigate the process without using any biased method like metadynamics, which let us **understand correctly the Ca desorption mechanism**. Our new energy calculations illustrate perfectly the key difference. Previous studies, as the **Ab initio mechanism revealing for tricalcium silicate dissolution (Nature Communications 13.1 (2022): 1253)**¹⁴ which recently raised attention in the community, use DFT and metadynamics to explore the Ca dissolution from the surface. The selected collective variable is the Ca-O coordination to surface:water, which changes from 3:3 to 0:6, while the rest of the system remains “constant”, i.e. no chemical reaction takes place, just change in the Ca coordination number. In our unbiased simulations, we observe that the desorption is not independent of the environment. Ca remains with a given coordination and hops only when the surface oxygen is hydroxylated. From the unbiased MD trajectory, we have performed umbrella sampling (US) simulations along the original transition path in two scenarios. First without any constraint on the system, which represents the MD simulations done in this work. Second, constraining the H hopping to prevent O hydroxylation, which mimics the scenario in common DFT simulations or unreactive force field studies.

The US simulations were done with LAMMPS and the Colvars module. The reaction coordinate has been chosen as the main distance from the dissolving Ca atom (id = 360) to the surrounding Ca atoms (ids = 7792, 220, 2904, 260, and 20). The calculations were done at room temperature using the Noose-Hoover thermostat in the NVT ensemble. The time step was set to 0.1 fs and the total simulation time was 100 ps. For each scenario, 12 simulations were done where the reaction coordinate was taken from 5.15 Å to 5.7 Å in steps of 0.05 Å. For the "H-jump" scenario the spring constant for the umbrella simulations was set to 50 kcal/(mol·Å) and for the "no-H-jump" was set to 200 kcal/(mol·Å). In both scenarios, we made sure that all the path was correctly sampled. To recover the free energy from the different simulations we used WHAM¹⁵, with 100 bins.

Clearly, the free energy is considerably different if H hopping is allowed or not. When H hopping is allowed, a dissociated water molecule creates a hydroxyl group in a surface oxygen coordinated to Ca when Ca is at ~ 5.25 Å. That makes the Ca-O(H) bond weaker and therefore facilitates the movement of the Ca, which reaches a stable state at ~ 5.55 Å. The energy barrier is less than 1 kcal mol^{-1} (4 kJ mol^{-1}). When the water molecule is not allowed to dissociate, the Ca-O bond remains strong, and increasing the distance for the Ca desorption is unfavorable. The free energy increases constantly up to 4 kcal/mol (16 kJ/mol), and no final stable state was found. **This key result illustrates the necessity of unbiased simulations like those from this work to understand the dissolution mechanism** before applying metadynamics or targeted molecular dynamics to characterize the energy barriers.

Discussion section:

“Moreover, it should be noted that there are significant differences in the MD simulation in this study compared to previous ones about cement hydration. Firstly, a long simulation time of 40 ns is performed in this study compared to previous MD studies (ranging from tens of ps to a few ns)^{11, 19}. The extension of the simulation time allows Ca ions to dissolve in an unbiased way, which is not possible in previous MD studies. Secondly, a very short time step of 0.1 fs is chosen in this study, compared to 0.2-0.5 fs in many previous MD studies^{19, 20}, which ensures a more accurate and realistic simulation. Third, the simulations in this study are more realistic. The simulation employs a large slab model (~ 23000 in this study compared with <2000 in the literature⁵) together with the ReaxFF reactive force field, which allows for dynamic bonds formation and breaking, and accurately describes atomic motion in complex chemical environments^{1, 4}. Specifically, a large number of chemical reactions have occurred at the surfaces before Ca ions dissolution, and amorphous products are continuously formed at the surfaces, which is consistent with the real Ca ions dissolution process²¹. Fourth, we do not use any accelerated dissolution method, i.e. adding biasing forces or energies to the system. This means that a large number of chemical reactions have occurred on the surface before Ca ions dissolution and amorphous products are continuously formed on the surface, which is consistent with the real Ca ions dissolution process²¹. A detailed comparison of this work with two representative papers is provided in Supplementary Information (Supplementary Note 3).

In conclusion, using the unbiased method, this study makes four main contributions to the initial cement hydration, namely a new Ca dissolution process, two new general dissolution pathways, a key structure for Ca ions dissolution, and a detailed characterization of the hydration process (Supplementary Note 4). Briefly, we find that Ca ions on the M3-C₃S (010) surface must undergo two dissolution processes, whereas Ca ions in β -C₂S (100) mainly remain in the first dissolution process during the studied 40-ns period. The general dissolution pathways of Ca ions in C₃S and C₂S are resolved in this work. A new ligand teeth structure is discovered in the dissolution of Ca ions in C₃S and C₂S. In addition, the solid/water arrangement at the interface and the surface structure after the initial hydration process, such as the appearance of aqueous layers and the small pores favoring H transfer in M3-C₃S, are characterized. Moreover, the free energy calculation illustrates that the hydroxylation state of the neighboring atoms is a key factor which enables Ca ions desorption, implying the necessity of unbiased MD simulations. Due to the fast reactivity of

dissolution at the initial stage of cement hydration, there are obvious technical limitations to solve the dissolution mechanism at the molecular level through laboratory experiments²². The detailed characterization of the cement/water interface in this study will provide an important reference for revealing the dissolution mechanism of initial cement hydration and clarifying the reaction pathway for Ca ions dissolution. Overall, these results fill a gap in the current understanding of the dissolution process of Ca ions and open a new chapter in the study of the initial hydration process of cement.”

Result section:

“From the unbiased trajectory, we perform umbrella sampling (US) simulations along the original transition path in two scenarios (Fig. 2c). First without any constraint on the system, which represents the MD simulations done in this study (the H-jump scenario). Second, constraining the H hopping to prevent O hydroxylation, which mimics the scenario in common DFT simulations or unreactive force field studies (the no H-jump scenario). When H hopping is allowed, a dissociated water molecule creates a hydroxyl group in a surface oxygen coordinated to Ca ion when Ca ion is at ~ 5.25 Å. That makes the Ca-O(H) bond weaker and therefore facilitates the movement of the Ca ion, which reaches a stable state at ~ 5.55 Å. The energy barrier is less than 1 kcal mol⁻¹ (4 kJ mol⁻¹). When the water molecule is not allowed to dissociate, the Ca-O bond remains strong, and increasing the distance for the Ca ion desorption is unfavorable. The potential of mean force increases constantly up to 4 kcal mol⁻¹ (16 kJ mol⁻¹), and no final stable state is found. This illustrates the necessity of unbiased simulations to understand the dissolution mechanism before applying metadynamics or targeted molecular dynamics to characterize the energy barriers.”

Fig. 2 The second dissolution process of Ca ions from M3-C₃S (010). **a** The initial Ca ion structure of the second dissolution process, namely Structure 1. **b** The conceptualized flowchart illustrating the second dissolution process. **c** The potential of mean force of the transition from Structure 2 to 3. The transformation from Structure 2 to 3 might require a higher energy barrier than that from Structure 1 to 2 since it takes a much longer transformation time. **d** The evolution of Ca ions' structures with time. The structures of Ca ions are presented in parentheses as the number of Ca-O_s bonds, the number of ligand teeth, and the number of Ca-H₂O_w bonds. The surface is shown as the shaded box. The raw data curves in Fig. 2d are partially transparent and the smoothed curves are highlighted.

Discussion section:

“In other words, the dissolution of Ca ions is not a simple process of bonding a water molecule and breaking a Ca-O_s bond, and it is not independent of the environment. Ca ions remain with a given coordination and hop only when the surface oxygen is hydroxylated. Ca ions form ligand

teeth structures with O_w ions, which facilitate the detachment of Ca ions from O_s ions. The umbrella sampling results illustrate that unbiased simulations are necessary to understand the dissolution mechanism before applying metadynamics or targeted molecular dynamics to characterize the energy barriers.”

Supplementary Note section:

Supplementary Note 3

“The first comparison is made with the Nature Communications 13.1 (2022): 1253, in which the Ca dissolution process of M3-C₃S was initially discussed. However, the model in the literature contained less than 1000 atoms. Meanwhile, the simulation time remained on the scale of ps, approximately at 0.01 ns. The metadynamics approach was used in the literature to accelerate the structural evolution and pull the target Ca ions with special initial coordination out of a perfectly ordered crystal. Therefore, the simulation procedure in the literature was biased and not fully realistic, considering that the Ca dissolution only happened at a very limited and specific region. In this study, we do not use any accelerated dissolution method. This means that a large number of chemical reactions have occurred on the surface before Ca ions dissolution and amorphous products are continuously formed on the surface, which is consistent with the real Ca ions dissolution process. Therefore, we have a completely different configuration compared to this literature, which is believed to be more realistic. This will provide a more realistic interface and reaction path for cement hydration, which is essential for the comprehensive analysis of the cement hydration mechanism.

The second comparison is made with the study written by the authors in 2015, ACS Appl. Mater. Interfaces 2015, 7, 27, 14726-14733. The ACS study used the molecular dynamics of reaction force fields to understand the hydration of tricalcium silicate. First, static characteristics such as surface energy and water adsorption energy were analyzed. This led to the conclusion that a dynamic study of mineral hydration is necessary to observe local chemical reactions and reveal the hydration process. However, limited by the time scale (2 ns), it remained at the stage of water molecule adsorption, including the formation of hydroxyl pairs and the hopping of H ions to the interior, and no dissolution of Ca ions was observed. Therefore, the ACS paper could not shed light on the Ca dissolution, which is the major contribution of this study. Moreover, the simulation model is larger and a detailed comparison between M3-C₃S and β -C₂S has been presented in the current study.”

Supplementary Note 4

“This study provides four main contributions to the initial cement hydration, namely a new Ca dissolution process, two new general dissolution pathways, a key structure for Ca ions dissolution, and a detailed characterization of the hydration process.

A new Ca dissolution process: For M3-C₃S in previous studies, it was considered that Ca ions dissolve directly after breaking all bonds with surface O ions, which is the dissolution of Ca ions by definition. However, the results show that the complete detachment of Ca ions from surface O ions in M3-C₃S does not achieve free movement in the water layer, but enters a new dissolution

process, namely the second dissolution process. The second dissolution process requires a tens of thousands fold longer period than the first dissolution process. The discovery of this new process revises our current understanding of Ca ions dissolution in M3-C₃S and provides a benchmark knowledge for the dissolution mechanism of other silicate minerals.

Two new general dissolution pathways: In this study, the general pathways of Ca ions dissolution in M3-C₃S and β-C₂S have clarified after extensive statistics analysis, which is a breakthrough in Ca dissolution. The authors consider the revealed Ca ions dissolution pathways to be more realistic and representative since they are obtained from non-speculative, highly-accurate, and non-accelerated structural evolution MD simulations. That marks the opening of a new chapter in the study of initial cement hydration, from the stage of water molecules adsorption to the stage of Ca dissolution. The general Ca dissolution pathway will be an important theoretical basis for the Ca dissolution stage and a key step toward exploring the nucleation of cement hydration.

A key structure for Ca dissolution: In this study, a new ligand teeth structure is observed, which is specifically expressed as Ca-O_w-H_{ab}-Ca. The ligand teeth widely exist in the Ca ions dissolution of M3-C₃S and β-C₂S, the appearance of which facilitates the detachment of Ca ions from the surface O ions. The formation of more ligand teeth structures is observed during the Ca ions dissolution of M3-C₃S. Moreover, multiple ligand teeth structures are also observed in the intermediate layer of C-S-H structure, and such ligand teeth structures will play an important role in hydrates formation.

A detailed characterization of the hydration process: The initial hydration process of the cement at 40 ns is characterized in multiple ways. These characterizations also led to many new findings that are not previously found, for example, the appearance of aqueous layers and the small pores favoring H transfer in M3-C₃S. These characterization results demonstrate in great detail the differences in the early hydration of M3-C₃S and β-C₂S. Moreover, this study provides a huge simulation data for the initial cement hydration process, which paves the way for further in-depth as well as multi-faceted studies of cement hydration in the future.”

Has this work provided more understandings in initial cement hydration?

We are convinced that this work has provided more understanding in initial cement hydration, yet maybe not at the “traditional” or “service” scale of cement, since we are dealing with the atomic scale. It has been long recognized that understanding the nanoscale structure, properties, and mechanism of materials can help us to manipulate and improve their macroscopic properties. In cement, the relationship between nanoscale and macroscale is not trivial due to the multiscale and multicomponent nature of the material. We understand that a 40 ns long simulation of the dissolution process cannot give us results directly comparable with experiments. However, it is certain that without nanoscale knowledge we cannot know if we are missing something important.

At the atomic scale, it is clear for us that this work has provided understanding beyond the literature. As explained before, we have characterized in great detail several key features of the dissolution process, namely a new Ca dissolution process, two new general dissolution

pathways, a key structure for Ca ions dissolution, and a detailed characterization of the hydration process.

In particular, for M3-C₃S in previous studies, it was considered that Ca ions dissolve directly after breaking all bonds with surface O ions, which is the dissolution of Ca ions by definition. However, our results show that the complete detachment of Ca ions from surface O ions in M3-C₃S does not achieve free movement in the water layer, but enters a new dissolution process, namely the second dissolution process. The second dissolution process requires a tens of thousands fold longer period than the first dissolution process. The discovery of this new process revises our current understanding of Ca ions dissolution in M3-C₃S and provides a benchmark knowledge for the dissolution mechanism of other silicate minerals.

Secondly, we clarified the general pathways of Ca ions dissolution in M3-C₃S and β -C₂S after extensive statistical analysis, which we believe is a breakthrough in Ca dissolution. We consider the revealed Ca ions dissolution pathways to be more realistic and representative since they are obtained from non-speculative, highly-accurate, and non-accelerated structural evolution MD simulations. That marks the opening of a new chapter in the study of initial cement hydration, from the stage of water molecules adsorption to the stage of Ca ions dissolution. The general Ca dissolution pathway will be an important theoretical basis for the Ca dissolution stage and a key step toward exploring the nucleation of cement hydration.

The third point is that we found this new ligand teeth structure, which is specifically expressed as Ca-O_w-H_{ab}-Ca (Fig. 2a). The ligand teeth widely exist in the Ca ions dissolution of M3-C₃S and β -C₂S, the appearance of which facilitates the detachment of Ca ions from the surface O ions. The formation of more ligand teeth structures is observed during the Ca ions dissolution of M3-C₃S. Moreover, multiple ligand teeth structures are also observed in the intermediate layer of C-S-H structure²⁴, and such ligand teeth structures will play an important role in hydrates formation.

Together with the new free energy calculation, it is proven that the hydroxylation state of the neighboring atoms is a key factor which enables Ca desorption. This gives us a different viewpoint in relation to prior DFT calculations or accelerated MD simulations, i.e., the validity of their results, which further illustrates the contribution of this work. We have re-summarized the main contributions of this study with respect to the initial cement hydration in the revised manuscript.

Introduction section:

“In this study, we use ReaxFF reactive force field simulations to reveal the initial hydration processes of C₃S and C₂S from multiple perspectives, including chemical bonding, dissolution of Ca ions, and surface structural evolution. Here, we present the breakthrough discovery of a new dissolution process of Ca ions on the C₃S surface and reveal, for the first time, the general dissolution pathways of Ca ions from C₃S and C₂S surfaces. The unbiased molecular dynamics (MD) simulation, as validated by the free energy calculation, reveals that the hydroxylation state

of the neighboring atoms is a key factor which enables Ca desorption. A new ligand teeth structure is discovered during the dissolution of Ca ions, which is critical for its dissolution process. In addition, the solid/water arrangement near the interface is characterized after the initial hydration process, including the appearance of aqueous layers and the small pores favoring H transfer in M3-C₃S. Overall, our discoveries fill a gap in the understanding of the dissolution process of Ca ions as well as represent a new milestone in the study of the initial hydration processes of C₃S and C₂S.”

Discussion section:

“In conclusion, using the unbiased method, this study makes four main contributions to the initial cement hydration, namely a new Ca dissolution process, two new general dissolution pathways, a key structure for Ca ions dissolution, and a detailed characterization of the hydration process (Supplementary Note 4). Briefly, we find that Ca ions on the M3-C₃S (010) surface must undergo two dissolution processes, whereas Ca ions in β-C₂S (100) mainly remain in the first dissolution process during the studied 40-ns period. The general dissolution pathways of Ca ions in C₃S and C₂S are resolved in this work. A new ligand teeth structure is discovered in the dissolution of Ca ions in C₃S and C₂S. In addition, the solid/water arrangement at the interface and the surface structure after the initial hydration process, such as the appearance of aqueous layers and the small pores favoring H transfer in M3-C₃S, are characterized. Moreover, the free energy calculation illustrates that the hydroxylation state of the neighboring atoms is a key factor which enables Ca ions desorption, implying the necessity of unbiased MD simulations. Due to the fast reactivity of dissolution at the initial stage of cement hydration, there are obvious technical limitations to solve the dissolution mechanism at the molecular level through laboratory experiments²². The detailed characterization of the cement/water interface in this study will provide an important reference for revealing the dissolution mechanism of initial cement hydration and clarifying the reaction pathway for Ca ions dissolution. Overall, these results fill a gap in the current understanding of the dissolution process of Ca ions and open a new chapter in the study of the initial hydration process of cement.”

Supplementary Note section:

Supplementary Note 4

“This study provides four main contributions to the initial cement hydration, namely a new Ca dissolution process, two new general dissolution pathways, a key structure for Ca ions dissolution, and a detailed characterization of the hydration process.

A new Ca dissolution process: For M3-C₃S in previous studies, it was considered that Ca ions dissolve directly after breaking all bonds with surface O ions, which is the dissolution of Ca ions by definition. However, the results show that the complete detachment of Ca ions from surface O ions in M3-C₃S does not achieve free movement in the water layer, but enters a new dissolution process, namely the second dissolution process. The second dissolution process requires a tens of thousands fold longer period than the first dissolution process. The discovery of this new process revises our current understanding of Ca ions dissolution in M3-C₃S and provides a benchmark knowledge for the dissolution mechanism of other silicate minerals.

Two new general dissolution pathways: In this study, the general pathways of Ca ions dissolution in M3-C₃S and β-C₂S have clarified after extensive statistics analysis, which is a breakthrough in Ca dissolution. The authors consider the revealed Ca ions dissolution pathways to be more realistic and representative since they are obtained from non-speculative, highly-accurate, and non-accelerated structural evolution MD simulations. That marks the opening of a new chapter in the study of initial cement hydration, from the stage of water molecules adsorption to the stage of Ca dissolution. The general Ca dissolution pathway will be an important theoretical basis for the Ca dissolution stage and a key step toward exploring the nucleation of cement hydration.

A key structure for Ca dissolution: In this study, a new ligand teeth structure is observed, which is specifically expressed as Ca-O_w-H_{ab}-Ca. The ligand teeth widely exist in the Ca ions dissolution of M3-C₃S and β-C₂S, the appearance of which facilitates the detachment of Ca ions from the surface O ions. The formation of more ligand teeth structures is observed during the Ca ions dissolution of M3-C₃S. Moreover, multiple ligand teeth structures are also observed in the intermediate layer of C-S-H structure, and such ligand teeth structures will play an important role in hydrates formation.

A detailed characterization of the hydration process: The initial hydration process of the cement at 40 ns is characterized in multiple ways. These characterizations also led to many new findings that are not previously found, for example, the appearance of aqueous layers and the small pores favoring H transfer in M3-C₃S. These characterization results demonstrate in great detail the differences in the early hydration of M3-C₃S and β-C₂S. Moreover, this study provides a huge simulation data for the initial cement hydration process, which paves the way for further in-depth as well as multi-faceted studies of cement hydration in the future.”

Can the simulation results agree with the experiments?

Again, a 40ns simulation is hardly comparable with long term experiments. Nevertheless, there are several interpretations that have been proposed. There is a high degree of agreement between our MD simulation results and the experimental results, which we have added in the discussion section of the revised manuscript.

First, it is well known that C₃S dissolves faster than C₂S. The undersaturated dissolution rate of anhydrous Monoclinic C₃S in deionized water at 20 °C was $-74.00 \mu\text{mol m}^{-2} \text{s}^{-1}$ while that of anhydrous β-C₂S was $-16.78 \mu\text{mol m}^{-2} \text{s}^{-1}$ ¹⁸, implying a qualitative agreement between conclusions obtained from MD simulations and laboratory experiments.

Second, our simulation process is consistent with experimental observations. For example, Pustovgar et al.²¹ analyzed the hydration of C₃S by nuclear magnetic resonance (NMR) spectroscopy. The NMR experiments revealed the formation of hydroxylated Q⁰ silicate species on the particle surface before contact with large amounts of water. In the simulations, the hydroxylated species rapidly appeared at the instant of contact of the surface with water (1E-4 ns). Meanwhile, the NMR experiments show that only the near surface of Ca₃SiO₅ particles were hydroxylated at low amounts of polymeric silicate hydration products²¹, which is consistent with the simulated process (Fig. 4d).

Discussion section:

“ Regarding the hydration rate, the faster hydration reaction of C_3S than C_2S also agrees well with the experimental data. Specifically, the experimental undersaturated dissolution rate of anhydrous monoclinic C_3S in deionized water at 20 °C is determined to be $-74.00 \mu\text{mol m}^{-2} \text{s}^{-1}$ while that of anhydrous $\beta\text{-}C_2S$ is $-16.78 \mu\text{mol m}^{-2} \text{s}^{-1}$ ¹⁸.

The simulated results are consistent with the experimental results. For the simulated process, hydroxylated species have been observed at the instant of surface contact with water (1E-4 ns). This is consistent with the experimental results of Pustovgar et al.²¹ where hydroxylated Q^0 silicate species are formed on the particles surface before contact with a large amount of water. Also, the experiments show that only the near surface of C_3S particles is hydroxylated in low amounts of polymeric silicate hydration products²¹, which is consistent with the simulated process (Fig. 4a).”

Furthermore, the mechanism of dissolution of Ca ions needs to be explained more concisely, which is the key part of hydration and not adequately discussed.

With all respect, we think that the Ca dissolution mechanism is explained in great detail. We have a complete section devoted to it, and then we also tackle it in the discussion section.

Nevertheless, we have further summarized the dissolution process of Ca ions in the revised manuscript. The dissolution of Ca ions is not a simple process of bonding a water molecule and breaking a Ca- O_s bond. Ca ions form ligand teeth structures with O_w ions, which help Ca ions to detach from O_s ions. The desorption is not independent of the environment. Ca ions remain with a given coordination and hop only when the surface oxygen is hydroxylated. The Ca ions of M3- C_3S form multiple ligand teeth structures and require a new dissolution process to break this structure to achieve free movement.

With all this and the additional calculation of the energy barriers, we believe that the detailed discussion of the general dissolution pathways can help the readers to clearly understand the dissolution process of Ca ions.

Discussion section:

“In other words, the dissolution of Ca ions is not a simple process of bonding a water molecule and breaking a Ca- O_s bond, and it is not independent of the environment. Ca ions remain with a given coordination and hop only when the surface oxygen is hydroxylated. Ca ions form ligand teeth structures with O_w ions, which facilitate the detachment of Ca ions from O_s ions. The umbrella sampling results illustrate that unbiased simulations are necessary to understand the dissolution mechanism before applying metadynamics or targeted molecular dynamics to

characterize the energy barriers.”

And the following questions should be answered before consideration for publication in Nature Communications.

1. Fig 1: the number of Ca-O_w bond increase slowly before 1.5 ns, more discussions are needed.

Thanks for your valuable suggestions. The discussion we have added in the revised manuscript is as follows:

“The under-coordination of surface Ca ions is significantly reduced, or even depleted, after the rapid water adsorption, resulting in the slow increase of Ca-O_w bonds afterwards.”

2. In “Initial hydration process” section: what does the “H_{ab}” means? Does it mean the H atoms bonded to O_s and O_w (bonded to Ca)?

Thank you for your careful review. The H_{ab} represents H ions are bonded to O_s and O_w with Ca ions and we have clarified it in the revised manuscript.

“In the following text, we investigate the structure of H_{ab} ions (where H_{ab} represents H ions bonded to O_s and O_w with Ca ions) to characterize the initial hydration process.”

3. Line 105-107: the curves of the number of bonds, both the Ca-O_w bonds and O-H_{ab} bonds, is fluctuating, so the gradients of them are preferred to be adopted rather than the specific increased values.

Thanks for your valuable suggestions. We have added the increase proportion of O_w-H_{ab} and Ca-O_w in the revised manuscript. We believe that the expression can be understood in a clear way through both the incremental values and the proportion.

“Notably, an increase of 32 O_w-H_{ab} bonds (15.92% increase relative to the O_w-H_{ab} bonds at 1.5 ns) is observed in M3-C₃S (010) between 1.5–6 ns, which is identical to the increase in the number of Ca-O_w bonds (16.58% increase relative to the Ca-O_w bonds at 1.5 ns). In β-C₂S (100), an increase of 36 O_w-H_{ab} bonds (23.52% increase relative to the O_w-H_{ab} bonds at 1.5 ns) occurs at 1.5–6 ns,

which is 1.9 times the number of increased Ca-O_w bonds (14.72% increase relative to the Ca-O_w bonds at 1.5 ns).”

4. The reason for the differences in the increase rates of O_w-H_{ab} bond in M3-C3S and β-C2S is not clear. How does the H_{ab} bond to Ca in the Ca-O_w-H_{ab}-Ca structure (Line 110–112:)?

Sincere thanks for your suggestions. We have made it clear that the reason for the differences. Specifically, M3-C₃S (010) tends to form more Ca-O_w-H_{ab}-Ca structures than β-C₂S (100) during Ca ions dissolution. This structure results in the formation of multiple Ca-O_w bonds along with one O_w-H_{ab} bond. We have explained the above-mentioned differences and added a schematic diagram of the Ca-O_w-H_{ab}-Ca structure during the revision (Supplementary Fig. 2).

“The differences in the rates of O_w-H_{ab} bond increase may be due to the M3-C₃S (010) tending to form more Ca-O_w-H_{ab}-Ca structures (Supplementary Fig. 2) than β-C₂S (100) during Ca ions dissolution. In this structure, multiple Ca-O_w bonds are formed along with one O_w-H_{ab} bond, resulting in the decreased ratio of Ca-O_w and O_w-H_{ab} (smaller than two in the molecular adsorption of water). Note that the presence of Ca-O_w-H_{ab}-Ca still dominates in β-C₂S (100) even at 1.5–6 ns, though the increased ratio of Ca-O_w and O_w-H_{ab} is high (Supplementary Tables 1, 2 and Note 1). A more detailed discussion is provided in the dissolution process of Ca ions section.”

Supplementary Figure 2. The Ca-O_w-H_{ab}-Ca structures of the dissolved Ca ion. The dissolved Ca ion consists of three Ca_{dis}-O_wH (tridentate) bonds and three Ca_{dis}-H₂O_w bonds.

5. Line 116: the word, “bonds”, is repetitive.

Thanks for your careful review. We have removed the extra “bonds”.

6. The ligand teeth are responsible for the difference between the O_w-H_{ab} bond increase and twice that of $Ca-O_w$ bonds. For the dissolution of Ca in $\beta-C_2S$, why does the increase of O_w-H_{ab} bonds is nearly twice (1.9 times, line 107) that of $Ca-O_w$ bonds while “there is a higher proportion of Ca ions with ligand teeth in most dissolution steps” (line 190–191)?

Sincere thanks for your valuable comments. We have provided detailed explanations for the above-mentioned results. In the following, we will first answer your questions and place the revised part of the manuscript after the response section.

Just a kind reminder that the 1.9 times is the comparison between the incremental value (Supplementary Table 1). The O_w-H_{ab} bonds have increased almost twofold compared to $Ca-O_w$ bonds in $\beta-C_2S$ (100), compared to almost equal in $M3-C_3S$ (010). The above results indicate $M3-C_3S$ (010) tends to form more $Ca-O_w-H_{ab}-Ca$ structures (Supplementary Fig. 2) than $\beta-C_2S$ (100) during 1.5 – 6 ns. Since the above ratio is calculated using the incremental values during 1.5 – 6 ns, it doesn't imply the overall structural statistics.

We have counted the structures of six-coordinated Ca ions of $\beta-C_2S$ (100) between 1.5 and 6 ns (Supplementary Table 2). The Ca ions with ligand teeth were recorded 77075 times (68.62%) and the Ca ions without ligand teeth were recorded 35251 times (31.38%). Therefore, we believe that ligand teeth are widely present in the dissolution process of $\beta-C_2S$ (100) between 1.5 and 6 ns. We have added the above explanation in the revised manuscript and Supplementary Note 1.

“The differences in the rates of O_w-H_{ab} bond increase may be due to the $M3-C_3S$ (010) tending to form more $Ca-O_w-H_{ab}-Ca$ structures (Supplementary Fig. 2) than $\beta-C_2S$ (100) during Ca ions dissolution. In this structure, multiple $Ca-O_w$ bonds are formed along with one O_w-H_{ab} bond, resulting in the decreased ratio of $Ca-O_w$ and O_w-H_{ab} (smaller than two in the molecular adsorption of water). Note that the presence of $Ca-O_w-H_{ab}-Ca$ still dominates in $\beta-C_2S$ (100) even at 1.5–6 ns, though the increased ratio of $Ca-O_w$ and O_w-H_{ab} is high (Supplementary Tables 1, 2 and Note 1). A more detailed discussion is provided in the dissolution process of Ca ions section.”

Supplementary Note 1

“The 1.9 times is a comparison between the incremental values within 1.5–6 ns. The number of bonds at 1.5 ns and 6 ns is provided in the Supplementary information (Supplementary Table 1). The overall $Ca-O_w$ and O_w-H_{ab} ratios for $\beta-C_2S$ (100) are 1.19 and 1.28 at 1.5 and 6 ns, respectively. The overall ratios for $M3-C_3S$ (010) are lower, 1.04 at both 1.5 and 6 ns (Supplementary Table 1).

Moreover, the structures of six-coordinated Ca ions of $\beta-C_2S$ (100) between 1.5 and 6 ns are counted (Supplementary Table 2). The Ca ions with ligand teeth are recorded 77075 times

(68.62%) and the Ca ions without ligand teeth are recorded 35251 times (31.38%). Therefore, it is believed that ligand teeth are widely present in the dissolution process of β -C₂S (100) between 1.5 and 6 ns.”

Supplementary Table 1. The number of Ca-O_w and O_w-H_{ab} bonds of β -C₂S (100) and M3-C₃S (010).

Type	Time (ns)	Counts		Ratios of Ca-O _w
		Ca-O _w bonds	O _w -H _{ab} bonds	bonds and O _w - H _{ab} bonds
β -C ₂ S (100)	1.5	129	153	1.19
	6	148	189	1.28
	1.5–6	19 (Increased)	36 (Increased)	1.89
M3-C ₃ S (010)	1.5	193	201	1.04
	6	225	233	1.04
	1.5–6	32 (Increased)	32 (Increased)	1

Supplementary Table 2. The structures of six-coordinated Ca ions of β -C₂S (100) between 1.5 and 6 ns

Structures*	Counts	Structures*	Counts
(4, 1, 1)	25337	(2, 2, 2)	159
(4, 0, 2)	5441	(2, 3, 1)	0
(4, 2, 0)	3012	(2, 4, 0)	0
(3, 1, 2)	11374	(1, 0, 5)	0
(3, 0, 3)	19538	(1, 1, 4)	2502
(3, 2, 1)	179	(1, 2, 3)	1375
(3, 3, 0)	0	(1, 3, 2)	0
(2, 1, 3)	33137	(1, 4, 1)	0
(2, 0, 4)	10272	(1, 5, 0)	0

* The structures of Ca ions are presented in parentheses as the number of Ca-O_s bonds, the number of ligand teeth, and the number of Ca-H₂O_w bonds.

7. Line 185: the “Fig. 2a” is initial Ca ion structure of M3-C₃S, should this be Fig. 3a?

Thank you for your careful review. We have revised the Fig. 2a to Fig. 3a.

8. In “Solid/Water interface and ...” section: more discussions are needed for the five regions of the interface. Details of the atomic density change with time will be more informative.

Sincere thanks for your suggestions. We have added more details to characterize the region in the revised manuscript and added the variation of atomic density with time (Supplementary Fig. 9 and Fig. 10). We found that the first region has clear crystal structure characteristics. The second region penetrates to the interior with the hopping of H ions and the reaction of O_w with Ca ions. The third region gradually appears with the dissolution of Ca ions. The fourth region mainly consists of free and molecularly adsorbed water molecules.

“In M3-C₃S (010), we observe that the interface can be divided into five regions at 40 ns (Fig. 4a). The first region corresponds to the bulk, up to 133 Å, which is mainly characterized by the crystalline structure. The second region (133–139 Å) is the gugenheim interface³⁰, in which the bulk structure is gradually lost and the solid and water coexist. The second region penetrates to the interior with the hopping of H ions and the reaction of O_w with Ca ions (Supplementary Fig. 9). The third region is the stern layer (139–142 Å), and consists of dissolved Ca ions still linked to the surface and water molecules, with an O_w density similar to that of bulk water. The third region gradually appears with the dissolution of Ca ions (Supplementary Fig. 9). We note that the third region contains not only free water molecules but also dissociative and molecularly adsorbed water molecules. The diffuse (fourth region) layer (142–147 Å) is defined as the transition layer, in which only a small amount of Ca ions and almost no dissociative adsorbed water molecules are present. It mainly consists of free and molecularly adsorbed water molecules. The fifth region is considered to be the bulk liquid and consists entirely of free water molecules. The same regions are also observed in β-C₂S (100) (Fig. 4b and Supplementary Fig. 10). The atomic and structural density of these four layers of β-C₂S (100) are similar to those in M3-C₃S (010).”

Fig. 4 The solid/water interface at 40 ns of the initial hydration process. The atomic and structure density of a M3-C₃S (010) and b β-C₂S (100). The density is averaged 1 ps before 40 ns to avoid excessive volatility.

Supplementary Figure 9. The solid/water interface of the initial hydration process with time. The atomic and structure density of M3-C₃S (010) at a 0.1 ns, b 1.5ns, c 6 ns, and d 40 ns. The density is averaged 1 ps before 40 ns to avoid excessive volatility.

Supplementary Figure 10. The solid/water interface of the initial hydration process with time. The atomic and structure density of β - C_2S (100) at a 0.1 ns, b 1.5 ns, c 6 ns, and d 40 ns. The density is averaged 1 ps before 40 ns to avoid excessive volatility.

9. Fig.4: the legend of subgraph a and b should be consistent, otherwise it may confuse the readers.

Thank you for your careful review. We have used the same legend in the revised manuscript to avoid any possible misunderstanding.

Fig. 4 The solid/water interface at 40 ns of the initial hydration process. The atomic and structure density of **a** M3-C₃S (010) and **b** β -C₂S (100). The density is averaged 1 ps before 40 ns to avoid excessive volatility.

10. At 0 ns, the range of large pore sizes in β -C₂S is larger than that in M3-C₃S, and the number of pores in β -C₂S is also more than that in M3-C₃S (Fig. 5c-d), but why does the “M3-C₃S (010) has higher reactivity than β -C₂S (100)” (line 272)?

Sincerely thank you for your suggestions. We have added more discussion about the initial pore distribution in the revised manuscript and Supplementary Information (Supplementary Fig. 11 and Note 2). The large diameter pore size is contributed by the vacuum layer (Supplementary Fig. 11). The vertical coordinate of Fig. 5c and Fig 5d is the derivative distribution of the pore size. The value of the vertical coordinate of β -C₂S (100) is larger than

that of M3-C₃S (010), indicating that the β-C₂S (100) pore size is concentrated between 8 and 9 Å. The surface atoms of β-C₂S (100) are more uniformly arranged than that of M3-C₃S (010). The pore distributions of β-C₂S (100) and M3-C₃S (010) are wider at 30 ns than at 0 ns, indicating a more disordered arrangement of surface ions after hydration.

Moreover, since the large diameter pores are mainly contributed by the vacuum layer, we are more interested in the formation of small diameter pores. It is believed that these pores are beneficial to further proton transfers in the near-surface region.

“At 0 ns, the large diameter pores distribution of β-C₂S (100) is more concentrated between 8–9 Å, having a higher peak value than that of M3-C₃S (010), indicating that the initial surface atomic arrangement of β-C₂S (100) is more ordered than that of M3-C₃S (010).”

“A more detailed explanation and schematic figure of the large diameter pores are provided in the Supplementary Information (Supplementary Fig. 11 and Note 2).”

Supplementary Note 2

“The initial pore size is contributed by the vacuum layer of the pore model (Supplementary Fig. 11). The vertical coordinate of Fig. 5c and Fig 5d is the derivative distribution of the pore size. The value of the vertical coordinate of β-C₂S (100) is larger than that of M3-C₃S (010), indicating that the β-C₂S (100) pore diameter is concentrated between 8 and 9 Å. The surface atoms of β-C₂S (100) are more uniformly arranged than that of M3-C₃S (010).

Moreover, since the large diameter pores are mainly contributed by the vacuum layer, we are more interested in the formation of small diameter pores, since these pores are beneficial to proton transfers in the near-surface region.”

Supplementary Figure 11. The schematic figure of the large pore distribution of M3-C₃S (010) and β-C₂S (100) at 0 ns.

Reviewer #4 (Remarks to the Author):

This paper titled ‘The initial stages of cement hydration at the molecular level’ looks at the very early age dissolution of C3S and C2S surfaces using molecular dynamics with reactive force field. The paper is well written and well organized. But, I am not entirely convinced at the novelty and the impact of this paper.

ReaxFF has been used previously by the authors to look at C3S dissolution (2015 paper?). In the current paper, from what I understand some of the key additional aspects are the longer simulation (30ns instead of 10ns) and the coordination evolution of calcium ions during dissolution. The same methodology is applied to C2S to get a comparison. However, we do know from some of the DFT simulations (as authors themselves wrote) that the origin of low reactivity or dissolution of C2S is due to the absence of ionic oxygen.. What are we learning new and impactful here?

Sincerely thank you for your suggestions. We have re-summarized the differences from previous studies and new findings in the revised manuscript to clarify the novelty of the study. We will then first answer your questions and place the revised part of the manuscript after the response section.

The significant difference is the simulation model and time. We have achieved a simulation time of 40 ns, which is the longest simulation time for the initial hydration of cement and even hydration of other calcium-based minerals achieved by ReaxFF. The simulation times in previous MD studies were usually between a few tens of ps and a few ns^{11, 19}. The extension of simulation time allows us to discover the dissolution mechanism of Ca ions, which cannot be achieved by previous MD studies. Meanwhile, we chose a very short time step of 0.1 fs, compared to the usual 0.2-0.5 fs in previous MD studies^{19, 20}, which ensures more accurate and realistic simulations. Our simulation is more realistic in that we used a large model (~23000 in this study compared with <2000 in the literature¹⁹) together with the ReaxFF, which can correctly describe chemical reactions, allow for dynamic bonds formation and breaking, and accurately describe atomic motion in complex chemical environments^{1,4}. In our study, we did not use any accelerated dissolution method, i.e. adding biasing forces or energies to the system. This means that a large number of chemical reactions have occurred on the surface before Ca ions dissolution and amorphous products are continuously formed on the surface, which is consistent with the real Ca ions dissolution process²¹. Moreover, a comparison between M3-C₃S and β-C₂S has been presented in the current study, in contrast to previous literature which analyses one phase or the other¹¹. A detailed comparison of this work with two representative papers is provided in Supplementary Information (Supplementary Note 3).

In the following two paragraphs, we provided a detailed comparison with two representative studies in the literature. The first study was mentioned by the reviewer as written by the

authors in 2015, ACS Appl. Mater. Interfaces 2015, 7, 27, 14726-14733. The ACS study used the molecular dynamics of reaction force fields to understand the hydration of tricalcium silicate. First, static characteristics such as surface energy and water adsorption energy were analyzed. This led to the conclusion that a dynamic study of mineral hydration is necessary to observe local chemical reactions and understand them. However, limited by the time scale (the simulation time in ACS 2015 study was only 2 ns), it remained at the stage of water molecule adsorption, including the formation of hydroxyl pairs and the hopping of H ions to the interior, and no dissolution of Ca ions was observed. Therefore, the ACS paper could not shed light on the dissolution mechanism of Ca ions, which is the major contribution of this study. Moreover, the simulation model is larger and a detailed comparison between M3-C₃S and β -C₂S has been presented in the current study.

The second one is Nature Communications 13.1 (2022): 1253, in which the Ca dissolution process of M3-C₃S was discussed. However, the model in the literature contained less than 1000 atoms. Meanwhile, the simulation time remained on the scale of ps, approximately at 0.01 ns. The metadynamics approach was used in the literature to accelerate the structural evolution and pull the target Ca ions with special initial coordination out of a perfectly ordered crystal. Therefore, the simulation procedure in the literature was biased and not fully realistic, considering that the Ca ions dissolution only happened at a very limited and specific region. In our study, we did not use any accelerated dissolution method. This means that a large number of chemical reactions have occurred on the surface before Ca ions dissolution and amorphous products are continuously formed on the surface, which is consistent with the real Ca ions dissolution process²¹. Therefore we have a completely different configuration compared to this literature, and as supported by the free energy results in the revised manuscript, our results are believed to be more realistic. Our unbiased simulation will also provide a more realistic interface and reaction path for cement hydration, which is essential for the comprehensive analysis of the cement hydration mechanism.

At the atomic scale, it is clear for us that this work has provided understanding beyond the literature. As explained before, we have characterized in great detail several key features of the dissolution process, namely a new Ca dissolution process, two new general dissolution pathways, a key structure for Ca ions dissolution, and a detailed characterization of the hydration process.

In particular, for M3-C₃S in previous studies, it was considered that Ca ions dissolve directly after breaking all bonds with surface O ions, which is the dissolution of Ca ions by definition. However, our results show that the complete detachment of Ca ions from surface O ions in M3-C₃S does not achieve free movement in the water layer, but enters a new dissolution process (the second dissolution process). The second dissolution process requires a tens of thousands fold longer period than the first dissolution process. The discovery of this new process revises our current understanding of Ca ions dissolution in M3-C₃S and provides a benchmark knowledge for the dissolution mechanism of other silicate minerals.

Secondly, we clarified the general pathways of Ca ions dissolution in M3-C₃S and β -C₂S after extensive statistical analysis, which we believe is a breakthrough in Ca dissolution. We

consider the revealed Ca ions dissolution pathways to be more realistic and representative since they are obtained from non-speculative, highly-accurate, and non-accelerated structural evolution MD simulations. That marks the opening of a new chapter in the study of initial cement hydration, from the stage of water molecules adsorption to the stage of Ca ions dissolution. The general Ca dissolution pathway will be an important theoretical basis for the Ca dissolution stage and a key step toward exploring the nucleation of cement hydration.

The third point is that we found this new ligand teeth structure, which is specifically expressed as Ca-O_w-H_{ab}-Ca (Fig. 2a). The ligand teeth widely exist in the Ca ions dissolution of M3-C₃S and β-C₂S, the appearance of which facilitates the detachment of Ca ions from the surface O ions. The formation of more ligand teeth structures is observed during the Ca ions dissolution of M3-C₃S. Moreover, multiple ligand teeth structures are also observed in the intermediate layer of C-S-H structure²⁴, and such ligand teeth structures will play an important role in hydrates formation.

Together with the new free energy calculation, it is proven that the hydroxylation state of the neighboring atoms is a key factor which enables Ca desorption. This gives us a different viewpoint in relation to prior DFT calculations or accelerated MD simulations, i.e., the validity of their results, which further illustrates the contribution of this work. We have re-summarized the main contributions of this study with respect to the initial cement hydration in the revised manuscript.

We have also provided a more comprehensive comparison between C₂S and C₃S, in addition to the ionic oxygen mentioned by the reviewer. Firstly, in the first paragraph of the discussion, we summarized the results of the reactivity difference between the two clinkers obtained in this study. These results corroborate that M3-C₃S (010) has higher reactivity than β-C₂S (100) during the initial hydration process. The dissolution of Ca ions undergoes two dissolution processes in M3-C₃S, while in β-C₂S it remains at dissolution process one. Then, in the second paragraph of the discussion, we indicated that the reasons for the higher reactivity of M3-C₃S than β-C₂S would be the O_i ions in the unit cell of M3-C₃S, which has the lowest number of valence electrons¹⁶. The Ca ions have a higher valence electron number in M3-C₃S than in β-C₂S due to bonding with the O_i ions. The higher reactivity of Ca and O_i ions in M3-C₃S will facilitate the hydration reaction. Specifically, in water molecule adsorption, the adsorption energies were more negative on M3-C₃S (010) than on β-C₂S (100), indicating that M3-C₃S (010) is more energetically-favorable for water molecule adsorption¹⁷. Meanwhile, these O_i ions provide more favorable adsorption sites for H ions than O_{si} ions, which thus benefits proton transfers and the formation of ligand teeth in M3-C₃S (010).

Introduction section:

“In this study, we use ReaxFF reactive force field simulations to reveal the initial hydration processes of C₃S and C₂S from multiple perspectives, including chemical bonding, dissolution of Ca ions, and surface structural evolution. Here, we present the breakthrough discovery of a new dissolution process of Ca ions on the C₃S surface and reveal, for the first time, the general dissolution pathways of Ca ions from C₃S and C₂S surfaces. A new ligand teeth structure is

discovered during the dissolution of Ca ions, which is critical for its dissolution process. In addition, the solid/water arrangement near the interface is characterized after the initial hydration process, including the appearance of aqueous layers and the small pores favoring H transfer in M3-C₃S. Overall, our discoveries fill a gap in the understanding of the dissolution process of Ca ions as well as represent a new milestone in the study of the initial hydration processes of C₃S and C₂S.”

Discussion section:

“During the initial hydration within 40 ns, dozens of Ca ions in M3-C₃S (010) are completely detached from O_s ions and form a stable six-coordinated structure with water. The Ca ions in M3-C₃S (010) start to undergo the second dissolution process at 0.0017 ns. However, the Ca ions in β-C₂S (100) remain within the first dissolution process until 40 ns, and no stable detachment from O_s ions is observed. At 40 ns, the stern layer and small pores appear in M3-C₃S (010); these features are not observed in β-C₂S (100). These results corroborate that M3-C₃S (010) has higher reactivity than β-C₂S (100) even during the initial hydration process, and the difference originates from the crystal structure. The subsequent hydration of M3-C₃S might be further promoted by the appearance of the pore solution saturation. Furthermore, ligand teeth structures are observed during the Ca ions dissolution for both M3-C₃S (010) and β-C₂S (100). In other words, the dissolution of Ca ions is not a simple process of bonding a water molecule and breaking a Ca-O_s bond, and it is not independent of the environment. Ca ions remain with a given coordination and hop only when the surface oxygen is hydroxylated. Ca ions form ligand teeth structures with O_w ions, which facilitate the detachment of Ca ions from O_s ions. The umbrella sampling results illustrate that unbiased simulations are necessary to understand the dissolution mechanism before applying metadynamics or targeted molecular dynamics to characterize the energy barriers.

The reason for the higher reactivity of M3-C₃S (010) than β-C₂S (100) is due to the contribution of the O_i ions in M3-C₃S. The O ions in the unit cell of M3-C₃S can be divided into four groups based on the Bader charge, with the first group (O_i ions) having the lowest number of valence electrons¹⁶. The Ca ions have an average higher valence electron number in M3-C₃S than in β-C₂S due to bonding with the O_i ions. The higher reactivity of Ca and O ions in M3-C₃S will facilitate the hydration reaction. Specifically, in water molecule adsorption, the adsorption energies are more negative on M3-C₃S (010) than on β-C₂S (100)¹⁷, indicating that the water adsorption on M3-C₃S (010) is more energetically favorable. Meanwhile, these O_i ions provide more favorable adsorption sites for H ions than O_{si} ions, which thus benefits proton transfers and the formation of ligand teeth in M3-C₃S (010). Regarding the hydration rate, the faster hydration reaction of C₃S than C₂S also agrees well with the experimental data. Specifically, the experimental undersaturated dissolution rate of anhydrous monoclinic C₃S in deionized water at 20 °C is determined to be -74.00 μmol m⁻² s⁻¹ while that of anhydrous β-C₂S is -16.78 μmol m⁻² s⁻¹¹⁸.”

Discussion section:

“Moreover, it should be noted that there are significant differences in the MD simulation in this study compared to previous ones about cement hydration. Firstly, a long simulation time of 40 ns

is performed in this study compared to previous MD studies (ranging from tens of ps to a few ns)^{11, 19}. The extension of the simulation time allows Ca ions to dissolve in an unbiased way, which is not possible in previous MD studies. Secondly, a very short time step of 0.1 fs is chosen in this study, compared to 0.2-0.5 fs in many previous MD studies^{19, 20}, which ensures a more accurate and realistic simulation. Third, the simulations in this study are more realistic. The simulation employs a large slab model (~23000 in this study compared with <2000 in the literature⁵) together with the ReaxFF reactive force field, which allows for dynamic bonds formation and breaking, and accurately describes atomic motion in complex chemical environments^{1, 4}. Specifically, a large number of chemical reactions have occurred at the surfaces before Ca ions dissolution, and amorphous products are continuously formed at the surfaces, which is consistent with the real Ca ions dissolution process²¹. Fourth, we do not use any accelerated dissolution method, i.e. adding biasing forces or energies to the system. This means that a large number of chemical reactions have occurred on the surface before Ca ions dissolution and amorphous products are continuously formed on the surface, which is consistent with the real Ca ions dissolution process²¹. A detailed comparison of this work with two representative papers is provided in Supplementary Information (Supplementary Note 3).

In conclusion, using the unbiased method, this study makes four main contributions to the initial cement hydration, namely a new Ca dissolution process, two new general dissolution pathways, a key structure for Ca ions dissolution, and a detailed characterization of the hydration process (Supplementary Note 4). Briefly, we find that Ca ions on the M3-C₃S (010) surface must undergo two dissolution processes, whereas Ca ions in β-C₂S (100) mainly remain in the first dissolution process during the studied 40-ns period. The general dissolution pathways of Ca ions in C₃S and C₂S are resolved in this work. A new ligand teeth structure is discovered in the dissolution of Ca ions in C₃S and C₂S. In addition, the solid/water arrangement at the interface and the surface structure after the initial hydration process, such as the appearance of aqueous layers and the small pores favoring H transfer in M3-C₃S, are characterized. Moreover, the free energy calculation illustrates that the hydroxylation state of the neighboring atoms is a key factor which enables Ca ions desorption, implying the necessity of unbiased MD simulations. Due to the fast reactivity of dissolution at the initial stage of cement hydration, there are obvious technical limitations to solve the dissolution mechanism at the molecular level through laboratory experiments²². The detailed characterization of the cement/water interface in this study will provide an important reference for revealing the dissolution mechanism of initial cement hydration and clarifying the reaction pathway for Ca ions dissolution. Overall, these results fill a gap in the current understanding of the dissolution process of Ca ions and open a new chapter in the study of the initial hydration process of cement.”

Supplementary Note section:

Supplementary Note 3

“The first comparison is made with the Nature Communications 13.1 (2022): 1253, in which the Ca dissolution process of M3-C₃S was initially discussed. However, the model in the literature contained less than 1000 atoms. Meanwhile, the simulation time remained on the scale of ps, approximately at 0.01 ns. The metadynamics approach was used in the literature to accelerate the

structural evolution and pull the target Ca ions with special initial coordination out of a perfectly ordered crystal. Therefore, the simulation procedure in the literature was biased and not fully realistic, considering that the Ca dissolution only happened at a very limited and specific region. In this study, we do not use any accelerated dissolution method. This means that a large number of chemical reactions have occurred on the surface before Ca ions dissolution and amorphous products are continuously formed on the surface, which is consistent with the real Ca ions dissolution process. Therefore, we have a completely different configuration compared to this literature, which is believed to be more realistic. This will provide a more realistic interface and reaction path for cement hydration, which is essential for the comprehensive analysis of the cement hydration mechanism.

The second comparison is made with the study written by the authors in 2015, ACS Appl. Mater. Interfaces 2015, 7, 27, 14726-14733. The ACS study used the molecular dynamics of reaction force fields to understand the hydration of tricalcium silicate. First, static characteristics such as surface energy and water adsorption energy were analyzed. This led to the conclusion that a dynamic study of mineral hydration is necessary to observe local chemical reactions and reveal the hydration process. However, limited by the time scale (2 ns), it remained at the stage of water molecule adsorption, including the formation of hydroxyl pairs and the hopping of H ions to the interior, and no dissolution of Ca ions was observed. Therefore, the ACS paper could not shed light on the Ca dissolution, which is the major contribution of this study. Moreover, the simulation model is larger and a detailed comparison between M3-C₃S and β-C₂S has been presented in the current study.”

Supplementary Note 4

“This study provides four main contributions to the initial cement hydration, namely a new Ca dissolution process, two new general dissolution pathways, a key structure for Ca ions dissolution, and a detailed characterization of the hydration process.

A new Ca dissolution process: For M3-C₃S in previous studies, it was considered that Ca ions dissolve directly after breaking all bonds with surface O ions, which is the dissolution of Ca ions by definition. However, the results show that the complete detachment of Ca ions from surface O ions in M3-C₃S does not achieve free movement in the water layer, but enters a new dissolution process, namely the second dissolution process. The second dissolution process requires a tens of thousands fold longer period than the first dissolution process. The discovery of this new process revises our current understanding of Ca ions dissolution in M3-C₃S and provides a benchmark knowledge for the dissolution mechanism of other silicate minerals.

Two new general dissolution pathways: In this study, the general pathways of Ca ions dissolution in M3-C₃S and β-C₂S have clarified after extensive statistics analysis, which is a breakthrough in Ca dissolution. The authors consider the revealed Ca ions dissolution pathways to be more realistic and representative since they are obtained from non-speculative, highly-accurate, and non-accelerated structural evolution MD simulations. That marks the opening of a new chapter in the study of initial cement hydration, from the stage of water molecules adsorption to the stage of Ca dissolution. The general Ca dissolution pathway will be an important theoretical basis for the

Ca dissolution stage and a key step toward exploring the nucleation of cement hydration.

A key structure for Ca dissolution: In this study, a new ligand teeth structure is observed, which is specifically expressed as Ca-O_w-H_{ab}-Ca. The ligand teeth widely exist in the Ca ions dissolution of M3-C₃S and β-C₂S, the appearance of which facilitates the detachment of Ca ions from the surface O ions. The formation of more ligand teeth structures is observed during the Ca ions dissolution of M3-C₃S. Moreover, multiple ligand teeth structures are also observed in the intermediate layer of C-S-H structure, and such ligand teeth structures will play an important role in hydrates formation.

A detailed characterization of the hydration process: The initial hydration process of the cement at 40 ns is characterized in multiple ways. These characterizations also led to many new findings that are not previously found, for example, the appearance of aqueous layers and the small pores favoring H transfer in M3-C₃S. These characterization results demonstrate in great detail the differences in the early hydration of M3-C₃S and β-C₂S. Moreover, this study provides a huge simulation data for the initial cement hydration process, which paves the way for further in-depth as well as multi-faceted studies of cement hydration in the future.”

Now, a major drawback of this paper is the lack of discussion on how their results can be compared to the reality. Would dissolution happen at such perfect flat surfaces? What does the current understanding brings to the community.

In this work, we do not aim to compare our results with “reality”. Our aim is to seed light to a complex process at the nanoscale that cannot be characterized by experimental methods. We understand that a 40 ns long simulation of the dissolution process cannot give us results directly comparable with experiments. However, it has been long recognized that understanding the nanoscale structure, properties, and mechanism of materials can help us to manipulate and improve their macroscopic properties. In cement, the relationship between nanoscale and macroscale is not trivial due to the multiscale and multicomponent nature of the material. But it is certain that without nanoscale knowledge we cannot know if we are missing something important.

We are convinced that this work has provided more understanding in initial cement hydration, yet maybe not at the “traditional” or “service” scale of cement. The cement community is very heterogeneous in their expertise and despite part of them may be focused on the engineering scale, chemists, physicists, and geochemists are more and more interested in the atomic scale understanding of the material. At the same time, we think that these results could be relevant for any other Ca-based mineral as the reviewer suggests in the following question. That includes biomaterials like Ca phosphates, technologically relevant materials like Ca sulphates, and Ca-carbonates which have a huge interest in carbon capture and sequestration.

In addition, our new energy calculations demonstrate that previous works using DFT + metadynamics like the **Ab initio mechanism revealing for tricalcium silicate dissolution Nature Communications 13.1 (2022): 1253**¹⁴ calculations employed models that might not

represent properly the dissolution conditions. This is in our opinion a very relevant result, as the Nat. Comm. paper has raised important attention and the results might be incorrect. Due to the access to larger times, we can investigate the process without using any biased method like metadynamics, which let us **understand correctly the Ca desorption mechanism**. Our new energy calculations illustrate perfectly the key difference. In the **Nature Communications 13.1 (2022): 1253**¹⁴, the selected collective variable is the Ca-O coordination to surface:water, which changes from 3:3 to 0:6, while the rest of the system remains “constant”, i.e. no chemical reaction takes place, just change in the Ca coordination number. In our unbiased simulations, we observe that the desorption is not independent of the environment. Ca remains with a given coordination and hops only when the surface oxygen is hydroxylated. From the unbiased MD trajectory, we have performed umbrella sampling (US) simulations along the original transition path in two scenarios. First without any constraint on the system, which represents the MD simulations done in this work. Second, constraining the H hopping to prevent O hydroxylation, which mimics the scenario in common DFT simulations or unreactive force field studies.

The US simulations were done with LAMMPS and the Colvars module. The reaction coordinate has been chosen as the main distance from the dissolving Ca atom (id = 360) to the surrounding Ca atoms (ids = 7792, 220, 2904, 260, and 20). The calculations were done at room temperature using the Noose-Hoover thermostat in the NVT ensemble. The time step was set to 0.1 fs and the total simulation time was 100 ps. For each scenario, 12 simulations were done where the reaction coordinate was taken from 5.15 Å to 5.7 Å in steps of 0.05 Å. For the "H-jump" scenario the spring constant for the umbrella simulations was set to 50 kcal/(mol·Å) and for the "no-H-jump" was set to 200 kcal/(mol·Å). In both scenarios, we made sure that all the path was correctly sampled. To recover the free energy from the different simulations we used WHAM¹⁵, with 100 bins.

Clearly, the free energy is considerably different if H hopping is allowed or not. When H hopping is allowed, a dissociated water molecule creates a hydroxyl group in a surface oxygen coordinated to Ca when Ca is at ~5.25 Å. That makes the Ca-O(H) bond weaker and therefore facilitates the movement of the Ca, which reaches a stable state at ~5.55 Å. The energy barrier is less than 1 kcal mol⁻¹ (4 kJ mol⁻¹). When the water molecule is not allowed to dissociate, the Ca-O bond remains strong, and increasing the distance for the Ca desorption is unfavorable. The free energy increases constantly up to 4 kcal/mol (16 kJ/mol), and no final stable state is found.

Regarding the specific question on the dissolution of flat surfaces. It has been proved that the main source of dissolution are dislocations, as in any other mineral. However, the fact that the dissolution rate is faster on dislocations does not imply that dissolution does not take place on flat surfaces, just that it is slower. For example, as the experimental process, numerous chemical reactions have occurred on the surfaces before the dissolution of Ca ions, and the surfaces have lost their initial perfect structure. Numerous chemical reactions continued to occur within the 40 ns, resulting a large amount of amorphous material on the surfaces. That is to say, the dissolution processes of Ca ions are unanticipated, without artificial acceleration and intervention, and occur naturally with hydration. The mechanism that we observe and

describe in this work is therefore completely valid.

Considering the reviewer's comment regarding the comparison with the reality, we have added more comparison and discussion on this aspect. It is noted that only if we would like to compare the dissolution rates of a flat surface with the macroscopic one or just present a qualitative comparison, we would be likely making a mistake. Generally speaking, there is a high degree of agreement between our MD simulation results and the experimental results, as supported by the following two aspects.

First, it is well known that C_3S dissolves faster than C_2S . The undersaturated dissolution rate of anhydrous Monoclinic C_3S in deionized water at 20 °C was $-74.00 \mu\text{mol m}^{-2} \text{s}^{-1}$ while that of anhydrous $\beta\text{-}C_2S$ was $-16.78 \mu\text{mol m}^{-2} \text{s}^{-1}$ ¹⁸, implying a qualitative agreement between conclusions obtained from MD and laboratory experiments.

Second, our simulation process is consistent with experimental observations. For example, Pustovgar et al.²¹ analyzed the hydration of C_3S by nuclear magnetic resonance (NMR) spectroscopy. The NMR experiments revealed the formation of hydroxylated Q^0 silicate species on the particle surface before contact with large amounts of water. In the simulations, the hydroxylated species rapidly appeared at the instant of contact of the surface with water (1E-4 ns). Meanwhile, the NMR experiments show that only the near surface of Ca_3SiO_5 particles were hydroxylated at low amounts of polymeric silicate hydration products²¹, which is consistent with the simulated process (Fig. 4d).

Result section:

“From the unbiased trajectory, we perform umbrella sampling (US) simulations along the original transition path in two scenarios (Fig. 2c). First without any constraint on the system, which represents the MD simulations done in this study (the H-jump scenario). Second, constraining the H hopping to prevent O hydroxylation, which mimics the scenario in common DFT simulations or unreactive force field studies (the no H-jump scenario). When H hopping is allowed, a dissociated water molecule creates a hydroxyl group in a surface oxygen coordinated to Ca ion when Ca ion is at $\sim 5.25 \text{ \AA}$. That makes the Ca-O(H) bond weaker and therefore facilitates the movement of the Ca ion, which reaches a stable state at $\sim 5.55 \text{ \AA}$. The energy barrier is less than 1 kcal mol⁻¹ (4 kJ mol⁻¹). When the water molecule is not allowed to dissociate, the Ca-O bond remains strong, and increasing the distance for the Ca ion desorption is unfavorable. The potential of mean force increases constantly up to 4 kcal mol⁻¹ (16 kJ mol⁻¹), and no final stable state is found. This illustrates the necessity of unbiased simulations to understand the dissolution mechanism before applying metadynamics or targeted molecular dynamics to characterize the energy barriers.”

Fig. 2 The second dissolution process of Ca ions from M3-C₃S (010). **a** The initial Ca ion structure of the second dissolution process, namely Structure 1. **b** The conceptualized flowchart illustrating the second dissolution process. **c** The potential of mean force of the transition from Structure 2 to 3. The transformation from Structure 2 to 3 might require a higher energy barrier than that from Structure 1 to 2 since it takes a much longer transformation time. **d** The evolution of Ca ions' structures with time. The structures of Ca ions are presented in parentheses as the number of Ca-O_s bonds, the number of ligand teeth, and the number of Ca-H₂O_w bonds. The surface is shown as the shaded box. The raw data curves in Fig. 2d are partially transparent and the smoothed curves are highlighted.

Discussion section:

“In other words, the dissolution of Ca ions is not a simple process of bonding a water molecule and breaking a Ca-O_s bond, and it is not independent of the environment. Ca ions remain with a given coordination and hop only when the surface oxygen is hydroxylated. Ca ions form ligand

teeth structures with O_w ions, which facilitate the detachment of Ca ions from O_s ions. The umbrella sampling results illustrate that unbiased simulations are necessary to understand the dissolution mechanism before applying metadynamics or targeted molecular dynamics to characterize the energy barriers.”

Discussion section:

“The simulated results are consistent with the experimental results. For the simulated process, hydroxylated species have been observed at the instant of surface contact with water ($1E-4$ ns). This is consistent with the experimental results of Pustovgar et al.²¹ where hydroxylated Q^0 silicate species are formed on the particles surface before contact with a large amount of water. Also, the experiments show that only the near surface of C_3S particles is hydroxylated in low amounts of polymeric silicate hydration products²¹, which is consistent with the simulated process (Fig. 4a).”

The complexity of dissolution is not discussed to give the perspective on how these simulations are original or a ‘breakthrough’.

We believe that the main breakthrough is the full characterization of the mechanism itself. As we discussed before, the dissolution is often described as a single Ca desorption step or a multi-step desorption but without considering the evolution of the neighboring atoms. We characterized the processes will all its complexity and set the ground for future studies, not only on cement but also on other minerals.

The chemistry of calcium silicate dissolution is accompanied by electron transfer and bond breakage and formation at the solid/liquid interface. Previous studies of cluster/water systems have shown that even a simple isotope exchange in one molecule contains many counterintuitive pathways involving the concerted motion of many atoms²⁵. Not to mention the large number of interactions between water molecules involved in the dissolution of calcium silicate as well as the water/interface reactions and the dynamics change within the solid. The complexity of the dissolution process and the difficulty in accurately describing it have been agreed upon as the calcium silicate dissolution is influenced by multiple coupling parameters^{19, 21}. This is also the reason why calcium silicate hydration is still unclear. Specifically, only the water molecule adsorption of the initial hydration process contains a variety of chemical reactions such as initial hydroxylation, molecular and dissociative water adsorption, and proton exchange. Numerous chemical reactions occur at the solid-liquid interface, which together influences the hydration process. In this study, the ReaxFF reaction force field was chosen to determine only the initial conditions (e.g., temperature and pressure), and numerous chemical reactions occur naturally with time. For the first time, two general dissolution pathways for Ca ions of C_3S and C_2S were identified in the large amount of data recorded in the 40ns simulation. This fills the gap in the Ca dissolution phase, clarifies the reaction mechanism of the dissolution phase in the initial cement hydration process, and opens a new chapter in the study of initial cement hydration. Moreover, the authors believe

that the differences from previous MD simulations and the new findings such as the new free energy calculation make it clear that the simulations are original and present a major breakthrough.

“However, numerous chemical reactions coupled influence the initial hydration process, the dissolution pathway of Ca ions remains unclear.”

Though the paper might look interesting from the dissolution process of Ca ions, isn't it something one can expect during the dissolution process of any calcium based mineral? Would the 2015 paper already have such mechanisms observed in the trajectory?

We agree with the reviewer, it is very likely that the same dissolution processes of Ca ions could be expected for any Ca-based mineral. But it is not a negative point! On the contrary, that increases the potential impact of the paper, since Ca-based minerals are very important from a technological, environmental, and geological point of view. We focused on two minerals present in cement, but as far as we know, the Ca dissolution process has not been described in such detail and with such a complex model for any other mineral.

In the dissolution of calcium-based minerals, it is often assumed that the dissolution process of Ca is a simple process of water molecules binding and breaking Ca-O_s bonds. Previous studies of cluster/water systems have shown that even a simple isotope exchange in one molecule contains many counterintuitive pathways involving the concerted motion of many atoms²⁵. Therefore, this is still only a guess, and the dissolution process of Ca ions in C₃S and C₂S remains unclear due to the experimental technical limitations of Ca ions dissolution and the fact that Ca dissolution is influenced by multiple coupled parameters. In addition, our results revise the general knowledge of Ca dissolution processes in C₃S and C₂S and clarify the general reaction pathways. We have summarized the dissolution process of Ca ions in the revised manuscript. The dissolution of Ca ions is not a simple process of bonding a water molecule and breaking a Ca-O_s bond. Ca ions form ligand teeth structures with O_w ions, which help Ca ions to detach from O_s ions. The Ca ions of M3-C₃S form multiple ligand teeth structures and require a new dissolution process to break this structure to achieve free movement. These new findings are also of significant reference for understanding the dissolution processes of other minerals.

Regarding the 2015 paper, it was unable to observe the Ca dissolution due to the limitation of simulation time (2 ns). It is true that the methodology is the same as in the 2015 paper, but extending the simulation time x20 has allowed us to observe new and important phenomena. After that, substantial characterization and analysis were performed, and accordingly, many novel findings have been reached.

In the revised manuscript, we have clearly presented the novel findings, the comparison with

previous MD calculations, and the significance of the current study, as detailed in the response to the first comment.

It is my personal view that I don't see if it brings anything significant to the understanding of dissolution of these minerals to the cement community. It is possibly an incremental knowledge that is gained and still limited to 30ns.

This comment is already answered with our previous responses. To summarize our main points:

- It is true that the methodology is the same, but extending the simulation time gave us access to mechanisms that have not been observed before. It is important to note that the general dissolution process of Ca ions in C₃S and C₂S was unknown for the cement community before this study.
- Our study cannot be directly compared to the macroscale dissolution. However, understanding nanoscale processes might give us hints and clues on how to modify or control it.
- We have not only extended the previous simulation time 20 times, but also investigated a second dissolution process, and compare the mechanism in both systems.
- Our results indicate that previous works used incomplete models, and the DFT energy barriers from the literature, may be wrong.

The authors believe that the innovative findings of this paper and the huge simulation data of this paper are both of great significance to the cement community.

Another aspect is how well the coordination chemistries of such a dynamic process is realistically captured by the Reaxff. I don't know if you compare this with the recent DFT simulations , reference number 19.

Sincerely thank you for your suggestions. The authors have strong confidence of ReaxFF reactive force field in capturing the reality of these dynamic coordination chemistry processes. First, ReaxFF is a reactive force field based on a dynamic bond order, which is determined on the fly with respect to the instantaneous bond distance¹. Thus, ReaxFF allows for bond dynamics formation and dissociation and a correct description of chemical reactions^{1, 2}. Meanwhile, MD simulations using ReaxFF do not require a priori knowledge of the reaction; only the initial conditions (e.g., temperature and pressure) need to be determined and the reaction occurs naturally with time³. For large and complex molecular systems involving chemical reactions, MD simulations using ReaxFF reactive force field can be performed

faster than DFT calculations and with an accuracy closer to that of DFT⁴. Currently, ReaxFF has been successfully applied to various systems such as carbon-based systems^{5,6}, silicon and silicon oxide (crystalline structure) systems^{2,7}, aluminum and alumina systems⁸, etc. ReaxFF has also been shown to accurately describe the hydration of systems such as calcium silicate hydrate^{9,10}. Second, the accuracy of the ReaxFF parameters used in this study for describing the hydration process have been verified in detail in previous studies^{11,12,13}. Therefore, the authors believe that the ReaxFF reactive force field can accurately describe the initial hydration reaction with an accuracy closer to that of DFT. Moreover, in this study a large slab model with approximately 23000 atoms is used, which is not possible in DFT calculations, so the force field of ReaxFF is chosen for the simulation.

We could make a detailed comparison of reference number 19 (Nature Communications 13.1 (2022): 1253) in the revised manuscript. The model in the literature contained less than 1000 atoms, which in this study is about 23000 atoms. Meanwhile, the simulation time remains on the scale of ps, approximately at 0.01 ns, which is 40 ns in the current study. But more important, our results indicate that in ref 19 the authors used incomplete models, and the energy barriers may be wrong. The metadynamics approach is used in the literature to accelerate the structural evolution and pull the target Ca ions with special initial coordination out of a perfectly ordered crystal. The water speciation and the hydroxylation of the surface was not taken into account, and we prove here with the new free energy calculations that they are indispensable for the dissolution. In our study, we did not use any accelerated dissolution method. This means that a large number of chemical reactions have occurred on the surface before Ca ions dissolution and amorphous products are continuously formed on the surface, which is consistent with the real Ca ions dissolution process²¹. Therefore, we have a completely different configuration compared to this literature, which is believed to be more realistic. This will provide a more realistic interface and reaction path for cement hydration, which is essential for the comprehensive analysis of the cement hydration mechanism.

“Third, the simulations in this study are more realistic. The simulation employs a large slab model (~23000 in this study compared with <2000 in the literature¹⁹) together with the ReaxFF reactive force field, which allows for dynamic bonds formation and breaking, and accurately describes atomic motion in complex chemical environments^{1,4}.”

Supplementary Note section:

Supplementary Note 3

“The first comparison is made with the Nature Communications 13.1 (2022): 1253, in which the Ca dissolution process of M3-C₃S was initially discussed. However, the model in the literature contained less than 1000 atoms. Meanwhile, the simulation time remained on the scale of ps, approximately at 0.01 ns. The metadynamics approach was used in the literature to accelerate the structural evolution and pull the target Ca ions with special initial coordination out of a perfectly ordered crystal. Therefore, the simulation procedure in the literature was biased and not fully realistic, considering that the Ca dissolution only happened at a very limited and specific region.

In this study, we do not use any accelerated dissolution method. This means that a large number of chemical reactions have occurred on the surface before Ca ions dissolution and amorphous products are continuously formed on the surface, which is consistent with the real Ca ions dissolution process. Therefore, we have a completely different configuration compared to this literature, which is believed to be more realistic. This will provide a more realistic interface and reaction path for cement hydration, which is essential for the comprehensive analysis of the cement hydration mechanism.

The second comparison is made with the study written by the authors in 2015, ACS Appl. Mater. Interfaces 2015, 7, 27, 14726-14733. The ACS study used the molecular dynamics of reaction force fields to understand the hydration of tricalcium silicate. First, static characteristics such as surface energy and water adsorption energy were analyzed. This led to the conclusion that a dynamic study of mineral hydration is necessary to observe local chemical reactions and reveal the hydration process. However, limited by the time scale (2 ns), it remained at the stage of water molecule adsorption, including the formation of hydroxyl pairs and the hopping of H ions to the interior, and no dissolution of Ca ions was observed. Therefore, the ACS paper could not shed light on the Ca dissolution, which is the major contribution of this study. Moreover, the simulation model is larger and a detailed comparison between M3-C₃S and β-C₂S has been presented in the current study.”

The authors conclude ‘this study reproduces the initial hydration processes of ...’ How is it ‘reproducing’ ?

Sincerely thank you for your suggestions. We have replaced ‘replace’ with ‘investigate’ in the revised manuscript to avoid confusing the readers.

“Here, we use molecular dynamics simulations to investigate the unbiased initial hydration processes of tricalcium silicate (C₃S) and dicalcium silicate (C₂S) up to 40 ns.”

References

1. Yu Y, Wang B, Wang M, Sant G, Bauchy M. Revisiting silica with ReaxFF: Towards improved predictions of glass structure and properties via reactive molecular dynamics. *Journal of Non-Crystalline Solids* **443**, 148–154 (2016).
2. van Duin ACT, Strachan A, Stewman S, Zhang Q, Xu X, Goddard WA. ReaxFFSiO Reactive Force Field for Silicon and Silicon Oxide Systems. *The Journal of Physical Chemistry A* **107**, 3803–3811 (2003).
3. Newsome DA, Sengupta D, Foroutan II, Russo MF, van Duin ACT. Oxidation of Silicon Carbide by O₂ and H₂O: A ReaxFF Reactive Molecular Dynamics Study, Part I. *The Journal of Physical Chemistry C* **116**, 16111–16121 (2012).
4. Li X, Mo Z, Liu J, Guo L. Revealing chemical reactions of coal pyrolysis with GPU-enabled ReaxFF molecular dynamics and cheminformatics analysis. *Molecular Simulation* **41**, 13–27 (2015).
5. Mueller JE, van Duin ACT, Goddard WA, III. Development and Validation of ReaxFF Reactive Force Field for Hydrocarbon Chemistry Catalyzed by NiCl₂. *The Journal of Physical Chemistry C* **114**, 4939–4949 (2010).
6. Jensen BD, Bandyopadhyay A, Wise KE, Odegard GM. Parametric Study of ReaxFF Simulation Parameters for Molecular Dynamics Modeling of Reactive Carbon Gases. *Journal of Chemical Theory and Computation* **8**, 3003–3008 (2012).
7. Norman P, *et al.* The Structure of Silica Surfaces Exposed to Atomic Oxygen. *The Journal of Physical Chemistry C* **117**, 9311–9321 (2013).
8. Zhang Q, Çağın T, van Duin A, Goddard WA, Qi Y, Hector LG. Adhesion and nonwetting–wetting transition in the Al/ α -Al₂O₃ interface. *Physical Review B* **69**, 045423 (2004).
9. Abdolhosseini Qomi MJ, *et al.* Combinatorial molecular optimization of cement hydrates. *Nature Communications* **5**, 4960 (2014).
10. Bauchy M, Abdolhosseini Qomi MJ, Bichara C, Ulm F-J, Pellenq RJM. Nanoscale Structure of Cement: Viewpoint of Rigidity Theory. *The Journal of Physical Chemistry C* **118**, 12485–12493 (2014).
11. Manzano H, Durgun E, López-Arbeloa I, Grossman JC. Insight on Tricalcium Silicate Hydration and Dissolution Mechanism from Molecular Simulations.

ACS Applied Materials & Interfaces **7**, 14726–14733 (2015).

12. Manzano H, Pellenq RJM, Ulm F-J, Buehler MJ, van Duin ACT. Hydration of Calcium Oxide Surface Predicted by Reactive Force Field Molecular Dynamics. *Langmuir* **28**, 4187–4197 (2012).
13. Fogarty JC, Aktulga HM, Grama AY, van Duin ACT, Pandit SA. A reactive molecular dynamics simulation of the silica–water interface. *Journal of Chemical Physics* **132**, (2010).
14. Qi C, *et al.* Ab initio calculation of the adsorption of As, Cd, Cr, and Hg heavy metal atoms onto the illite(001) surface: Implications for soil pollution and reclamation. *Environmental Pollution* **312**, 120072 (2022).
15. Arctxabalcta XM, López-Zorrilla J, Labbez C, Etxebarria I, Manzano H. A potential CSH nucleation mechanism: atomistic simulations of the portlandite to CSH transformation. *Cement and Concrete Research* **162**, 106965 (2022).
16. Qi C, Xu X, Chen Q. Hydration reactivity difference between dicalcium silicate and tricalcium silicate revealed from structural and Bader charge analysis. *International Journal of Minerals, Metallurgy and Materials* **29**, 335–344 (2022).
17. Qi C, Spagnoli D, Fourie A. DFT-D study of single water adsorption on low-index surfaces of calcium silicate phases in cement. *Applied Surface Science* **518**, 146255 (2020).
18. Nicoleau L, Nonat A, Perrey D. The di- and tricalcium silicate dissolutions. *Cement and Concrete Research* **47**, 14–30 (2013).
19. Li Y, Pan H, Liu Q, Ming X, Li Z. Ab initio mechanism revealing for tricalcium silicate dissolution. *Nature Communications* **13**, 1253 (2022).
20. Claverie J, Bernard F, Cordeiro JMM, Kamali-Bernard S. Ab initio molecular dynamics description of proton transfer at water–tricalcium silicate interface. *Cement and Concrete Research* **136**, 106162 (2020).
21. Pustovgar E, *et al.* Understanding silicate hydration from quantitative analyses of hydrating tricalcium silicates. *Nature Communications* **7**, 10952 (2016).
22. Juilland P, Gallucci E. Morpho-topological investigation of the mechanisms and kinetic regimes of alite dissolution. *Cement and Concrete Research* **76**,

180–191 (2015).

23. Claverie J, Wang Q, Kamali-Bernard S, Bernard F. Assessment of the reactivity and hydration of Portland cement clinker phases from atomistic simulation: A critical review. *Cement and Concrete Research* **154**, 106711 (2022).
24. Svenum I-H, Ringdalen IG, Bleken FL, Friis J, Höche D, Swang O. Structure, hydration, and chloride ingress in C-S-H: Insight from DFT calculations. *Cement and Concrete Research* **129**, 105965 (2020).
25. Casey WI, Rustad JR. Reaction Dynamics, Molecular Clusters, and Aqueous Geochemistry. *Annual Review of Earth and Planetary Sciences* **35**, 21–46 (2007).

REVIEWER COMMENTS

Reviewer #1 (Remarks to the Author):

Key results:

The Revised manuscript explains the significance of the results more elaborately. The authors have improved the manuscript by extending the simulation from 30 ns to 40 ns, additional supplementary information, and extending the explanation of the proposed comments. All the corrections are done regarding visualization.

It is understandable for the Ca dissolution from β -C2S will be computationally expensive using the current method (unbiased MD simulation). Still, the revised version has explained the early hydration phenomena, especially dissolution. However, the Authors would consider a detailed study of the belite system in future research.

The current version is recommended for publication.

Reviewer #2 (Remarks to the Author):

Authors modified the manuscript well based on reviewers' comments.

Reviewer #3 (Remarks to the Author):

In the revised manuscript, differences from other studies and contributions to initial cement hydration are emphasized. However, it is doubtful to bring many valuable and novel findings to investigation of cement hydration, particularly due to the lack of experimental validation for the proposed two-step dissolution mechanism of Ca ions. And the following points may be helpful for improving this article.

- 1) Line 114-130: for the fluctuating curves of Ca-Ow bonds and O-Habs bonds, the incremental value or proportion can cause significant error. The fitting slope between 1.5-6ns are still preferred to be adopted.
- 2) The proposed Ca-Ow-Hab-Ca structures would give the impression that the Hab are bonded to Ca, but in fact as displayed in Supplementary Figure2, the Hab are bonded to Ow. Ca-OwHab-Ca may be a more appropriate expression.
- 3) Line 363-370: the fourth difference seems to be repetitive from the third one, and the cited references (in line 366 and 370, respectively) are different.
- 4) Supplementary Figure9: the atomic density distributions of Ca, Si, Os at 1.5ns show weird shifts compared to those at 0.1ns, 6ns and 40ns.

Reviewer #4 (Remarks to the Author):

Thank you for the elaborate replies. Following are my comments -

The Ncomms paper, Li et al., may or may not be correct. However, it is well known that metal oxide surfaces can be hydroxylated. This is especially true for C3S which will be in equilibrium with a supposedly high pH solution as already illustrated by Pustovgar et al. Figure 4. Irrespective of the correctness of the paper by Li et al., I am not convinced how your paper brings a substantial knowledge. It is simply a longer simulation of your old paper, now you are able to see the dissolution dynamics for a longer time, which is of course interesting. The hydroxylation of the surface of C3S was already discussed by Pustovgar et al., however not to this dynamic details and reactive environment. But that is something Li et al. have done, whether it is correctly done or not. So, in my opinion I do not find something substantially new or significant, but very good scientific work.

I do appreciate that ReaxFF can give us how the hydroxylation of oxide ions is important, however if the work of Li et al. were based on Pustovgar et al., this would have been already done, although I do not know the relevance, see the next comment. So, can one say that biased simulations will lead to wrong conclusions and this paper is 'very accurate'?

In the work of Li et al. it is (111) surface. Here it is (010) surface. I guess Manzano et al. had investigated such surfaces? How are these reactive oxide ions arranged in these two cases on the surface? Would that have affected this protonation of oxides as claimed to be incorrectly done in the Li et al. paper? Similar to the limitation of 40ns, their simulation could also have the limitation of not converging to the underlying free energy surface. How do you compare these two then?

At the moment, one of the key highlights of this paper is that they have opened up a new avenue for cement chemists to look at the dissolution which is realistic and not incorrect as it is non-biased. I am not even sure if you are comparing the correct surfaces to claim one is wrong and the other is correct! Probably the mechanism might be similar, but the comparison is not justified this way.

In short, I still think there is a lack of novelty in this work, some over interpretation (which is subjective), although correct simulations. 2 to 40 ns is still not enough to cover the range of dissolution in my opinion, however it still provides some mechanisms that needs to be understood. But to the extent that it is a good progress.

Li Y, Pan H, Liu Q, Ming X, Li Z. Ab initio mechanism revealing for tricalcium silicate dissolution. Nature Communications 13, 1253 (2022).

Manzano H, Durgun E, López-Arbeloa I, Grossman JC. Insight on Tricalcium Silicate Hydration and Dissolution Mechanism from Molecular Simulations. ACS Applied Materials & Interfaces 7, 14726-14733 (2015).

Pustovgar et al. DOI: 10.1016/j.cemconres.2017.06.006

The initial stages of cement hydration at the molecular level

Xinhang Xu, Chongchong Qi, Xabier M. Aretxabaleta, Chundi Ma, Dino Spagnoli, Hegoi Manzano

Point-by-point responses to the reviewers

We thank all reviewers for their constructive comments on the article. The paper has been further improved following their suggestions. We have responded to the reviewers' point-by-point responses in **blue** below, and have indicated which text in the manuscript was revised in **red**.

Reviewer #1 (Remarks to the Author):

Key results:

The Revised manuscript explains the significance of the results more elaborately. The authors have improved the manuscript by extending the simulation from 30 ns to 40 ns, additional supplementary information, and extending the explanation of the proposed comments. All the corrections are done regarding visualization.

It is understandable for the Ca dissolution from β -C2S will be computationally expensive using the current method (unbiased MD simulation). Still, the revised version has explained the early hydration phenomena, especially dissolution. However, the Authors would consider a detailed study of the belite system in future research.

The current version is recommended for publication.

We greatly appreciate the reviewer's positive comments on our manuscript. Thanks again for the efforts and time spent in reviewing our research.

Reviewer #2 (Remarks to the Author):

Authors modified the manuscript well based on reviewers' comments.

We greatly appreciate the reviewer's positive comments on our manuscript. Thanks again for the efforts and time spent in reviewing our research.

Reviewer #3 (Remarks to the Author):

In the revised manuscript, differences from other studies and contributions to initial cement hydration are emphasized. However, it is doubtful to bring many valuable and novel findings to investigation of cement hydration, particularly due to the lack of experimental validation for the proposed two-step dissolution mechanism of Ca ions. And the following points may be helpful for improving this article.

Sincerely thank you for your suggestions. In this work, our aim is to shed light to a complex process at

the nanoscale that cannot be characterized by experimental methods. It has been long recognized that understanding the nanoscale structure, properties, and mechanism of materials can help us to manipulate and improve their macroscopic properties. In cement, the relationship between nanoscale and macroscale is not trivial due to the multiscale and multicomponent nature of the material. But it is certain that without nanoscale knowledge we cannot know if we are missing something important. As described in the manuscript (Discussion section and Supplementary Note 3), we conducted unbiased simulations in order to ensure the accuracy of the simulations. Therefore, we are convinced that this work has provided more understanding in initial cement hydration. At the same time, we think that these results could be relevant for any other Ca-based mineral. That includes biomaterials like Ca phosphates, technologically relevant materials like Ca sulphates, and Ca-carbonates which have a huge interest in carbon capture and sequestration.

Regarding the lack of experimental validation, it must be noted that there is no experimental technique capable to characterize the dissolution process with enough resolution to validate the atomic scale mechanism obtained by simulations. As in many other cases, simulations are used here to investigate processes that cannot be followed by experimental techniques, being a complementary tool to build the whole picture.

1) Line 114–130: for the fluctuating curves of Ca-O_w bonds and O-H_{ab}s bonds, the incremental value or proportion can cause significant error. The fitting slope between 1.5–6ns are still preferred to be adopted.

We have calculated the fitted slopes to represent the fluctuation curves of Ca-O_w and O_w-H_{ab} bonds in the revised manuscript.

“Notably, an increase of O_w-H_{ab} bonds (0.78 new bonds nm⁻² ns⁻¹) is observed in M3-C₃S (010) between 1.5–6 ns, which is identical to the increase in the number of Ca-O_w bonds. In β-C₂S (100), an increase of 36 O_w-H_{ab} bonds (0.79 new bonds nm⁻² ns⁻¹) occurs at 1.5–6 ns, which is 1.9 times the number of increased Ca-O_w bonds (0.41 new bonds nm⁻² ns⁻¹).”

2) The proposed Ca-O_w-H_{ab}-Ca structures would give the impression that the H_{ab} are bonded to Ca, but in fact as displayed in Supplementary Figure2, the H_{ab} are bonded to O_w. Ca-O_wH_{ab}-Ca may be a more appropriate expression.

We have modified Ca-O_w-H_{ab}-Ca to Ca-O_wH_{ab}-Ca throughout the revised manuscript.

3) Line 363–370: the fourth difference seems to be repetitive from the third one, and the cited references (in line 366 and 370, respectively) are different.

Thank you for your careful review. We have combined the third and fourth points and corrected the references.

“Third, the simulation models in this study are more realistic. The simulation employs a large slab model (~23000 in this study compared with <2000 in the literature¹⁹) together with the ReaxFF reactive force field, which allows for dynamic bonds formation and breaking, and accurately describes atomic motion in complex chemical environments^{34, 35}. Meanwhile, we do not use any accelerated dissolution method, i.e. adding biasing forces or energies to the system. Specifically, a large number of chemical reactions have

occurred at the surfaces before Ca ions dissolution, and amorphous products are continuously formed at the surfaces, which is consistent with the real Ca ions dissolution process³². A detailed comparison of this work with two representative papers is provided in Supplementary Information (Supplementary Note 3).”

4) Supplementary Figure9: the atomic density distributions of Ca, Si, Os at 1.5ns show weird shifts compared to those at 0.1ns, 6ns and 40ns.

Thank you for your careful review. We have double-checked all data in Supplementary Fig. 9 and corrected the errors in it. The error comes from the structural density at 0.1 ns, which causes the illusion of a weird shift at 1.5 ns.

Supplementary Figure 9. The solid/water interface of the initial hydration process with time. The atomic and structure density of M3-C₃S (010) at **a** 0.1 ns, **b** 1.5ns, **c** 6 ns, and **d** 40 ns. The density is averaged 1 ps before different timelines to avoid excessive volatility.

We greatly appreciate the valuable suggestions and comments, which we have carefully revised to improve the manuscript quality. Thanks again for the efforts and time spent in reviewing our research.

Reviewer #4 (Remarks to the Author):

Thank you for the elaborate replies. Following are my comments -

The Ncomms paper, Li et al., may or may not be correct. However, it is well known that metal oxide surfaces can be hydroxylated. This is especially true for C3S which will be in equilibrium with a supposedly high pH solution as already illustrated by Pustovgar et al. Figure 4. Irrespective of the correctness of the paper by Li et al., I am not convinced how your paper brings a substantial knowledge. It is simply a longer simulation of your old paper, now you are able to see the dissolution dynamics for a longer time, which is of course interesting. The hydroxylation of the surface of C3S was already discussed by Pustovgar et al., however not to this dynamic details and reactive environment. But that is something Li et al. have done, whether it is correctly done or not. So, in my opinion I do not find something substantially new or significant, but very good scientific work.

Sincerely thank you for your suggestions on our research. We noticed that most of the comments are related to the comparison with the work published in Nature Communications paper by Li et al. We believe that the Nature Communications paper by Li et al. has made a great contribution to the field, by providing possible reaction pathways of the cement hydration. In our previous revision, we made statements too strong about the correctness of the results that we have to nuance in this version to be fair. The results from Li et al are not incorrect within the selected model and methodology. What we want to stress is that the model has important limitations that we try to overcome in the current study. We are trying to shed light on the cement hydration mechanism from different perspectives using MD simulations. As indicated in our previous response, it is believed that the main findings of this study have brought important progress in the investigation of the initial cement hydration.

To avoid any misinterpretation, we have focused on methodology comparison and removed all over-interpretations about the Li et al. study ¹. Below we continue answering this question point by point.

Regarding C₃S surface hydroxylation, the reviewer states that “it is well known that metal oxide surfaces can be hydroxylated. This is especially true for C3S which will be in equilibrium with a supposedly high pH solution as already illustrated by Pustovgar et al. Figure 4.” We agree with this, and the changes that such hydroxylation induces on the surface are indeed a key point of the work. The hydroxylation is not superficial as in other oxides including dicalcium silicate (Tao, Y., Zare, S., Wang, F., & Qomi, M. J. A. (2022). Atomistic thermodynamics and kinetics of dicalcium silicate dissolution. *Cement and Concrete Research*, 157, 106833.). The hydroxylation degree is high, and modifies the surface structure to a great extent, creating a thick layer of hydrated material, and distorting the surface to a point that the crystalline structure is lost. The resulting surface stoichiometry and arrangement is very different than the crystalline cleavage used in most simulation studies (including Li et al.) and consequently the properties will also be different. Therefore, a proper surface structure description is key to understand not only the dissolution process but also other relevant issues like the adsorption of

additives as temperature reducing admixtures, superplasticisers, etc.

Regarding the novelty, the reviewer says that “It is simply a longer simulation of your old paper, now you are able to see the dissolution dynamics for a longer time, which is of course interesting. The hydroxylation of the surface of C3S was already discussed by Pustovgar et al., however not to this dynamic details and reactive environment”. It is true that the methodology is the same as in our previous paper, but extending the simulation time gave us novel and relevant insights. As an analogy, TEM images of hydrated cement grains have been collected for more than 10 years, and yet, new studies with increased resolution give more detailed information about the process. Nowadays almost any paper could be described as incremental, but it is important to analyze if the increment brings new insight, which we believe is the case. To support the importance of the work, we would like to mention that our previous publication has been awarded with Lechatelier Awards, which illustrates the value of the research for the community.

I do appreciate that ReaxFF can give us how the hydroxylation of oxide ions is important, however if the work of Li et al. were based on Pustovgar et al., this would have been already done, although I do not know the relevance, see the next comment. So, can one say that biased simulations will lead to wrong conclusions and this paper is ‘very accurate’ ?.

Let us first refer to the last sentence “So, can one say that biased simulations will lead to wrong conclusions and this paper is ‘very accurate’?.” **As we said before and we want to stress here, the investigation by Li et al. is not wrong.** We believe in the accuracy of their methodology. What we propose is that the model is not as realistic as it should be to mimic C3S dissolution. Therefore, the conclusions are valid for their observations, but might be wrong regarding to the real dissolution mechanism and the quantification of the energy barriers. Regarding to this paper, we cannot claim it to be “very accurate” from a quantitative point of view, as we are using an empirical force field. Although ReaxFF has proven to be accurate in most cases, it is impossible to benchmark a model with >1K atoms as the used here with DFT. In any case, from a qualitative point of view, the conclusions from this paper undoubtedly suppose an advance, as the model is more representative than that of Li et al.

Regarding the statement that “the work of Li et al. were based on Pustovgar et al., this would have been already done”, we must express our disagreement. In the methodology section, it is stated that the surface of C3S was pre-hydroxylated. However, the figures show that the hydroxylation is very superficial, with few dangling OH groups and the system retains the crystallinity. Despite the good intentions to mimic the hydroxylated surface, the time limitations of DFT (hundreds of ps) lead to a surface structure that does not match with the experimental observation of Pustovgar et al. which state that “hydroxylated Q0 (h) species are predominant at particle surfaces during the induction period”. In their Nature Communications paper they detect a major hydroxylation “of the initial sample (that is, non-hydrated)”. Therefore, the simulations conducted by Li et al. were not on realistically hydroxylated surfaces.

In the work of Li et al. it is (111) surface. Here it is (010) surface. I guess Manzano et al. had investigated such surfaces? How are these reactive oxide ions

arranged in these two cases on the surface? Would that have affected this protonation of oxides as claimed to be incorrectly done in the Li et al. paper? Similar to the limitation of 40ns, their simulation could also have the limitation of not converging to the underlying free energy surface. How do you compare these two then?

With all respect, the fact that the surface cleavage is different does not affect the qualitative comparison between works. The conclusion by Manzano et al. (Manzano H, Durgun E, López-Arbeloa I, Grossman JC. Insight on Tricalcium Silicate Hydration and Dissolution Mechanism from Molecular Simulations. ACS Applied Materials & Interfaces 7, 14726-14733 (2015).) was that all the surfaces will undergo amorphization due to an important hydroxylation. The atomic arrangement may change the very initial hydration rate, on the order of few ps, but after 1 ns all the surfaces would virtually lose their crystal structure and converge to a similar hydrated layer. Therefore, irrespective of the cleavage, the reaxFF simulations indicate that the hydroxylation degree used by Li et al. is considerably lower than the expected. Again, this result is supported by the experiments from Pustovgar et al.

Regarding the statement “their simulation could also have the limitation of not converging to the underlying free energy surface” that is in fact another way of expressing our concerns about the model of Li et al. It is clear that the initial dissolution process is out of equilibrium, until the pore solution reaches a certain saturation level that we don't reach. That is why our analysis focuses on the evolution of the structure and not on thermodynamic properties. A thermodynamic property like the free energy barrier can only be evaluated for a “local” event, but the remaining part of the system continues evolving. As we illustrate here with the new calculations the free energy (or better said the potential of mean force PMF) depends very much on the evolution of neighboring atoms, and that is why it is crucial to have a good estimation from unbiased simulations.

Finally, we do not compare the results from Li et al. and our results. The results cannot be compared quantitatively because of the huge differences of methodology and model. What we do say is that the model used here is more realistic and therefore more suitable for a description of the dissolution mechanism.

At the moment, one of the key highlights of this paper is that they have opened up a new avenue for cement chemists to look at the dissolution which is realistic and not incorrect as it is non-biased. I am not even sure if you are comparing the correct surfaces to claim one is wrong and the other is correct! Probably the mechanism might be similar, but the comparison is not justified this way.

Sincerely thanks for your kind comments. As we mentioned in the previous point, we are not comparing the data from both papers, we are comparing the methodology and the model, which is in our opinion independent of the studied surface. Nevertheless, we have removed all ‘over’-interpretations about the Li et al. study from the paper and we have soften our statements of what is correct and what is not, as we are not holding the absolute true. Further research and methodology development will be needed in the coming years to be able to perform larger and more accurate simulations.

In short, I still think there is a lack of novelty in this work, some over interpretation (which is subjective), although correct simulations. 2 to 40 ns is still not enough to cover the range of dissolution in my opinion, however it is still provides some mechanisms that needs to be understood. But to the extent that it is a good progress.

We greatly appreciate the valuable suggestions and comments. We hope that the novelty and importance of the study will be more clear in this second round. Specially with regard to the comparison with the work of Li et al. We have reduced the subjective interpretation and substituted strong statements, which did not express properly our points. Thanks again for the efforts and time spent in reviewing our research.

Li Y, Pan H, Liu Q, Ming X, Li Z. Ab initio mechanism revealing for tricalcium silicate dissolution. *Nature Communications* 13, 1253 (2022).

Manzano H, Durgun E, López-Arbeloa I, Grossman JC. Insight on Tricalcium Silicate Hydration and Dissolution Mechanism from Molecular Simulations. *ACS Applied Materials & Interfaces* 7, 14726–14733 (2015).

Pustovgar et al. DOI: 10.1016/j.cemconres.2017.06.006

References

1. Li Y, Pan H, Liu Q, Ming X, Li Z. Ab initio mechanism revealing for tricalcium silicate dissolution. *Nature Communications* **13**, 1253 (2022).
2. Pustovgar E, *et al.* Influence of aluminates on the hydration kinetics of tricalcium silicate. *Cement and Concrete Research* **100**, 245–262 (2017).
3. Qi C, *et al.* Ab initio calculation of the adsorption of As, Cd, Cr, and Hg heavy metal atoms onto the illite(001) surface: Implications for soil pollution and reclamation. *Environmental Pollution* **312**, 120072 (2022).

REVIEWER COMMENTS

Reviewer #3 (Remarks to the Author):

Authors modified the manuscript well based on reviewers' comments.

Reviewer #4 (Remarks to the Author):

Thank you for the detailed replies.

Regarding the novelty, my point was that the hydroxylation and corresponding surface features are proposed/explored by Pustovgar et al. CCR 2017 already. Of course, not to this extent here where making and breaking of bonds are possible with ReaxFF. Hence, my comment that we already know about hydroxylation degree that you are observing.

I do not understand the argument on Q0(H) and crystallinity. Not sure if Li et al. is relevant here since they don't get the Q0 species that is measured for C3S at several minutes of hydration and not in picoseconds or nanoseconds time frame.

The authors wrote "Therefore, irrespective of the cleavage, the ReaxFF simulations indicate that the hydroxylation degree used by Li et al. is considerably lower than the expected. Again, this result is supported by the experiments from Pustovgar et al."

-I'm sorry which Pustovgar et al. experiment? NMR? Do you mean to say that the Q0(H) signals from NMR show this ns time frame of hydroxylation?

Regarding the surface plane that is considered here and in Li et al. : Are the authors saying that irrespective of the surface direction, the dissolution mechanism is the same? Or in other words, the structural mechanism that they propose are true for all the surfaces or any material with calcium? That is, if we were to study calcite dissolution, wouldn't the same calcium dissolution mechanism be observed, with calcium being the most mobile cation in all these systems? What are we learning new? Perhaps a discussion would be interesting here in my opinion. Wouldn't this be similar to an inner and outer Helmholtz plane adsorption of ions?

Isn't the hydroxylation potential of the oxygen/Oi the rate determining for the dissolution and to be looked at rather than calcium dissolution mechanism? In other words, would the dissolution or mobility of Ca²⁺ complex be directly related to the amount of 'free OH' and its mobility? That can give the difference between the C3S and C2S structures.. This is linked to my next comment on Figure 1.

In Figure 1, The number of Ca-Ow bonds is nearly half for C2S compared to C3S. Would it be related to the lower calcium content for C2S? If you normalise with respect to the surface calcium density, then would it make more sense to compare?

The initial stages of cement hydration at the molecular level

Xinhang Xu, Chongchong Qi, Xabier M. Aretxabaleta, Chundi Ma, Dino Spagnoli, Hego Manzano

Point-by-point responses to the reviewers

We thank all reviewers for their constructive comments on the article. The paper has been further improved following Reviewer #4 suggestions. We have responded to the reviewers' point-by-point responses in **blue** below, and have indicated which text in the manuscript was revised in **red**.

Reviewer #3 (Remarks to the Author):

Authors modified the manuscript well based on reviewers' comments.

We greatly appreciate the reviewer's positive comments on our manuscript. Thanks again for the efforts and time spent in reviewing our research.

Reviewer #4 (Remarks to the Author):

Thank you for the detailed replies.

Regarding the novelty, my point was that the hydroxylation and corresponding surface features are proposed/explored by Pustovgar et al. CCR 2017 already. Of course, not to this extent here where making and breaking of bonds are possible with ReaxFF. Hence, my comment that we already know about hydroxylation degree that you are observing.

Sincerely thank you for your suggestions on our research. As stated by the reviewer, we used ReaxFF to characterize the cement hydration process, including the surface hydroxylation stage and the Ca dissolution stage, at the molecular level. The detailed characterization up to 40 ns at the molecular level brought us a lot of new insights, which have been described in great detail in previous responses. Moreover, new findings and differences with other calculations in the literature have also been summarized in previous response (we will not paste all related content again to save your time). These new findings are exactly the important things that experimental studies cannot provide insight into. Simulations are used here to investigate processes that cannot be followed by experimental techniques, being a complementary tool to build the whole picture. Therefore, we strongly believe that this work gives us a better understanding of initial cement hydration, which is valuable to the society.

I do not understand the argument on $Q_0(H)$ and crystallinity. Not sure if Li et al. is relevant here since they don't get the Q_0 species that is measured for C3S at several minutes of hydration and not in picoseconds or nanoseconds time frame.

The authors wrote "Therefore, irrespective of the cleavage, the ReaxFF simulations indicate that the hydroxylation degree used by Li et al. is considerably lower than

the expected. Again, this result is supported by the experiments from Pustovgar et al.”
-I’ m sorry which Pustovgar et al. experiment? NMR? Do you mean to say that the Q⁰(H) signals from NMR show this ns time frame of hydroxylation?

Firstly, we note that all the comparisons in our manuscript and supporting material with the Li et al. study focus only on the methodology. In other words, the discussion about the hydroxylation characterization is not closely related to the main text of the revised manuscript. Here, we are pleased to provide additional discussion of hydroxylation with the reviewer.

To avoid misunderstandings, we reinterpret the concept of Q⁰(h) according to reference¹. Q⁰(h) represents a specific hydroxy silicate structural unit present on the surface of unhydrated Ca₃SiO₅ particles. The literature¹ states that "The ²⁹Si{¹H} CPMAS NMR measurements of the initial sample (that is, non-hydrated) establish the presence of Q⁰ silicate species in proximity to protons (henceforth labelled Q⁰(h)) on Ca₃SiO₅ particle surfaces, even before contact with bulk water." Therefore, according to the description in the literature, a certain degree of hydroxylation already exists on the surface before the Ca ion dissolution.

In our study, Q⁰(h) (which we denote by the O_{si}-H bond) is already present before 0.2 ps (0.0002 ns), which is consistent with the qualitative statements in the literature¹. However, Q⁰(h) was not observed when the Ca_a ion was in the (3, 2) coordination (Fig. 2d) in the study of Li et al.². Therefore, we consider that our study might be closer to the real initial cement hydration process.

Finally, it should be clarified again that we do not quantitatively compare the degree of hydroxylation with the previous literature in the manuscript. The hydroxylation before Ca dissolution in the current study was supported by the experimental observation, and our comparison with the Li et al. study only focuses on the difference in methodology.

Figure 2d. Dissolution mechanism of Ca_a from the Ca₃SiO₅ surface (Li et al.²).

Regarding the surface plane that is considered here and in Li et al. : Are the authors saying that irrespective of the surface direction, the dissolution mechanism is the same? Or in other words, the structural mechanism that they propose are true for all

the surfaces or any material with calcium? That is, if we were to study calcite dissolution, wouldn't the same calcium dissolution mechanism be observed, with calcium being the most mobile cation in all these systems?

The dissolution mechanisms in our study are derived from two representative surfaces of C_2S and C_3S . Yet, the characterization process is not site-specific. For example, we did not distinguish the Ca ions with different initial coordination, and all Ca ions in C_3S undergo a new dissolution process, etc. Therefore, we believe the obtained dissolution process is somewhat general and can serve as an important reference when investigating other surfaces and Ca-based materials. Actually, the comments raised by the reviewer apply to almost all studies in the literature, since a specific study can be continuously expanded to make it more general. Thus, we cannot make statements that the revealed dissolution process is definitely applicable to other surfaces, or even Ca-based materials, and detailed explanation can be found in the following.

Regarding whether these mechanisms apply to all low-index surfaces of C_2S and C_3S . We are tempted to say yes, because in general different low-index surfaces tend to affect only the hydration rates, and the hydration mechanisms may be very similar. However, we can't say that since we did not investigate all 14 low-index surfaces of C_2S and C_3S . This is due to the fact that the ReaxFF calculation is very computational-intensive. The 40 ns simulation of the two surfaces in this study used about 6 million core hours spanning over ~2 years. In terms of time and computational resources, a lateral extension study of all surfaces is impractical for the current manuscript. As with all other studies, the simulation of initial cement hydration is also progressive, and the above issues will be further investigated in the future.

Regarding whether the dissolution mechanism applies to any other material with calcium. We think that if the material is composed exclusively of Ca, Si, and O elements, the mechanisms may be very similar, as discussed above. If the material constituent other elements, i.e., Al, we maintain our conservative view: this requires further detailed investigations. However, we always believe that the current dissolution mechanism is still highly informative.

In the revised manuscript, we have made it clear that further validation is needed when we are applying the revealed mechanism to other low-index surfaces or Ca-based minerals.

“The authors note that additional validation is necessary to determine whether the revealed dissolution process can be utilized on other low-index surfaces of C_2S and C_3S , as well as other Ca-based minerals.”

What are we learning new? Perhaps a discussion would be interesting here in my opinion.

Currently, the initial cement hydration mechanism is still obscure. Many scholars, including us and Li et al., are expecting to further understand the initial cement hydration process and provide new insights into the dissolution mechanism. Our study makes four main contributions to the initial cement hydration, namely a new Ca dissolution process, two new general dissolution pathways, a key structure for Ca ions dissolution, and a detailed characterization of the hydration process. The main findings

have been summarized in the revised manuscript and explained in our previous responses, which we believe is adequate to address this comment.

Wouldn't this be similar to an inner and outer Helmholtz plane adsorption of ions?

Although the process of initial cement hydration certainly includes an inner and outer Helmholtz plane adsorption of ions, it still includes other processes such as the surface reconstruction and the transfer of protons to the interior surface. As stated in the literature², the initial cement hydration process is influenced by multiple coupled factors. Therefore, the initial cement hydration process cannot be simply summarized as an inner and outer Helmholtz plane adsorption of ions process.

Isn't the hydroxylation potential of the oxygen/Oi the rate determining for the dissolution and to be looked at rather than calcium dissolution mechanism? In other words, would the dissolution or mobility of Ca²⁺ complex be directly related to the amount of 'free OH' and its mobility? That can give the difference between the C3S and C2S structures.. This is linked to my next comment on Figure 1.

We have further verified whether the dissolution or mobility of Ca²⁺ complexes is directly related to the amount of 'free OH' and its mobility. We counted the number of 'free OH' in the 10 Å region above the surface (Fig. 1). We found that in the early stage (before 0.001 ns), the 'free OH' can be clearly observed (number >= 2), but almost disappear after 0.01 ns. In other words, only a small amount of 'free OH' may be present near the surface during the dissolution of Ca. Therefore, there may not be a direct correlation between the free OH and Ca ion dissolution.

Fig. 1. The number of free OH. Note we counted the number of free OH at ten representative timelines along with the dissolution process.

In Figure 1, The number of Ca-Ow bonds is nearly half for C2S compared to C3S. Would it be related to the lower calcium content for C2S? If you normalise with respect to the surface calcium density, then would it make more sense to compare?

Thanks for your suggestions. The number of Ca ions on the C_2S and C_3S surfaces are 64 and 81, respectively. As suggested, the normalized Ca- O_w bonds with respect to the surface calcium density can be plotted (Supplementary Fig. 2). We find that it does not affect our main conclusion, that is the hydration rate of C_3S is higher than that of C_2S . The normalized Ca- O_w results have been added into the supplementary materials, and a brief discussion has been added in the revised manuscript.

Regarding C_3S forming more Ca- O_w bonds than C_2S . We think that the following two reasons have a greater effect on the number of Ca- O_w bonds than the surface calcium density. Firstly, in C_3S , some Ca ions undergo a second dissolution process and these calcium ions form six Ca- O_w bonds. Secondly, the detachment of Ca ions from the surface creates vacancies that allow O_w to form new Ca- O_w bonds with the second layer of Ca ions. This further increases the number of Ca- O_w bonds in C_3S . The above explanations have been summarized in the revised manuscript.

We have also normalized the results with respect to the surface area during the discussion about the hydration rate of C_2S and C_3S in the revised manuscript. We believe that normalization with respect to the surface area will further confirm our main conclusion, that is the hydration rate of C_3S is higher than that of C_2S , is not affected by the surface area or surface calcium density.

Supplementary Figure 2. The number of normalised Ca- O_w bonds of M3- C_3S (010) and β - C_2S (100) with respect to initial surface calcium density. Note that a calcium atom was considered to be a surface calcium atom when it was exposed to the vacuum. The total number of initial surface calcium atoms from M3- C_3S (010) and β - C_2S (100) were 81 and 64, respectively.

“Notably, we find that the more rapid increase of Ca- O_w bonds around the M3- C_3S (010) remains valid when the influence of initial surface calcium density is excluded (Supplementary Fig. 2).”

References

1. Pustovgar E, *et al.* Understanding silicate hydration from quantitative analyses of hydrating tricalcium silicates. *Nature Communications* **7**, 10952 (2016).
2. Li Y, Pan H, Liu Q, Ming X, Li Z. Ab initio mechanism revealing for tricalcium silicate dissolution. *Nature Communications* **13**, 1253 (2022).

REVIEWER COMMENTS

Reviewer #4 (Remarks to the Author):

Thank you again.

Regarding the discussion on Qo(H), please note that in the earlier reply the authors had mentioned that Li et al. has some pre-hydroxylation? However in the latest reply the authors say that 'QO(H) is present in their simulations before 0.2 ps but that was not found in Li et al. specifically in Figure 2d'.

But Li et al. methods section, as the authors themselves pointed out, says some pre hydroxylation? As per NMR, this pre-hydroxylated species is then visible. I disagree on this claim by the authors that the current work is closer to the experiment. This is kind of where I think that the novelty in this paper is not there or questionable.

The other point is regarding the number of Ca-O water bonds normalized to the number of calcium ions on the surface. Now, if I compare the new figure in the supplementary and Fig 1b, it seems like that Fig 1b, is misleading? I feel like the normalized curve shows there is not much difference between C3S and C2S contrary to the authors' claim that "Notably, we find that the more rapid increase of Ca-Ow bonds around the M3-C3S (010) remains valid when the influence of initial surface calcium density is excluded (Supplementary Fig. 2)."?

At 2 ns, the figure with the normalized Ca-Ow bonds shows the same value of ~2.2 bonds for both C3S and C2S... In short, I'm worried if your 40ns simulation actually shows any difference in relative reactivity? How does one compare to the observed experimental difference?

Sorry if I'm asking too much as I'm not really sure (or worried) if you are actually providing an evidence towards different reactivities for C3S and C2S at the molecular level. Would it be possible to plot Ca-Ow, Ca-Oh and Ca-O si as a normalized and non normalized plot with time?

In continuation to the comment above: Fig 1c, on the other hand is interesting but is it the total number of dissolved ca ions (no Ca-Os bonds by your definition?). Although, it can still be bound on the surface by OH ions as the authors have mentioned in their figures... so not really dissolved and free ionic species in solution... Please clarify this if I got it right..

Lastly, sorry for the confusion: by free OH, I mean hydroxyl ions. Especially in the case of C3S there should be more hydroxyls which are mobile compared to a covalently bonded Si-OH..

The initial stages of cement hydration at the molecular level

Xinhang Xu, Chongchong Qi, Xabier M. Aretxabaleta, Chundi Ma, Dino Spagnoli, Hegoi Manzano

Point-by-point responses to the reviewers

We thank all reviewers for their constructive comments on the article. The paper has been further improved following Reviewer #4 suggestions. We have responded to the reviewers' point-by-point responses in **blue** below, and have indicated which text in the manuscript was revised in **red**.

Reviewer #4 (Remarks to the Author):

Thank you again.

Happy New Year! We sincerely appreciate the effort and time you have spent in reviewing our research throughout 2023. Wish you a happy and productive 2024!

Regarding the discussion on Qo(H), please note that in the earlier reply the authors had mentioned that Li et al. has some pre-hydroxylation? However in the latest reply the authors say that

‘QO(H) is present in their simulations before 0.2 ps but that was not found in Li et al. specifically in Figure 2d’ .

But Li et al. methods section, as the authors themselves pointed out, says some pre hydroxylation? As per NMR, this pre-hydroxylated species is then visible. I disagree on this claim by the authors that the current work is closer to the experiment. This is kind of where I think that the novelty in this paper is not there or questionable.

Thanks for pointing out the linguistic error in our response. We have carefully re-read the article by Li et al. Indeed, as the reviewer stated, Li et al.'s method took into account the role of initial hydroxylation.

However, we must emphasize that our manuscript and supplementary materials do not contain the comparison of hydroxylation characterization with the article by Li et al. Though there might be certain differences in the hydroxylation degree before Ca dissolution, our discussion regarding Li et al. is confined strictly to their methodology. From our perspective, the hydroxylation comparison doesn't influence the main novelty/findings of the current study, as explained in the following.

Regarding the comparison with the experiment and the novelty/findings of the current study, we have already provided detailed discussions from various perspectives in our previous responses. These discussions have also received the consensus from the other three reviewers. Our methodology, devoid of any biased approaches such as metadynamics, has led to a more accurate description of the Ca dissolution mechanism. Additionally, the paper's innovation and contribution are also characterized in various aspects, such as the ReaxFF reaction force field, larger atomic models (from <2000 atoms in

previous studies to ~23000 in the current study), and longer simulation time (from tens of ps to 40 ns), etc. The unbiased simulation in such a long simulation time allows us to present a more realistic description of the dissolution mechanism, as detailed in the main text. Furthermore, we have included a discussion comparing our findings with the experiment in the manuscript, indicating our results are consistent with the experimental results.

In summary, we believe that innovation and contribution have been explained clearly in this response, as well as in our previous responses. We appreciate your valuable effort and time in reviewing our manuscript.

The other point is regarding the number of Ca-O water bonds normalized to the number of calcium ions on the surface. Now, if I compare the new figure in the supplementary and Fig 1b, it seems like that Fig 1b, is misleading? I feel like the normalized curve shows there is not much difference between C3S and C2S contrary to the authors' claim that "Notably, we find that the more rapid increase of Ca-O_w bonds around the M3-C3S (010) remains valid when the influence of initial surface calcium density is excluded (Supplementary Fig. 2)." ?

At 2 ns, the figure with the normalized Ca-O_w bonds shows the same value of ~2.2 bonds for both C3S and C2S... In short, I'm worried if your 40ns simulation actually shows any difference in relative reactivity? How does one compare to the observed experimental difference?

Sorry if I'm asking too much as I'm not really sure (or worried) if you are actually providing an evidence towards different reactivities for C3S and C2S at the molecular level. Would it be possible to plot Ca-O_w, Ca-O_h and Ca-O_{si} as a normalized and non normalized plot with time?

Thanks for your questions. We would like to clarify that normalizing the hydration process based on the number of surface Ca might not be suitable. For one thing, accurately defining the surface-exposed Ca is challenging. For another, since the position of Ca atoms are constantly changing during the hydration, the number of surface-exposed Ca atoms changes accordingly, making it hard to perform the accurate normalization in a dynamic manner. Therefore, we determined to normalize the hydration process according to the surface area, as suggested by Reviewer 3. The Ca-O_w result normalized by surface area is shown in Fig. S2.

Fig. S2. The number of normalized Ca-O_w bonds of M3-C₃S (010) and β-C₂S (100) with respect to the surface areas.

Based on Fig. S2, we can see a significant difference between C₂S and C₃S, which confirms that Fig. 1b is not misleading. Considering the comparison between Fig. S2 and Fig. 1b, as well as the surface area of C₂S (10.06 nm²) and C₃S (9.08 nm²), it is not necessary to normalize other results for clarity purposes.

It needs to be noted that the hydration difference between C₂S and C₃S has been well discussed based on our MD simulation within 40 ns. The difference is not only represented in the number of Ca-O_w bonds but also in their different dissolution process. The underlying reason for the hydration difference between C₂S and C₃S might be their atomic structure, for example, C₃S contains more Ca atoms in the unit cell and the surface, and a special O type (ionic O).

Though a comparison with experiments has been discussed in the main text, the major aim of our simulation is to shed light to a complex process at the molecular level that cannot be characterized by experimental methods. As in many other cases, simulations are used here to investigate processes that cannot be followed by experimental techniques, being a complementary tool to build the whole picture.

In continuation to the comment above: Fig 1c, on the other hand is interesting but is it the total number of dissolved Ca ions (no Ca-O_s bonds by your definition?). Although, it can still be bound on the surface by OH ions as the authors have mentioned in their figures... so not really dissolved and free ionic species in solution... Please clarify this if I got it right..

Yes, you are right. We have clarified the dissolved Ca ions in the manuscript.

“Dissolved Ca ions in Fig. 1c are defined as having no Ca-O_s bonds but do not represent free Ca ions in solution, which is further detailed in the 'Dissolution Process of Ca Ions' section.”

Lastly, sorry for the confusion: by free OH, I mean hydroxyl ions. Especially in the case of C3S there should be more hydroxyls which are mobile compared to a covalently bonded Si-OH..

If you mean (Ca)-OH, the answer is yes. In our manuscript, we found that (Ca)-OH is predominantly present in the ligand teeth. As noted in the manuscript, the dissolution process of Ca ions in M3-C₃S (010) tends to form more ligand teeth than β -C₂S (100). Thus, the M3-C₃S (010) has more (Ca)-OH. A detailed discussion is provided in the Dissolution process of Ca ions section.

In conclusion, we deeply appreciate your effort and time invested in reviewing our research.

REVIEWERS' COMMENTS

Reviewer #4 (Remarks to the Author):

Happy new year to the authors too and I am sorry you are stuck with me even in 2024!

1. I am not sure what kind of linguistic error is there on their response :

" QO(H) is present in their simulations before 0.2 ps but that was not found in Li et al. specifically in Figure 2d"

-All along, I worry your replies were following this pattern. It won't make sense for me to waste my time on this anymore with such replies.

2. Earlier reply: "Notably, we find that the more rapid increase of Ca-Ow bonds around the M3-C3S (010) remains valid when the influence of initial surface calcium density is excluded (Supplementary Fig. 2)".

After I pointed out that their curves were very similar and insignificant differences, now they say:

Current reply: "We would like to clarify that normalizing the hydration process based on the number of surface Ca might not be suitable. For one thing, accurately defining the surface-exposed Ca is challenging. For another, since the position of Ca atoms are constantly changing during the hydration, the number of surface-exposed Ca atoms changes accordingly, making it hard to perform the accurate normalization in a dynamic manner." But it was done for initial surface Ca density!!

Then the authors argue it is better to normalize with the surface area which I believe is not changing dynamically? It again poses the problem of dynamically changing surface structure ?? The values used is for the initial surface or the box dimensions per unit vector probably? surface area used for normalizing is not dynamic here and for some reason it is conveniently lower for C3S...

Again same pattern of replies, I'm becoming more and more worried about the paper than getting convinced.

Regarding Ca-OH, my earlier comment is still not answered: "Isn't the hydroxylation potential of the oxygen/Oi the rate determining for the dissolution and to be looked at rather than calcium dissolution mechanism? In other words, would the dissolution or mobility of Ca²⁺ complex be directly related to the amount of 'free OH ions' which are not covalently bonded and its mobility? That can give the difference between the C3S and C2S structures.. This is linked to my next comment on Figure 1." :

- The authors current reply says " the M3-C3S(010) has more Ca-OH". The simple question is why the authors didn't plot this one but everything else : Ca-Ow, Ow-H and so on. Isn't it very clear that more hydroxyls can form in C3S with the oxide ions and that is absent in C2S. Something which can be understood from Pustovgar et al. Cem Con Rés. 2017 for C3S. C3S is more reactive as it facilitates more Ca-OH bonds which will allow more conversion of Ca-Os to Ca-OH or Ca-Os bonds? I guess many DFT studies have already explored this. This is what you are essentially showing with all the ligand teeth and so on... Just plotting Ca-OH vs time would probably be a better plot than Ca-Ow..

As I have stated earlier, the work is good but just nothing novel other than that it is a longer simulation

and we can observe some of the dynamic events which is mostly seen in the Li et al. ncomms paper. their Ca-Ow count plot to show that C3S is more reactive is most probably misleading and I am still not convinced with their replies even if they go back and forth on their arguments.

The initial stages of cement hydration at the molecular level

Xinhang Xu, Chongchong Qi, Xabier M. Aretxabaleta, Chundi Ma, Dino Spagnoli, Hegoi Manzano

Point-by-point responses to the reviewers

We thank all reviewers for their constructive comments on the article. The paper has been further improved following Reviewer #4 suggestions. We have responded to the reviewers' point-by-point responses in **blue** below, and have indicated which text in the manuscript was revised in **red**.

Reviewer #4 (Remarks to the Author):

Happy new year to the authors too and I am sorry you are stuck with me even in 2024!

1. I am not sure what kind of linguistic error is there on their response :
" Q0(H) is present in their simulations before 0.2 ps but that was not found in Li et al. specifically in Figure 2d "
-All along, I worry your replies were following this pattern. It won' t make sense for me to waste my time on this anymore with such replies.

Thank you for your comments. In our previous response, we clarified that Li et al. also included a pre-hydroxylated surface in their model, as depicted in Figure 1b of their article. However, we must emphasize again that our manuscript and supplementary materials do not contain the comparison of hydroxylation characterization with the article by Li et al.. Basically, because in Li et al. there is no quantification of the OH/nm², so we cannot compare. A visual inspection of Figure 1b suggests that the number of OH groups on their surface is much lower than what we observe, probably due to the short simulation time. In fact, the formation of many OH groups induces an amorphization that is not observed in Li et al. We strongly believe that the fact that Li et al included some OH groups in the surface doesn't influence the main novelty/findings of the current study, which has been explained in previous responses.

2. Earlier reply: " Notably, we find that the more rapid increase of Ca-Ow bonds around the M3-C3S (010) remains valid when the influence of initial surface calcium density is excluded (Supplementary Fig. 2)" . After I pointed out that their curves were very similar and insignificant differences, now they say:

Current reply: "We would like to clarify that normalizing the hydration process based on the number of surface Ca might not be suitable. For one thing, accurately defining the surface-exposed Ca is challenging. For another, since the position of Ca atoms are constantly changing during the hydration, the number of surface-exposed Ca atoms changes accordingly, making it hard to perform the accurate normalization in a dynamic manner." But it was done for initial surface Ca density!!

Then the authors argue it is better to normalize with the surface area which I believe is not changing dynamically? It again poses the problem of dynamically changing

surface structure ?? The values used is for the initial surface or the box dimensions per unit vector probably? surface area used for normalizing is not dynamic here and for some reason it is conveniently lower for C3S...

Again same pattern of replies, I'm becoming more and more worried about the paper than getting convinced.

Thanks for your comments. To address this, we have provided two figures normalized to both the initial number of Ca ions and the initial surface areas. Potential readers can easily find relevant results, which will be beneficial to subsequent studies. A brief explanation is provided in the following.

During the dissolution process, more Ca-O_w bonds are observed near the surface of C₃S. A closer look at the initial surface structure revealed that there were 81 exposed Ca atoms in C₃S, compared to 64 in C₂S. Even though we normalised the Ca-O_w bonds with respect to initial surface Ca atoms (Supplementary Fig. 2b), the initial hydration of C₃S is still faster than that of C₂S. In other words, the normalization with respect to initial Ca atoms doesn't affect the major conclusions in the current study. Furthermore, the comparison of the total number of Ca-O_w bonds remains meaningful when the results are normalized with respect to the surface area, as shown in Supplementary Fig. 2a.

It must be clear that we did these 2 plots due to the requests of the reviewer. The dissolution rate of a material is always given in terms of mass per unit of time and units of surface. Normalizing with respect to the area is correct. Normalizing with respect to the exposed Ca atoms is something that we did in the initial version, which may even be correlated with the surface (a larger surface area implies more exposed Ca). But is not the correct way, and was corrected thanks to the revision process. For example, consider a material with a very low Ca/Si ratio, with 1 Ca exposed per nm². From the point of view of exposed Ca atoms, a 100% hydroxylation degree will be achieved with a single water molecule dissociation per nm². For a material with a higher Ca/Si ratio, we can have 3 water molecules per nm² without reaching 100% in terms of exposed Ca atoms. Clearly, in the second case, we have a larger hydroxylation degree of the surface (the relevant quantity), yet lower per exposed Ca.

Regarding Ca-OH, my earlier comment is still not answered: "Isn't the hydroxylation potential of the oxygen/Oi the rate determining for the dissolution and to be looked at rather than calcium dissolution mechanism? In other words, would the dissolution or mobility of Ca²⁺ complex be directly related to the amount of 'free OH ions' which are not covalently bonded and its mobility? That can give the difference between the C3S and C2S structures.. This is linked to my next comment on Figure 1." :

The hydroxylation potential of the oxygen/Oi is not the rate-determining step for the dissolution. For example, MD and DFT simulations indicate that the hydroxylation (water dissociation) energy barrier for gamma C₂S surface can be lower than that of beta C₂S, even barrier-less.[Wang, Q., Manzano, H., Guo, Y., Lopez-Arbeloa, I., & Shen, X. (2015). Hydration mechanism of reactive and passive dicalcium silicate polymorphs from molecular simulations. *The Journal of Physical Chemistry C*, 119(34), 19869-19875.]- Therefore, hydroxylation is part of the picture, we believe that it is an important one indeed, but not all of it.

- The authors current reply says “ the M3-C3S(010) has more Ca-OH ” . The simple question is why the authors didn’ t plot this one but everything else : Ca-O_w, O_w-H and so on. Isn’ t it very clear that more hydroxyls can form in C3S with the oxide ions and that is absent in C2S. Something which can be understood from Pustovgar et al. Cem Con Rés. 2017 for C3S. C3S is more reactive as it facilitates more Ca-OH bonds which will allow more conversion of Ca-O_s to Ca-OH or Ca-O_s bonds? I guess many DFT studies have already explored this. This is what you are essentially showing with all the ligand teeth and so on... Just plotting Ca-OH vs time would probably be a better plot than Ca-O_w.

Thanks for your comments. Regarding the Ca-OH (Fig. 1d), we clarify that all the O-H shown in Fig. 1d are bonded to the Ca ions. In other words, the data for Ca-OH in C₃S is equivalent to the sum of O_w-H, O_f-H, and O_{si}-H. The inclusion of various OH groups in Fig. 1d provides a more detailed characterization to potential readers about the Ca-OH bonds formed by different types of oxygen atoms. We believe that this approach enriches the dataset and offers a more complete understanding for other researchers.

As I have stated earlier, the work is good but just nothing novel other than that it is a longer simulation and we can observe some of the dynamic events which is mostly seen in the Li et al. ncomms paper. their Ca-O_w count plot to show that C3S is more reactive is most probably misleading and I am still not convinced with their replies even if they go back and forth on their arguments.

We appreciate the effort made during the review process. We also feel that the reviewer #4’s comments go back and forth around the same issues. In particular, the comparison with the work of Li et al., so we would like to write clearly again the difference:

- Li et al. used metadynamics to sample a single Ca dissolution process following a predefined mechanism imposed by the collective variable. Their surface model is likely too ordered, and there is no evolution (hydration) of the surrounding atoms in the surface while the Ca detachment takes place.
- We simulate an unbiased dissolution process, and then characterize it. We prove that Ca ion dissolution is indeed the first dissolution step, and we sample with umbrella sampling the detachment event that we observed. We show that hydroxylation reactions in neighbor O atoms trigger the dissolution reducing the detachment energy barriers.

Therefore, the simulations in Li et al. are indeed technically very good and very important information that contributes to understand the complete picture, but it is likely that the Ca detachment in a more realistic scenario simply doesn’t take place over the path that they are sampling. Our manuscript extensively discusses the novelty of our work, which has been highlighted in multiple rounds of responses as well as within the manuscript itself. This novelty has also been recognized by three other reviewers. Particularly, we have elaborated on the significant differences between our study and previous MD or Li et al. articles in detail within the Supplementary Information.

Overall, we firmly believe that this work contributes significantly to the understanding of initial

cement hydration processes and will serve as a valuable reference for researchers in the field. We sincerely appreciate your effort and time invested in reviewing our manuscript.